



# Air Pollution in The Upper Troposphere: Insights from In-Situ Airplane Measurements (1991-2018)

Kuo-Ying Wang[1], Philippe Nedelec[2], Valerie Thouret[3], Hannah Clark[4], Andreas Wahner[5], and Andreas Petzold[6]

[1]Department of Atmospheric Sciences, National Central University, Chung-Li, Taiwan
[2,3,4]Laboratoire d'Aérologie, Centre National de la Recherche Scientifique, Observatoire Midi-Pyrénées, 14 Avenue E. Belin, 31400 Toulouse, France
[5,6]Forschungszentrum Julich GmbH, Institut fur Energie und Klimaforschung, IEK-8 Troposphare, 52425 Julich, Germany

**Correspondence:** Kuo-Ying Wang (kuoying@mail.atm.ncu.edu.tw)

**Abstract.** Long-lived atmospheric species like carbon dioxide ($CO_2$), methane ($CH_4$), nitrous oxide ($N_2O$), and CFCs exhibit discernible trends reflecting anthropogenic emissions, with increases observed in $CO_2$, $CH_4$, $N_2O$, and decreases in CFCs. Conversely, trends for short-lived species like carbon monoxide (CO) and nitrogen oxides (NOx) remain less understood due to rapid chemistry and limited upper tropospheric observations. We utilize extensive in-
situ CO measurements spanning 2012-2023, supplemented by prior airplane campaigns from 1991-2019, to examine short-term fluctuations in CO influenced by anthropogenic emissions and rapid chemical removal. Comparisons with MOPITT satellite data and chemistry budgets from 1948-2003 simulations further elucidate the interplay of sources and sinks, revealing the significant impact of chemistry on CO profiles and trends.

## 1 Introduction

Given the significant scale of anthropogenic emissions originating from Asian regions since the 1970s (UNEP, 2023), there is a critical need for routine measurements of anthropogenic air pollutants in the Pacific upper troposphere. Such measurements would not only enhance our understanding of the spatial and temporal extent of anthropogenic emissions in this region but also facilitate the development and validation of models aimed at predicting the export

of air pollution from Asian sources (Petzold et al., 2015).

The impacts of atmospheric short-lived species such as carbon monoxide (CO) and nitrogen oxides (NOx) are challenging to track compared to longer-lived species like carbon dioxide ($CO_2$), methane ($CH_4$), and chlorofluoro-carbons (CFCs) (Seiler and Junge, 1970). This difficulty arises from the faster chemistry involved, compounded by the complex interactions of these short-lived species with various Earth system components such as the lithosphere,



biosphere, hydrosphere, and cryosphere, particularly within the troposphere. Moreover, the tropospheric dynamics, including temperature, wind patterns, humidity, and observational constraints, pose additional challenges in tracking the long-term trends of these short-lived species originating from anthropogenic emissions.

In the lower troposphere, elevated temperatures accelerate chemical reaction rates, while abundant gas-phase water vapor facilitates the production of hydroxyl radicals (OH), leading to shorter chemical lifetimes for short-lived

species like CO compared to the upper troposphere, where lower temperatures and drier air prevail (Seiler and Junge, 1970).

While in-situ observations of atmospheric gas-phase pollution levels in the upper troposphere over the North Pacific are scarce and challenging, satellite remote sensing techniques offer passive observations of emissions in near and thermal infrared wavelengths to infer total column density of gas-phase molecules (Martin, 2008). However,

obtaining good vertical resolution of CO in the upper troposphere remains a challenge. Ozone ($O_3$) measurements by ozonesonde provide valuable vertical profile data but are limited to ozone (Thompson et al., 2021).

The global atmosphere has been continuously impacted by significant inputs of ground-level anthropogenic pollutants since the industrial revolution (Barbante et al., 2004; Rose, 2015).

Routine and long-term in-situ measurements of Asian air pollutants over the North Pacific upper troposphere are

rare but crucial for understanding anthropogenic impacts in this region. In this study, we present routine in-situ measurements of CO from the PGGM/IAGOS air-based measurement project conducted during the period 2012-2018. Combining these measurements with previous in-situ data collected in various years between 1991 and 2019, we observed a significant increase in ground-level anthropogenic pollutant CO in the upper troposphere over the North Pacific.

Over the 28-year period, mean CO volume mixing ratios increased from 60 ppbv in 1991 to 80 ppbv in 2018, reflecting a 30% rise in observed CO mixing ratios. This increase correlates with a calculated 26% increase in ground-level CO emissions over Asian countries, from 180 MT in 1991 to 228 MT in 2018 (Wang et al., 2024).

In contrast, ground-level CO emissions over North America decreased by 51%, from 69 MT in 1994 to 35 MT in 2018. Similarly, observed CO mixing ratios over the North Atlantic upper troposphere decreased by 15%, from 80

ppbv in 1994 to 68 pptv in 2019.

The sensitivity of CO mixing ratios in the upper troposphere to ground-level CO emissions was assessed using a three-dimensional tropospheric chemistry model. Model simulations with constant ground-level CO emissions for the 20-year period of 1994-2003 demonstrated zero growth trends in CO mixing ratios over the North Pacific and North Atlantic.

Comparison of PGGM/IAGOS aircraft measurements during the 2012-2018 period with previous NASA Global Troposphere Experiment (GTE) aircraft missions revealed elevated CO mixing ratios in the upper troposphere. These elevated levels, predominantly observed during the summer months of July and August, indicate effective vertical transport of ground-level anthropogenic pollutants (Thompson et al., 1996; Blake et al., 1996, 1997; Jacob et al., 2003a, b; Streets et al., 2006; Wofsy, 2011; Brune et al., 2020).



In the following, we present new insights into air pollution in the upper troposphere, highlighting the importance of routine in-situ measurements for understanding the impacts of anthropogenic emissions on atmospheric chemistry and dynamics. By combining observational data with advanced modeling techniques, we can track and predict the spatial and temporal variations of air pollutants, thereby informing effective mitigation strategies and policies.

## 2   Data and Methods

### 2.1   In Situ Observations

### 2.1.1   PGGM/IAGOS

Figure 1(a) depicts the flight routes of three routine in-service commercial passenger Airbus aircraft operated by China Airlines (CAL) and equipped with the IAGOS Package 1 instruments for in-situ measurements from 2012 to 2018 (Petzold et al., 2015; Nédélec et al., 2015; Wang et al., 2024). These flight routes span various regions, covering
heavy emission areas over East Asia (Streets et al., 2006), upwind and downwind of Asian emission areas over the Euro-Asia continent, the North Pacific, tropical northwestern Pacific regions, as well as flights to the southern hemisphere, including Australia and New Zealand (Clark et al., 2015).

    It is noteworthy that the IAGOS project is a continuation of the MOZAIC project, which commenced in 1994 (Marenco et al., 1998). The innovative aspect of the MOZAIC project lies in its utilization of five in-service commer-
cial passenger Airbus A340-300 aircraft to routinely collect air pollutants in the upper troposphere during commercial passenger service flights. Unlike dedicated research aircraft, commercial passenger aircraft operate extensively, allowing for data collection during take-off, cruising, and landing stages, thus providing abundant data opportunities (Wang et al., 2024). This stands in stark contrast to research aircraft, which typically have limited operational windows spanning only a few days.

### 2.1.2   NASA GTE, HIPPO, and ATom Aircraft Measurements

Thanks to the visionary, courageous, and painstaking effort of David Keeling and collaborators, the Mauna Loa $CO_2$ data provide long-term, in-situ, and consistent monitoring of atmospheric $CO_2$ since March 1958 (Keeling et al., 1976; Heimann, 2005). These data serve as a crucial reference for quantifying the current status of atmospheric $CO_2$ in a climate characterized by rising temperatures and continuously increasing anthropogenic emissions (UNEP,
80 2023).

    For short-lived greenhouse gases such as CO (IPCC, 2013), long-term measurements downwind of heavy surface emission areas are rare but essential for understanding changes in air pollutants over the North Pacific upper troposphere. During the period 1970-1994, when technological emissions grew significantly year by year, the only in-situ measurements in the upper troposphere over the North Pacific were conducted as part of NASA's Global
Troposphere Experiments (GTE) (Thompson et al., 1996).





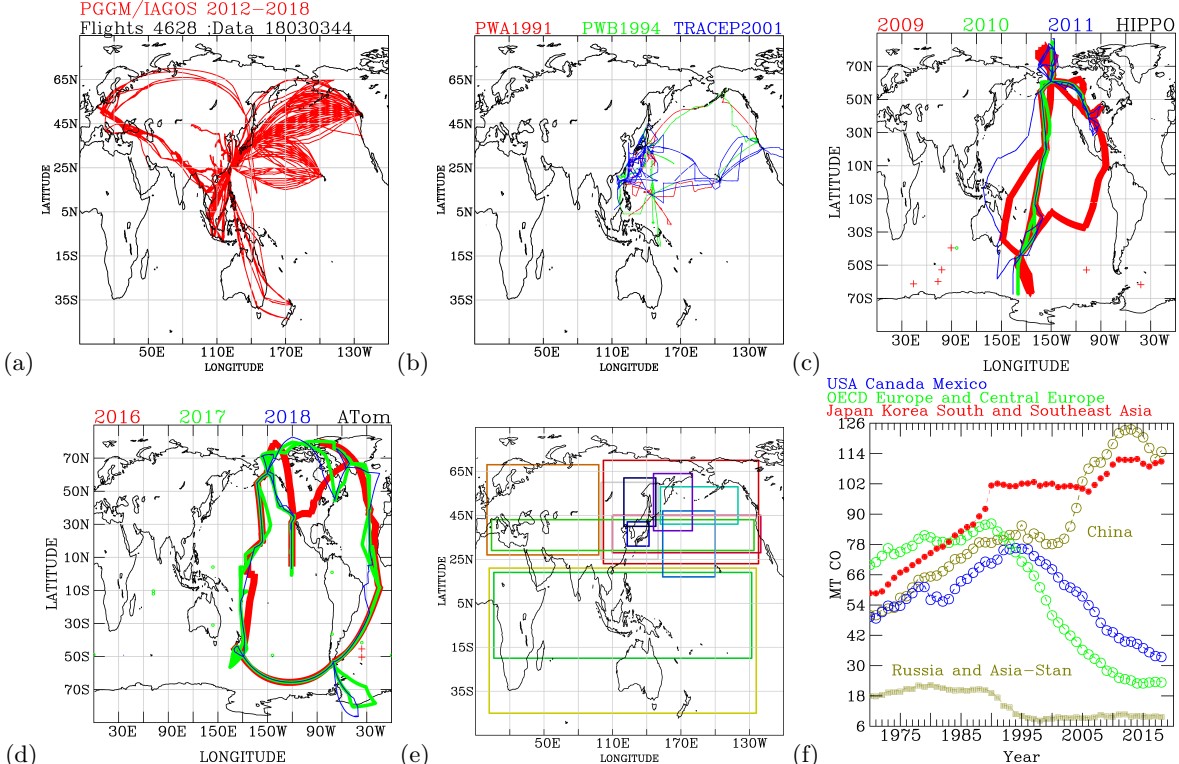

**Figure 1.** The flight routes of the measurement aircrafts and analysis domains. (a) The flight routues of the PGGM/IAGOS aircrafts during 2012-2018. (b) The flight routes of the NASA PEM-WEST A in 1991 (red), the PEM-WEST B in 1994 (green), and the TRACE-P in 2001 (blue). (c) The flight routes from the NASA HIPPO aircraft measurements during 2009 (red), 2010 (green), and 2011 (blue). (d) The flight routes from the NASA ATom aircraft measurements during 2016 (red), 2017 (green), and 2018 (blue). (e) The analysis domains. The area with intensive surface cyclogenesis (dark blue), and its dwonwind region to the central Pacific (purple), and over the entire North Paicific (sky blue). The northern hemisphere warm conveyor belt (light green in the NH), and its subdomains, including downwinds of the Asian continent, and across the entire North Pacific(magenta); in the exit of airs from Asia content over the Japan and Korea region (blue); and further downwinds, across central Pacific (navy blue). The western part of the Euro-Asia continent, including part of the warm conveyor and its northern latitudes (brown). The eastern part of the Euro-Asia continent and the North Pacific (red). The tropical regions and the sourthern hemisphere (dark green), and the tropical regions (fresh green). (f) Time-series technological emissions of carbon monoxides estimated by the EDGAR version 6.1 for UAS, OECD and Central Europe, Asian countries excluding China, China, and Russia and Central Asian countires.

Figure 1(b) illustrates the flight routes of NASA's PEM-WEST A in 1991 (Blake et al., 1996), PEM-WEST B in 1994 (Blake et al., 1997), and TRACE-P in 2001 (Jacob et al., 2003a). The PEM-WEST A and B flight routes focused on regions west of $140°E$ over the western North Pacific, near the East Asia emission areas. These flights provided valuable data on air pollutants over the North Pacific upper troposphere immediately downwind of East



Asia emission areas, especially during crucial periods such as 1991 and 1994, which marked the end of steady CO emission growth and significant growth thereafter, particularly from China (Streets et al., 2006). The visionary PEM-WEST measurements offer valuable data for comparison with later in-situ measurements from projects like IAGOS and ATom.

Figure 1(c) depicts the flight routes from NASA's HIPPO aircraft measurements during 2009, 2010, and 2011

(Wofsy, 2011), while Figure 1(d) shows the flight routes from NASA's ATom aircraft measurements during 2016, 2017, and 2018 (Brune et al., 2020). The HIPPO measurements focused on regions close to the International Dateline (ID) and longitudes east of it in the North Pacific upper troposphere. Similarly, the ATom measurements over the North Pacific focused on regions similar to those covered by HIPPO flights.

Based on these flight routes, we categorized the data into regions for analysis, as shown in Figure 1(e). These regions

represent key dynamical processes responsible for transporting ground-level air pollutants to the upper troposphere. They include regions of intensive surface cyclogenesis and their downwind areas, the northern hemisphere warm conveyor belt and its subdomains, areas of air exit from the Asian continent, as well as regions across the Euro-Asia continent and the North Pacific.

### 2.1.3   Mauna Loa Surface Long-Term Measurements

The Mauna Loa monitoring site, operated by the Global Monitoring Laboratory (GML) at the Earth System Research Laboratories (ESRL), National Oceanic and Atmospheric Administration (NOAA), provides high-quality, long-term, in-situ measurements of atmospheric CO (Andrews et al., 2009). These measurements play a crucial role as a standard reference for verifying long-term CO trends observed from satellite measurements by instruments like MOPITT and simulations by models such as the IMS model.

### 110   2.2   MOPITT Satellite Remote Sensing Measurements

Satellites serve as valuable tools for remote monitoring of CO on a continuous and global scale from low Earth orbit. The Measurement of Pollution In The Troposphere (MOPITT) instrument onboard NASA's Earth Observing System Terra has been monitoring CO globally since March 2000 (Buchho et al., 2022). The openly available MOPITT data are extensively utilized in documenting CO variations associated with various atmospheric processes. In this study,

we leverage MOPITT data as a consistent reference to assess the long-term trends of CO in the troposphere.

### 2.3   EDGAR Technological Emission Estimates

An openly available and widely used source for estimating gridded anthropogenic emissions in global chemistry and climate models is the Emissions Database for Global Atmospheric Research (EDGAR) from the European Commission (Olivier etr al., 1994).





In this study, we utilize EDGAR estimates to track variations in technological emissions of CO during the period 1970-2020. Figure 1(f) illustrates time-series technological emissions of carbon monoxide estimated by EDGAR version 6.1 for various regions including the United States (USA), OECD countries, Central Europe, Asian countries excluding China, China, and Russia and Central Asian countries. EDGAR data play a crucial role in revealing the variations of technological emissions in our analysis.

It is worth noting that accurate estimates of anthropogenic emissions are essential for quantifying the discrepancies between observed rising temperatures and anthropogenic emissions, as highlighted in the United Nations Environment Programme Emissions Gap Report 2023 (UNEP, 2023).

### 2.4 IMS Global Tropospheric Chemistry Model

The three-dimensional chemistry transport model IMS (Integrated Modelling System) is employed in this study
to simulate long-term CO trends and budgets in the troposphere. The IMS model encompasses full tropospheric chemistry, with particular focus on the conversion of hydrocarbons to CO (Wang et al., 2001b), along with a comprehensive suite of anthropogenic and natural emissions (Wang et al., 2001a). Validation of IMS model results has been conducted against observational data (Wang and Shallcross, 2000a, b; Wang et al., 2001c). Additionally, the IMS model has been utilized to investigate various phenomena, including the impact of stratospheric ozone on
tropospheric ozone (Wang and Kau, 2015) and the long-range transport of Asian dust over the North Pacific (Wang, 2007).

In this study, we conducted two long-term simulations using the IMS for tropospheric chemistry. The IMS 1948-1978 simulation spans 30 years, commencing from September 1948 to August 1978, while the IMS 1984-2003 simulation covers 20 years, starting from September 1984 to August 2003. These simulations were performed with constant
emissions from anthropogenic sources, while biogenic emissions were calculated online based on interactions among the atmosphere, ocean, and terrestrial biosphere (Wang and Shallcross, 2000b; Wang et al., 2001a).

The results from the IMS simulations were compared with in-situ measurements from IAGOS, satellite remote sensing measurements from MOPITT, and long-term in-situ measurements from Mauna Loa.

## 3 Results

### 3.1 Long-Term Time-Series CO and Trends

Figure 2 illustrates long-term time-series CO measurements downwind of the Asian emission areas and over the North Pacific upper troposphere. The upper panels depict data from high latitudinal regions north of $45°N$, spanning longitudes $150°E$ to $140°W$, and latitudes $45°N$ to $60°N$. The lower panels display data from lower latitudinal regions south of $45°N$, covering longitudes $150°E$ to $140°W$, and latitudes $25°N$ to $45°N$ (the Northern Hemisphere
warm conveyor belt, Figure 2(e)). The time-series data are stratified according to the statistics of all data collected in





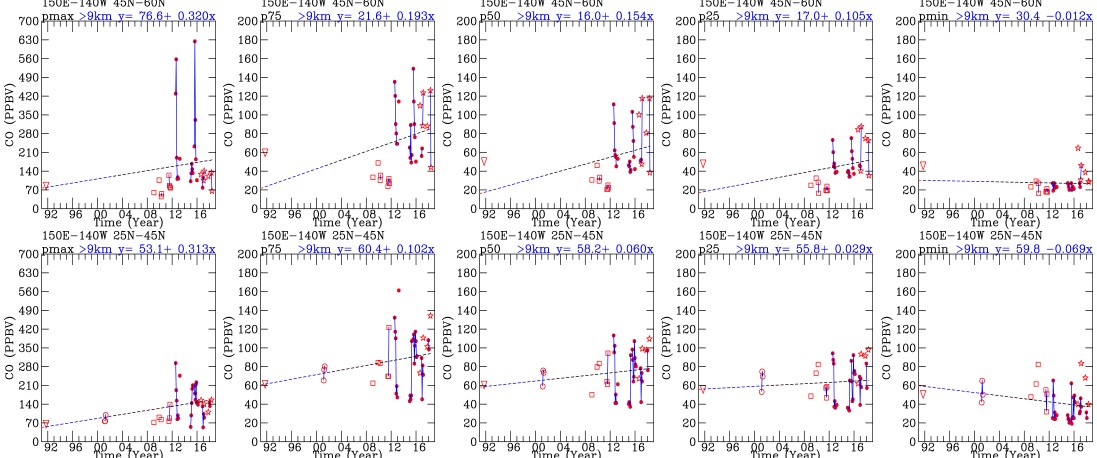

**Figure 2.** Time-series long-term CO measreuements downwind the Asia emission areas and over North Pacific upper troposphere. The upper panels shows data from the high latitudinal regions north of $45°N$, in longitudes $150°E - 140°W$, and latitudes $45°N - 60°N$. and over North Pacific upper troposphere. The lowers panels shows data from the lowe latitudinal regions south of $45°N$, in longitudes $150°E - 140°W$, and latitudes $25°N - 45°N$. and over North Pacific upper troposphere. The time-series data are stratified according to the statistics of all data collected in a given month: The maximum (left most), 75th percentile (second from left), 50th percentile (center), 25th percentile (second from the right), and the minimum (right most) observations.

a given month: the maximum (leftmost), 75th percentile (second from left), 50th percentile (center), 25th percentile (second from the right), and the minimum (rightmost) observations.

The maximum observed CO concentrations are higher over latitudes north of $45°N$ (exceeding 280 ppbv) than over latitudes south of $45°N$ (approximately 280 ppbv), indicating exceptionally high CO emission sources from

biomass burning over higher latitudes such as Siberia (Nédélec et al., 2005).

The linear regression of the 75th percentile observations exceeds 90 ppbv in lower latitudes and falls below 90 ppbv in higher latitudes. For the 50th percentile observations, the linear regression is higher in lower latitudes (close to 80 ppbv) than in higher latitudes (around 70 ppbv). Regarding the 25th percentile observations, the linear regression for CO is higher (around 70 ppbv) in lower latitudes than in higher latitudes (approximately 50 ppbv).

Regarding the minimum observed CO levels, both higher and lower latitudes exhibit a decline. The lower latitudes contain higher CO levels (around 40 ppbv) compared to the higher latitudes (approximately 30 ppbv).

In summary, key findings from the in-situ aircraft measurements over the middle North Pacific upper troposphere, downwind of the Asian emissions as shown in Figure 2, are as follows: except for the minimum CO levels, all observed CO concentrations increased over the 27-year period from 1991 to 2018. The minimum CO levels reflect background

CO, which decreased over time. The 25th, 50th, 75th, and maximum CO levels are impacted by emissions, with latitudes south of $45°N$ being more affected by Asian emissions compared to latitudes north of $45°N$.




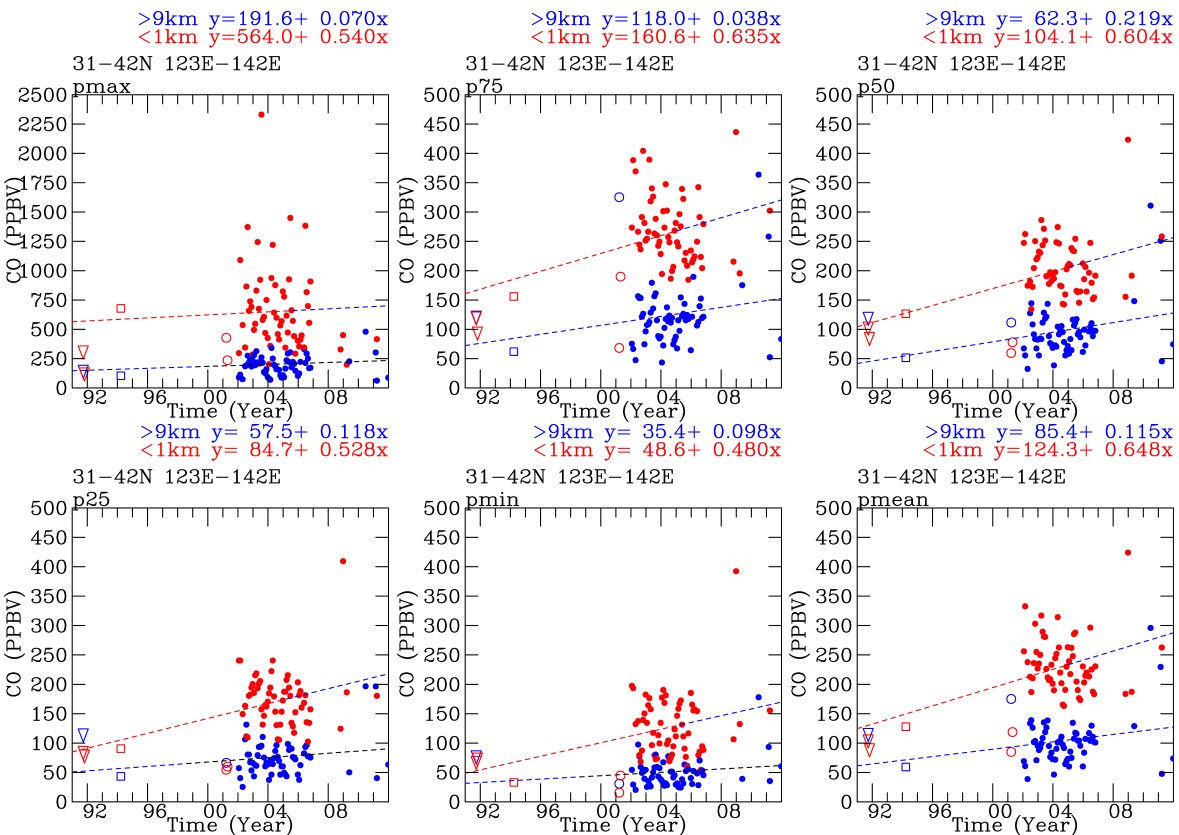

**Figure 3.** Time-series long-term CO in-situ measurements during 1991-2012 in the upper (blue) and the lower (red) troposphere over the Asian emission regions $123°E - 142°W$, and latitudes $31°N - 42°N$. The measurements are shown as the maximum (upper left), 75th (upper middle), 50th (upper right), 25th (lower left), minimum (lower middle), and the mean (lower right) of all in-situ measurements in a given month.

Figure 3 presents time-series of long-term CO in-situ measurements from 1991 to 2012 in both the upper (blue) and lower (red) troposphere over the Asian emission region spanning longitudes $123°E$ to $142°W$ and latitudes $31°N$ to $42°N$. This region corresponds to the Northern Hemisphere warm conveyor belt (Figure 1(e)), through which heavy emissions from the Asian continent enter the North Pacific atmosphere (also see transport of Asian dust storm over the North Pacific(Wang, 2007)). The measurements are depicted as the maximum (upper left), 75th percentile (upper middle), 50th percentile (upper right), 25th percentile (lower left), minimum (lower middle), and mean (lower right) of all in-situ measurements in a given month. Linear regression analysis of the measurements reveals increasing trends in CO concentrations in both the lower and upper troposphere over the Asian emission regions.



## 3.2 Seasonal Variability in the Upper and Lower Troposphere



**Figure 4.** Long-term measurements of CO (top panels), $O_3$ (second panels from the top), and $H_2O$ (second panels from the bottom) measurements during 2000-2018 over the western Pacific (longitudes $123°E - 142°W$, and latitudes $31°N - 42°N$). Long-term measurements of CO and $H_2O$ over western Europe (bottom panels; longitudes $1°E - 5°E$, and latitudes $46°N - 49°N$). The vertical panels shows the time-series measurements (left panels), monthly (middle panels), and scattered plot of two observed variables (right panels).

Figure 4 illustrates long-term measurements of CO (top panels), $O_3$ (second panels from the top), and $H_2O$ (second panels from the bottom) during the period 2000-2018 over the western Pacific (longitudes $123°E - 142°W$



and latitudes $31°N - 42°N$). Additionally, long-term measurements of CO and $H_2O$ over western Europe are de-
picted (bottom panels; longitudes $1°E - 5°E$ and latitudes $46°N - 49°N$). The vertical panels display the time-series
measurements (left panels), monthly averages (middle panels), and scatter plots of two observed variables (right
panels).

CO trends in the upper troposphere show positive values for the period 2001-2018, consistent with the positive
trends observed over the period 1991-2012 (see Figure 3).

Conversely, CO trends in the lower troposphere are close to zero but slightly negative over the period 2000-2018.
The CO trends over the period 1991-2012 were 0.6 ppbv yr$^{-1}$ (trends of the mean of the 50th percentile of CO, as
shown in Figure 3). The reductions in CO trends when comparing the 2000-2018 period with the 1991-2012 period
are consistent with the reduction in technological CO emissions over East Asia after 2012 (see Figure 1(f)).

Monthly variations of CO in the upper troposphere are high during summer months and low during winter months.
In contrast, monthly CO levels in the lower troposphere are low during summer and high during winter. The seasonal
variations of CO in the upper troposphere are out of phase with those in the lower troposphere. This out-of-phase
pattern is evident in the scatter plot, where CO levels in the upper troposphere are negatively correlated with
those in the lower troposphere. These results highlight the significance of vertical pumping processes during summer
months in transporting CO from the lower troposphere to the upper troposphere.

The evidence of vertical pumping process during the summer months can be seen when comparing the time-series
long-term measurements of CO and $O_3$ in the upper troposphere. The trends for CO are positive, while the trends
for $O_3$ are negatives. The monthly CO in the upper troposphere are higher in the summer months than in the winter
months. The monthly $O_3$ in the upper troposphere are higher in the spring months than in the summer months. The
negative $O_3$ indicate less downward intrusions of elevated stratospheric $O_3$ in the upper troposphere (ref), and more
pumping input of low $O_3$ from the lower troposphere to the upper troposphere. The $O_3$ are negatively correlated
with the CO in the upper troposphere.

Figure 4 futher compares CO and $H_2O$ in the upper troposphere. The trends for $H_2O$ are negative, while the
trends for CO are positive in the upper troposphere. The seasonal $H_2O$ are higher during the summer months than
during the winter months. The $H_2O$ are positively correlated with the CO in the upper troposphere, indicating the
same sources for the elevated CO and elevated $H_2O$ in the upper troposphere.

Figure 4 also compares CO and $H_2O$ in the upper troposphere over Western Europe ($1°E - 5°E$, $46°N - 49°N$). In
this region, both CO and $H_2O$ trends are negative. Monthly $H_2O$ levels are higher during summer months compared
to other months in the upper troposphere. Additionally, CO shows a negative correlation with $H_2O$ in the upper
troposphere.



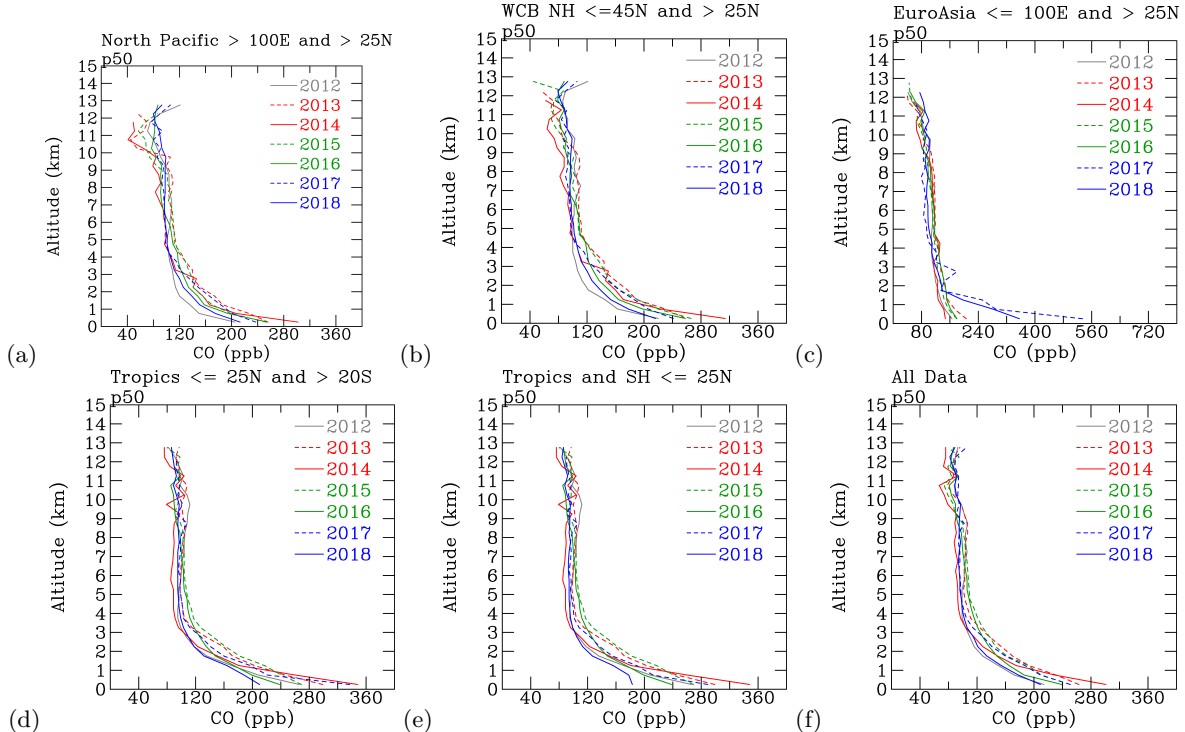

**Figure 5.** Annual profiles of in-situ CO measurements during the period 2012-2018. Profiles are shown for the 50th percentile of measurement data: (a) dwonstream of heavy Asian emission ares over north Pacific, in longitudes east of $100°E$ and latitudes north of $25°N$; (b) Northern Hemisphere warm conveyor belt, in latitudes $25°N - 45°N$; (c) upstream of Asian emission areas over Euro-Asia regions, in longitudes west of $100°E$ and latitudes north of $25°N$; (d) tropical regions, in latitudes $25°N - 20°S$; (e) tropical regions and Southern Hemisphere, in latitudes south of $25°N$; and (f) all data.

## 3.3 Variability of CO Profiles from 2012 to 2018

The previous section demonstrated evidence of the vertical pumping of CO and $H_2O$ from the lower troposphere to the upper troposphere, which plays a crucial role in the vertical redistribution of chemicals from ground-level sources throughout the troposphere.

In this section, we examine the vertical profiles of CO over various regions, as shown in Figure 1, using in-situ measurements collected by the IAGOS aircraft.

Figure 5 presents annual profiles of in-situ CO measurements from 2012 to 2018. The profiles represent the 50th percentile of measurement data in different regions: (a) downstream of heavy Asian emission areas over the North Pacific, in longitudes east of $100°E$ and latitudes north of $25°N$; (b) Northern Hemisphere warm conveyor belt, spanning latitudes $25°N - 45°N$; (c) upstream of Asian emission areas over Euro-Asia regions, in longitudes west





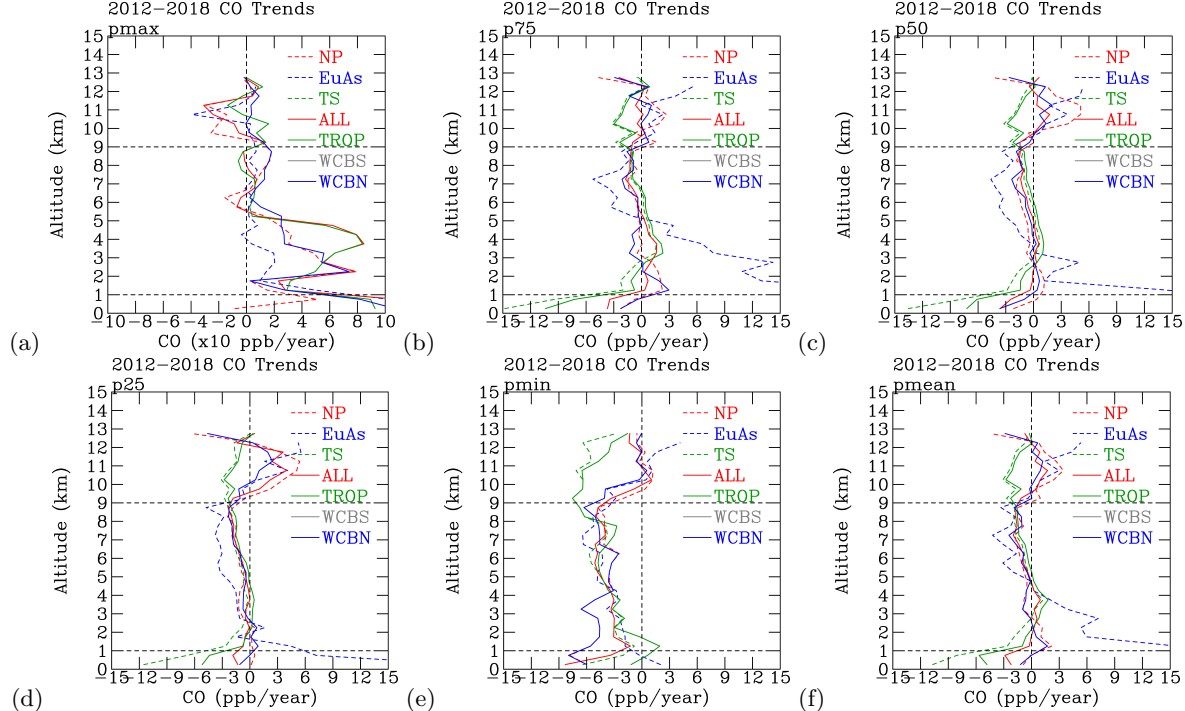

**Figure 6.** Profiles of CO trends for the period 2012-2018. The trends were shown for regions downwind of the east Asia emission areas over the North Pacific region (dashed red), upwind of the east Asia emission ares over the Euro-Asia region (dashed blue), tropical regions and the Southern Hemisphere (green), Southern Hemisphere warm conveyor belt (gray), Northern Hemisphere warm conveyor belt (blue), and regions (red). The trends are shown for the (a) maximum, (b) 75th percentile, (c) 50th percentile, (d) 25th percentile, (e) minimum, and (f) the mean of the in-situ measurement data.

of $100°E$ and latitudes north of $25°N$; (d) tropical regions, covering latitudes $25°N - 20°S$; (e) tropical regions and Southern Hemisphere, encompassing latitudes south of $25°N$; and (f) all data.

CO concentrations near the surface range between 160 ppbv and 560 ppbv, while concentrations in the upper troposphere vary between 40 ppbv and 120 ppbv. Significant variations in CO were observed at altitudes lower than 3 km and higher than 9 km. A sharp decrease in CO concentration was observed vertically, from near the surface to about 5 km, followed by a gradual reduction from 5 km to 12 km.

Figure 5 clearly illustrates that CO is produced from surface emissions and transported upward through vertical pumping processes.

It's worth noting that CO concentrations in the upper troposphere over the North Pacific (Figure 5(a)) and the warm conveyor belt (Figure 5(b)) exhibit more variability than in other regions during the 2012-2018 period. These measurements in the upper troposphere align with the reduction in surface emission sources of CO from Asian emission sources after 2012 (Figure 1(f)).



Figure 6 presents profiles of CO trends for the period 2012-2018 across various regions, showcasing different percentiles and means of in-situ measurement data:

Downwind of the East Asia emission areas over the North Pacific region (dashed red). Upwind of the East Asia
emission areas over the Euro-Asia region (dashed blue). Tropical regions and the Southern Hemisphere (green). Southern Hemisphere warm conveyor belt (gray). Northern Hemisphere warm conveyor belt (blue). All data (red). (a) Maximum, (b) 75th percentile, (c) 50th percentile, (d) 25th percentile, (e) minimum, and (f) mean trends are displayed.

Negative monthly mean CO trends are consistent with the mean and 50th percentile CO trends between 3 and 4
km altitudes, based on IAGOS observations during 2012-2018.

The airplane measurements indicate positive trends over East Asia and the North Pacific, while negative trends are observed over the North Atlantic, consistent with a steady decline in technological emissions estimated by EDGARv6.2.

Figure 6 shows profiles of CO tendencies averaged over the regions shown in Figure 5. Maximum observed CO
exhibits distinctive positive trends in the troposphere, particularly below 6 km altitudes. Conversely, minimum observed CO trends negatively below 10 km altitudes, representing background CO.

Positive trends suggest that chemical sources, including hydrocarbon emissions converted to CO and direct CO emissions, dominate the highest CO concentrations. Negative trends indicate that chemical sinks are the main factor controlling CO in the atmosphere, with CO generally declining without input from chemical sources.

Trends for the 75th percentile CO resemble those for the maximum CO, with the most significant positive trends occurring below 6 km altitudes. In contrast, trends for the 25th percentile CO are generally negative in the troposphere, similar to trends for the minimum CO, indicating dominance of chemical sinks below 9 km altitudes and chemical sources in the upper troposphere.

The 50th percentile CO trends mirror those for mean CO, with negative trends below 9 km altitudes except for
small positive trends between 3 and 5 km altitudes. Above 9 km altitudes, positive trends are observed.

Chemical sources contribute to positive tendencies below 5 km altitudes, while reduced chemical sinks lead to positive tendencies above 9 km altitudes. The minimum CO exhibits negative tendencies throughout most altitudes, indicating consistent removal of CO from the atmosphere.

These results highlight the importance of chemical sinks in controlling CO levels, especially in the upper tropo-
sphere where slow chemical sinks contribute to CO accumulation. Continuous emissions of CO and hydrocarbons sustain this accumulation, making the upper troposphere a sensitive region for monitoring trends in short-lived chemical species like CO. High-quality in-situ measurements from IAGOS Package 1 in the upper troposphere are invaluable for understanding these dynamics.





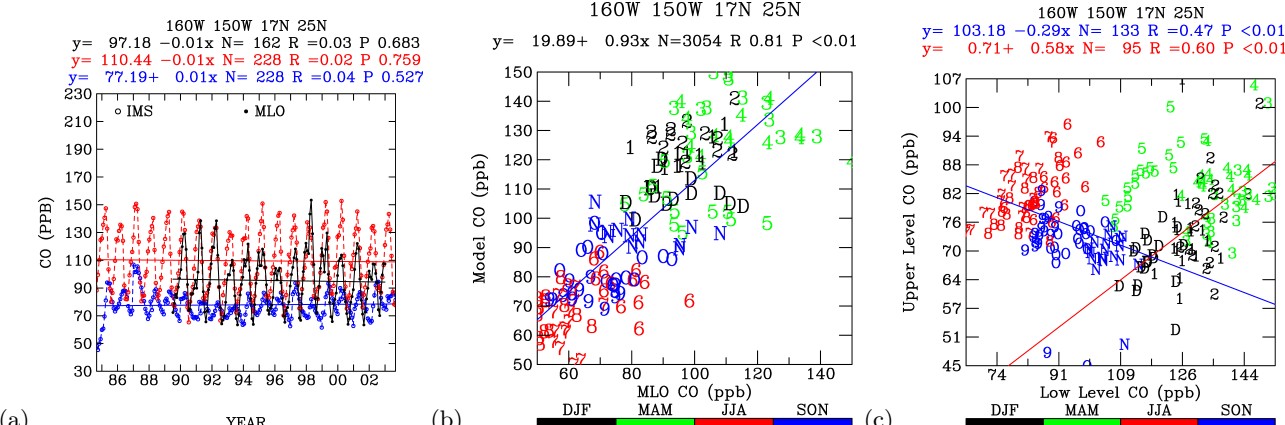

**Figure 7.** Time-series simulation results from the IMS model and the comparison with the Mauna Loa measurements. (a) Time-series CO from the IMS simulation, where red for the lower troposphere, and blue for the upper troposphere. The Mauna Loa measurements are shown in black. (b) Scattered plot analysis of the IMS simulated CO versus the Mauna Loa measurements of CO. The colors indicate the month of each data point. (c) Scattered plot analysis of the IMS simulated CO in the upper troposphere versus the IMS simulated CO in the lower troposphere.

## 3.4 What Controls Long-Term CO Trends in the Atmosphere

### 3.4.1 IMS simulations and verifications against MLO, IAGOS, MOPITT measurements

Figure 7 presents time-series simulation results from the IMS model, along with a comparison with Mauna Loa measurements. The IMS simulation results are depicted for both the lower troposphere (red) and the upper troposphere (blue) in Figure 7(a), while the Mauna Loa measurements are represented in black.

In Figure 7(b), a scatter plot analysis of the IMS simulated CO versus the Mauna Loa measurements of CO is shown. The comparison reveals a strong correlation between the IMS simulations and the Mauna Loa measurements, although the IMS model tends to overestimate CO compared to the Mauna Loa measurements.

Figure 7(c) illustrates a scatter plot analysis of the IMS simulated CO in the upper troposphere versus the IMS simulated CO in the lower troposphere, with colors indicating the months of the data.

During the summer to autumn months, modeled CO concentrations in the upper troposphere are negatively correlated with those in the lower troposphere. These months coincide with the period of intensive Asian summer monsoon over the North Pacific.

Conversely, during the winter to spring months, modeled CO concentrations in the upper troposphere show a positive correlation with those in the lower troposphere. These months correspond to the period of intensive Asian winter monsoon, characterized by frontal activity associated with the development of baroclinic waves.

Figure 8 displays time-series simulation results from the IMS model (blue), comparisons with the IAGOS measurements (sky blue), and linear regression predictions from the IMS 1984-2003 simulation extended to the end of 2020





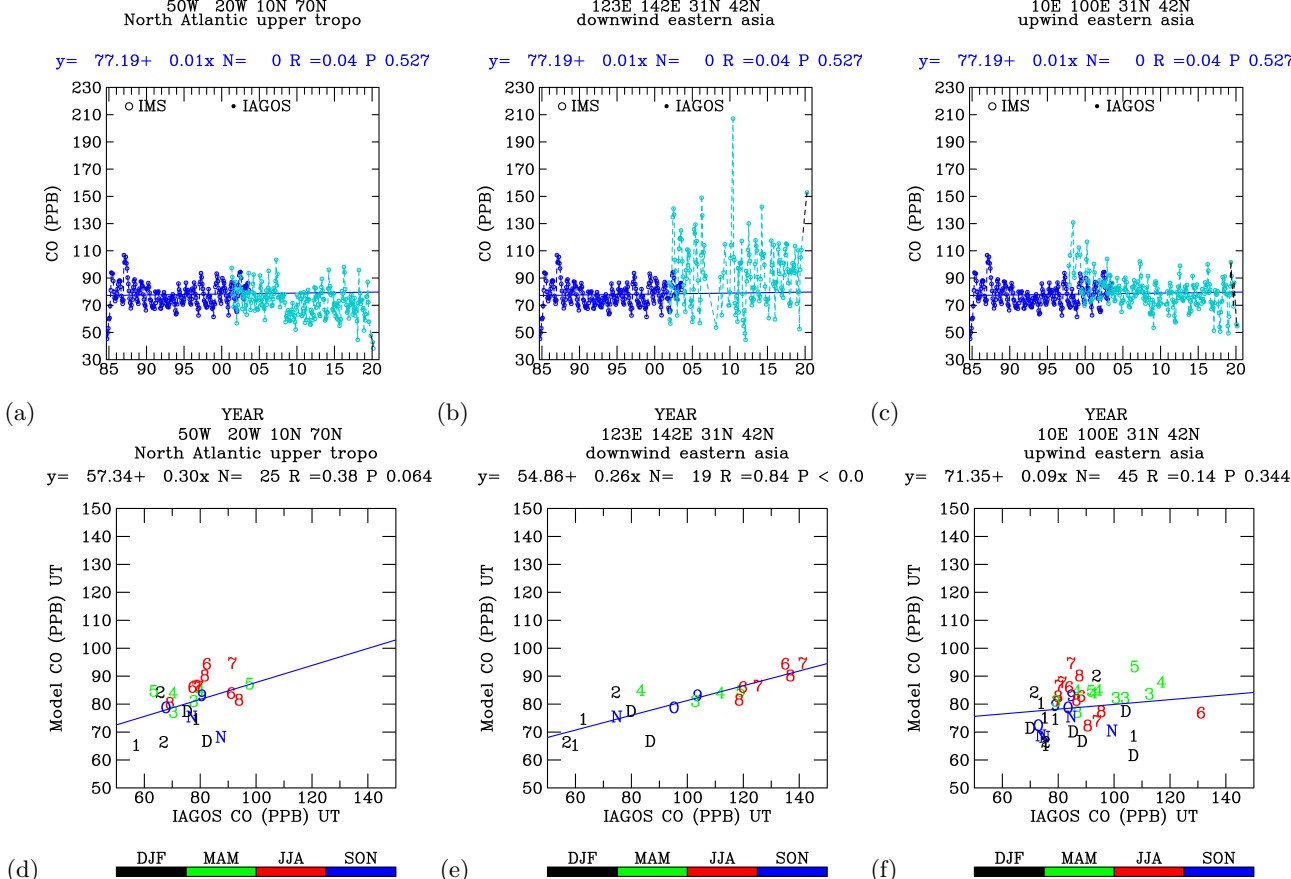

**Figure 8.** Time-series simulation results from the IMS model (blue), the comparisons with the IAGOS measurements (sky blue), and the linear regression prediction from the IMS 1984-2003 simulation to the end of 2020 over the upper troposphere of (a) the North Atlantic($50°W$-$20°W$, $10°N - 70°N$); and the scattered plot analysis in (d); (b) immediately downwind of the East Asia emission areas ($123°E$-$142°E$, $31°N - 42°N$), and the scattered plot analysis in (e); (c) upwind of the East Asia emission areas ($10°E$-$100°E$, $31°N - 42°N$), and the scattered plot analysis in (f).

over the upper troposphere of: (a) the North Atlantic ($50°W$-$20°W$, $10°N$-$70°N$), along with scattered plot analysis in (c); (b) immediately downwind of the East Asia emission areas ($123°E$-$142°E$, $31°N$-$42°N$), with scattered plot analysis in (e); (c) upwind of the East Asia emission areas ($10°E$-$100°E$, $31°N$-$42°N$), along with scattered plot analysis in (f).

The scatter plots demonstrate that while the IMS simulations correlate well with the IAGOS measurements, the model underestimates CO compared to the IAGOS in-situ measurements.

Figure 8 also showcases linear regression predictions of the 2004-2020 CO concentrations based on the IMS 1984-2003 simulations, alongside comparisons with the IAGOS 2004-2020 measurements.



These comparisons yield significant findings. Notably, the IMS 1984-2003 simulations were conducted with constant technological CO and emissions from hydrocarbons, while the IAGOS measurements reflect the real CO and hydrocarbon emissions during 2004-2021. Variations in technological CO emissions are estimated in Figure 1(f).

Over the North Atlantic upper troposphere (Figure 8(a)), IMS predictions exceed the IAGOS measurements. This suggests a significant reduction in CO chemical sources, including CO and hydrocarbon emissions, and an
increase in upper troposphere/lower stratosphere (UTLS) air exchange processes transporting low CO from the lower stratosphere to the upper troposphere.

Over the North Pacific upper troposphere and downwind of the East Asia emission areas (Figure 8(b)), IMS 2004-2020 predictions fall below the IAGOS 2004-2020 measurements. This indicates that the impacts of continuous CO emissions during 2004-2020 outweigh the effects of the constant CO emissions used in the 1984-2003 simulations.
Over the upper troposphere upwind of the East Asia emission areas (Figure 8(c)), IMS 2004-2020 predictions approximate the 2004-2020 in-situ measurements. This suggests reductions in chemical sources and emissions of CO and hydrocarbons upwind of the East Asia emission areas.

Figure 9 illustrates time-series simulation results from the IMS model alongside comparisons with MOPITT measurements. The IMS simulations depict the upper troposphere (in blue) and the lower troposphere (in red) for
(a) the North Atlantic ($50°W − 20°W, 10°N − 70°N$; downwind of North American emission sources), and (e) the North Pacific ($150°E − 140°W, 25°N − 45°N$; downwind of Asian emission sources). MOPITT measurements are represented for the lower troposphere (in light green) and the upper troposphere (in sky blue).

Comparison between the IMS model and MOPITT measurements is presented for the lower troposphere over (b) the Atlantic and (f) the Pacific, and for the upper troposphere over (c) the Atlantic and (g) the Pacific. Additionally,
the correlation between model CO in the upper troposphere and CO in the lower troposphere is depicted over the (d) Atlantic and (h) Pacific. The color coding and numerical annotations indicate the month of the data.

The IMS model exhibits strong correlation with MOPITT measurements across both the lower and upper troposphere in the North Atlantic and the North Pacific. Notably, MOPITT estimates tend to be higher than IMS results over the North Pacific (Figure 9(b)) and the North Atlantic (Figure 9(f)). Conversely, for the upper troposphere,
MOPITT estimates appear lower than IMS results over the North Pacific (Figure 9(c)) and the North Atlantic (Figure 9(g)).

During summer months, IMS CO levels in the upper troposphere show negative correlation with IMS CO in the lower troposphere over the North Pacific (Figure 9(d)) and the North Atlantic (Figure 9(h)). These findings align with seasonal trends observed in IAGOS in-situ measurements over the North Pacific (Figure 4).
The IMS model and MOPITT measurements stand out as the only sources capable of providing CO measurements on a global scale. The analysis confirms the consistency between IMS model outputs and MOPITT measurements, reinforcing the observation of CO transport from the lower to the upper troposphere during summer months in the Northern Hemisphere, as evidenced by IAGOS in-situ measurements over the North Pacific. Further validation of MOPITT measurements with Mauna Loa in-situ data is planned for the next phase.





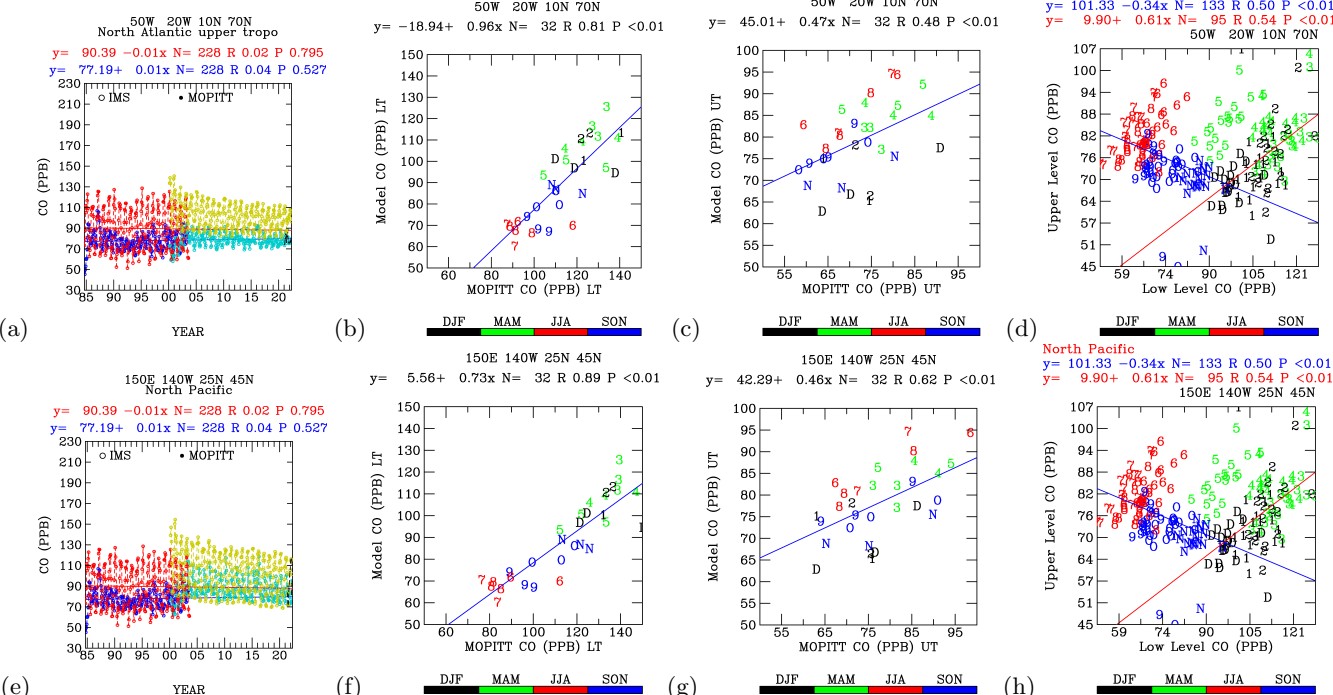

**Figure 9.** Time-series simulation results from the IMS model and the comparisons with the MOPITT measurements. The IMS simulations are shown for the upper troposphere (blue) and the lower troposphere (red) for (a) the North Atlantic ($50°W - 20°W, 10°N - 70°N$; downwind of the North America emission sources), and (e) the North Pacific ($150°E - 140°W, 25°N - 45°N$; downwind of the Asia emission sources). The MOPITT measurements are shown for the lower troposphere (light green) and the upper troposphere (sky blue). The IMS model compared with the MOPITT measurements for the lower troposphere over (b) the Atlantic and (f) the Pacific; and for the upper over (c) the Atlantic and (g) the Pacific. The model CO in the upper troposphere correlated with the CO in the lower troposphere over the (d) Atlantic and (h) the Pacific. The color and digits indicate the month of the data.

Figure 10 illustrates time-series CO measurements from MOPITT and their comparison with Mauna Loa measurements: (a) The MOPITT CO measurements during the period 2000-2022 over the upper (blue) and the lower (red) troposphere in an area containing Mauna Loa ($160°W$-$150°W$, $17°N$-$25°N$). The Mauna Loa measurements are shown for the period 1989-2022 (black). Also depicted are the linear regressions of the MOPITT data over the lower and the upper troposphere, respectively, and the Mauna Loa data. (b) The MOPITT data over the lower troposphere correlated with the Mauna Loa measurements. The colors and the digits indicate the months of the data.

The MOPITT measurements correlate well with the Mauna Loa measurements (Figure 10(b)). However, the MOPITT overestimates CO during the summer to autumn months when the CO concentrations are lower than 100



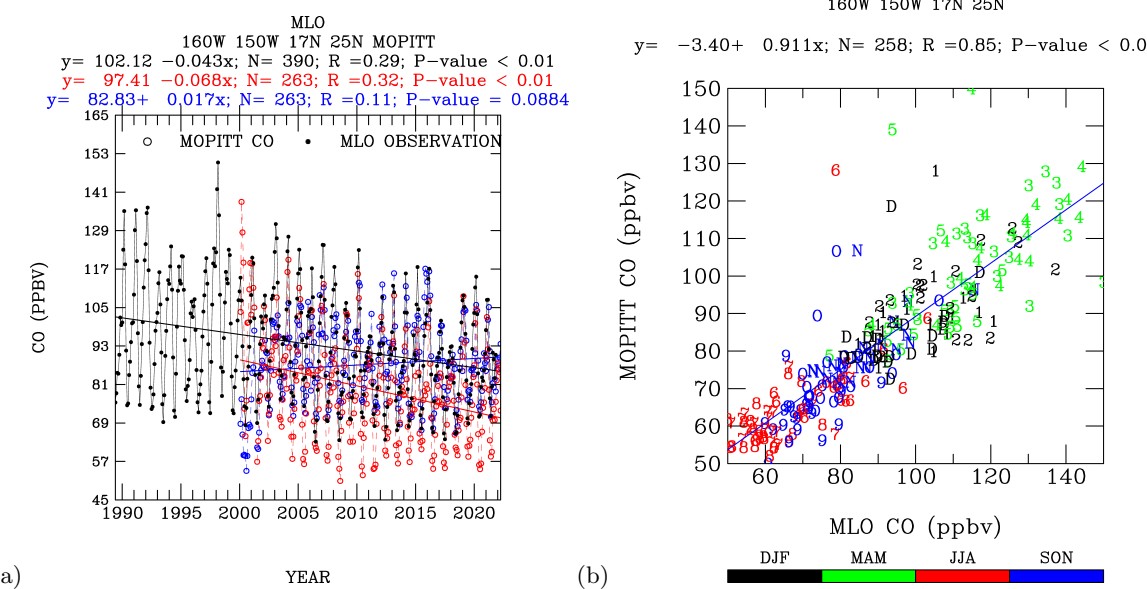

**Figure 10.** Time-series CO measurements from the MOPITT, and the comparisons with the Mauna Loa measurements. (a) The MOPITT CO measurements during the period 2000-2022 over the upper (blue) and the lower (red) troposphere in an area contains the Mauna Loa ($160°W - 150°W, 17°N - 25°N$). The Mauna Loa measurements are shown for the period 1989-2022 (black). Also shown are the linear regressions of the MOPITT data over the lower and the upper troposphere, respectively; and the Mauna Loa data. (b) The MOPITT data over the lower troposphere correlated with the Mauna Loa measurements. The colors and the digits indicate the months of the data.

ppbv, while underestimating CO during the winter to spring months when the CO concentrations are higher than
100 ppbv compared with the Mauna Loa measurements.

The linear regression exhibits negative trends in the CO from the Mauna Loa measurements during 1989-2022, which is consistent with the negative trends in the CO from the MOPITT measurements during 2000-2022 (Figure 10(a)). Conversely, the MOPITT measurements show positive linear regression over the Pacific upper troposphere, consistent with the IAGOS measurements (Figure 2).

Figure 11 compares MOPITT with IAGOS observations arranged following westerly flow patterns, starting from: (a) Downwind of the North America emission sources and over the North Atlantic ($50°W$-$20°W$, $10°N$-$70°N$). (b) Western Europe flight regions ($1°E$-$15°E$, $45°N$-$55°N$). (c) Paris region ($1°E$-$10°E$, $43°N$-$53°N$). (d) India ($73°E$-$87°E$, $10°N$-$23°N$). (e) China ($105°E$-$125°E$, $22°N$-$45°N$). (f) Downwind of the East Asia emission areas and over the North Pacific ($123°E$-$142°E$, $31°N$-$42°N$).

Over the lower troposphere, MOPITT data correlated well but underestimated CO concentrations compared with the IAGOS measurements. Similarly, over the upper troposphere, MOPITT measurements also correlated well but underestimated CO concentrations compared with the IAGOS in-situ measurements.





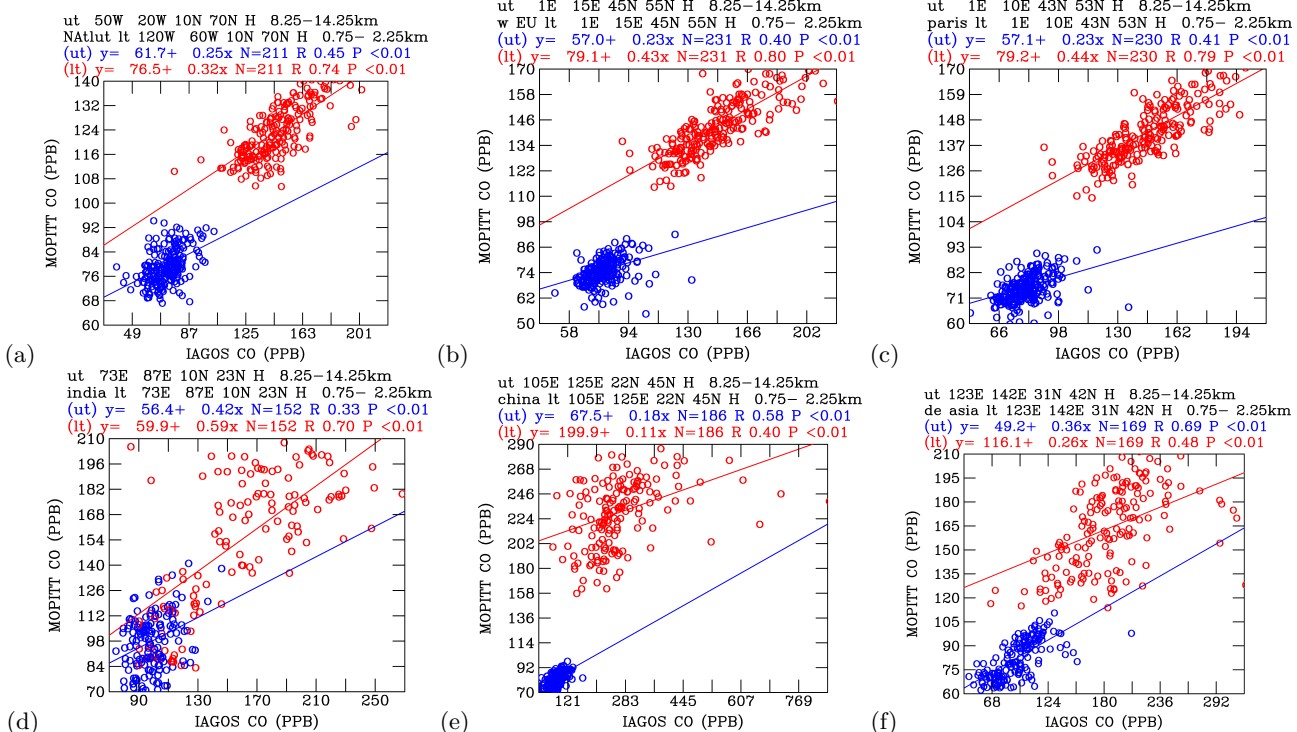

**Figure 11.** Comparisons of the MOPITT with the IAGOS observations. The panels are arranged following the westerly flow patterns, started from (a) downwind of the North America emission sources and over the North Atlantic ($50°W - 20°W$, $10°N - 70°N$), (b) western Europe flight regions ($1°E - 15°E$, $45°N - 55°N$), (c) Paris region ($1°E - 10°E$, $43°N - 53°N$), (d) India ($73°E - 87°E$, $10°N - 23°N$), (e) China ($105°E - 125°E$, $22°N - 45°N$), and (f) downwind of the east Asia emission areas and over the North Pacific ($123°E - 142°E$, $31°N - 42°N$).

Hence, while the MOPITT data are well correlated with the IAGOS measurements, they tend to underestimate the CO concentrations compared with the IAGOS in-situ measurements.

Figure 12 presents time-series CO concentrations from the IMS 1984-2003 simulations and linear regression predictions for the period 2004-2022, compared with MOPITT and IAGOS observations in various regions of the upper troposphere.

The purpose is to verify the impact of continuous growth of CO emissions during the 2004-2022 period in the upper troposphere. The controlled and predicted CO trends from the constant emissions of CO and hydrocarbons are based on the IMS 1984-2003 simulations, which used constant emissions for each year from 2004 to 2022. The observed CO trends are derived from IAGOS in-situ measurements and MOPITT remote sensing measurements during the period 2004-2022, reflecting the impacts of real emissions of CO and hydrocarbons.

By comparing the observed CO trends with the controlled CO trends, we can identify the impacts of increased emissions of CO and hydrocarbons in the upper troposphere.





**Figure 12.** Time-series CO concentrations from IMS 1984-2003 simulations, and liner gression prediction to the period 2004-2022 compared with the MOPITT and the IAGOS observations in the upper troposphere. The panels are arranged following the westerly flow patterns, started from over (a) the North America ($120°W - 80°W$, $22°N - 45°N$), (b) North Atlantic ($50°W - 20°W$, $10°N - 70°N$), (c) Paris region ($1°E - 10°E$, $43°N - 53°N$), (d) upwind of eastern Asia ($10°E - 100°E$, $31°N - 42°N$), (e) India ($73°E - 87°E$, $10°N - 23°N$), (f) China ($105°E - 125°E$, $22°N - 45°N$), (g) eastern Asian emission ($123°E - 142°E$, $10°N - 70°N$), (h) North Pacific ($150°E - 140°W$, $25°N - 45°N$), and (i) Mauna Loa ($160°W - 150°W$, $16°N - 25°N$).

The comparisons are arranged following westerly flow patterns, starting from: (a) North America ($120°W$-$80°W$, $22°N$-$45°N$), (b) North Atlantic ($50°W$-$20°W$, $10°N$-$70°N$), (c) Paris region ($1°E$-$10°E$, $43°N$-$53°N$), (d) Upwind of eastern Asia ($10°E$-$100°E$, $31°N$-$42°N$), (e) India ($73°E$-$87°E$, $10°N$-$23°N$), (f) China ($105°E$-$125°E$, $22°N$-





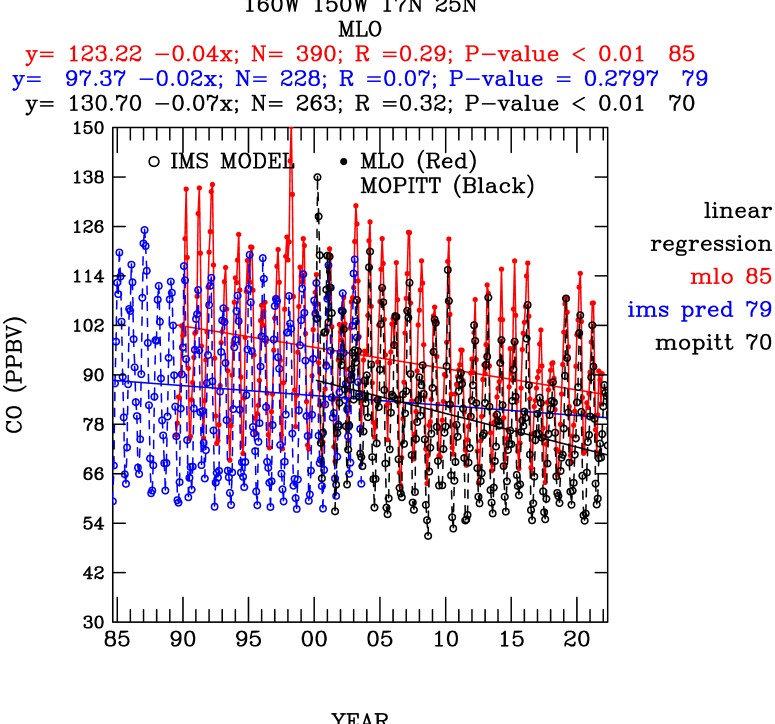

**Figure 13.** Time-series CO concentrations from IMS simulations (blue) and linear regression prediction compared with the MOPITT (black) and Mauna Loa observations (red).

45°N), (g) Eastern Asian emission (123°E-142°E, 10°N-70°N), (h) North Pacific (150°E-140°W, 25°N-45°N), and (i) Mauna Loa (160°W-150°W, 16°N-25°N).

The comparisons reveal that the IMS predicts CO trends in 2022 that are higher than the observed CO trends from IAGOS in-situ measurements and MOPITT remote sensing measurements over the upper troposphere of North America, the North Atlantic, Paris region, and upwind of the East Asia heavy emission areas.

    For the remaining regions, IMS predictions are lower than the observations. IMS predictions are lower than both IAGOS and MOPITT measurements over India and the North Pacific. Over China and downwind of East Asia,
IMS predictions are lower than IAGOS measurements and MOPITT observations are lower than IMS predictions over Mauna Loa. It's worth noting that there are few observations from IAGOS over the Mauna Loa area, and most available observations are actually higher than the trends predicted by the IMS model.

    Hence, these results indicate that for regions upwind of the East Asia heavy emission area, the upper troposphere CO trends have decreased. Conversely, for the upper troposphere downwind of the East Asia heavy emission area,
CO trends have increased during the period 2004-2022.

    Figure 13 displays time-series CO concentrations from the IMS simulations (blue) and the linear regression prediction compared with MOPITT (black) and Mauna Loa observations (red). The negative CO trends from the



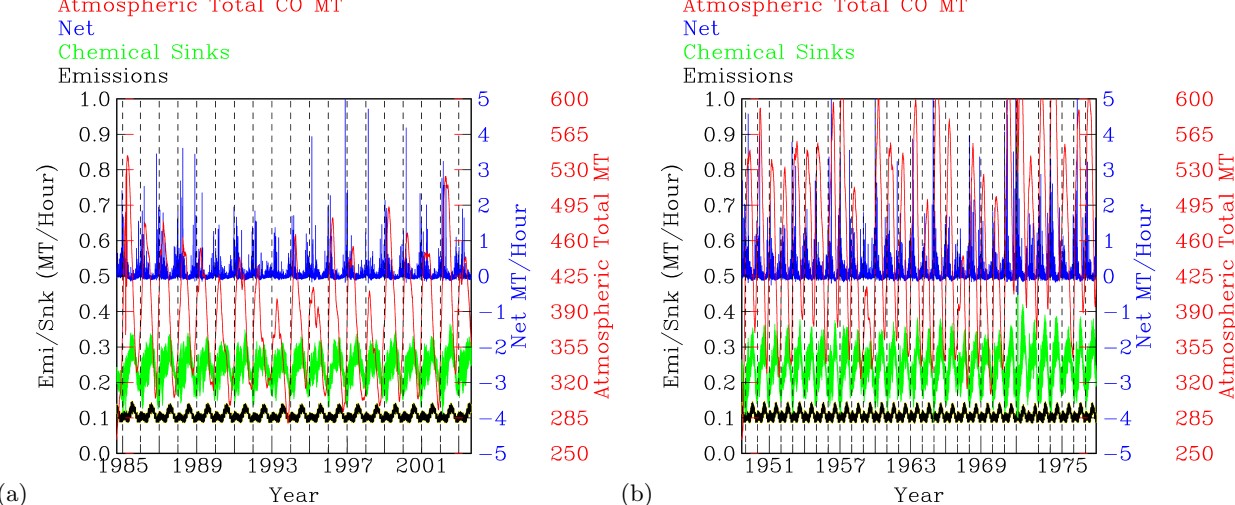

**Figure 14.** Time-series chemistry budget for CO concentrations from the IMS simulations. (a) The IMS 1984-2003 simulations. (b) The IMS 1948-1978 simulations.

IMS 1984-2003 simulations are consistent with the negative linear regression trends from the Mauna Loa in-situ measurements and the MOPITT remote sensing measurements.

The CO concentration in 2022 from the IMS linear regression prediction (79 ppbv) is lower than the CO concentration from the linear regression prediction of the Mauna Loa 1989-2021 observations (85 ppbv) and higher than the CO concentration from the linear regression prediction of the MOPITT measurements (70 ppbv). The higher CO from Mauna Loa compared to the IMS predictions indicates that the actual annual CO and hydrocarbon emissions during 2004-2022 are higher than the constant emissions of CO and hydrocarbons used in the IMS 1984-2003

simulations.

The lower MOPITT prediction of CO compared to Mauna Loa observations is consistent with previous analyses (Figure 11(f)), which show that MOPITT measurements underestimate CO compared to IAGOS in-situ measurements in regions downwind of East Asia emission areas.

What controls long-term CO trends in the atmosphere? We can understand the processes governing CO dynamics

through model simulations.

The CO budget, calculated from a three-dimensional global tropospheric chemistry transport model IMS, indicates constant CO mixing ratios during the 20-year period of 1984-2003 (Figure 14(a)), and nearly constant CO mixing ratios during the 30-year period of 1948-1978 (Figure 14(b)).

In these simulations, the total annual direct CO emissions are approximately 1100 MT per year. These emissions

consist of technological processes (set at constant emissions of 400 MT/yr), biomass burning emissions, and natural processes from the ocean and vegetation. Notably, EDGARv6.1 estimates total technological CO emissions at about 400 MT per year.





The IMS model calculates approximately 2200 MT of CO produced chemically (including direct CO emissions) per year, and approximately 2200 MT of CO removed chemically from the atmosphere per year. This indicates that the chemical sinks (2200 MT/yr) are roughly twice the CO emissions from direct sources (1100 MT/yr). The rates of chemical production (including direct CO emissions) of CO are balanced by the rates of chemical removal per year.

The total amount of CO in the atmosphere varies between 200 and 500 MT. According to EDGARv6.2 estimates, the total CO in the atmosphere is of the same order as CO directly emitted from technological processes.

The total amounts of CO in the atmosphere fluctuate between 200 and 500 MT, depending on seasonal variations that drive chemical reactions converting CO to other products and converting other hydrocarbons to CO. Consequently, CO mixing ratios are predominantly influenced by chemical processes.

The net chemical processes generally yield positive results from late autumn through winter to early spring months. Conversely, they tend to be negative from late spring through summer to early fall months. Consequently, the annual net chemical processes are close to zero, resulting in near-zero CO trends during the 20-year and 30-year model simulations, respectively.

The satellite remotely sensed CO trends during the period from March 2000 to May 2021 exhibit a range of behaviors, including zeros to negative trends (as observed over the North Atlantic, Figure 11(a)), and positive trends (as observed over India, Figure 11(d)).

The global mean trends derived from MOPITT observations are predominantly negative, indicating that, on a global scale, chemical sinks for CO exert a dominant influence on long-term mixing ratios in the atmosphere.

Positive trends for CO are observed over areas such as India and downwind of East Asia emission sources over the North Pacific (Figure 11(f)), suggesting the influence of chemical sources in these regions.

Overall, globally, CO mixing ratios are decreasing. Positive trends in CO are observed only in regions where chemical sources, arising from direct emissions of CO and hydrocarbons, outweigh chemical sinks, resulting in an increase in CO levels.

Therefore, MOPITT provides valuable insights into CO trends on a global scale.

The monthly mean observations from Mauna Loa exhibit negative trends from 1989 to 2021 (Figure 13). While emission trends over Asian countries were positive during the 1978-2011 period, they turned negative during 2012-2018 (Figure 1). Consequently, the negative trends in monthly mean CO over Mauna Loa suggest that chemical sinks dominate over chemical sources.

## 4   Conclusions

In this study, we demonstrate that IAGOS provides high-quality in-situ data suitable for analyzing trends in short-lived species like CO in the upper troposphere. Our IMS model simulations illustrate that chemical sinks continually



remove CO from the atmosphere unless countered by sources such as direct emissions of CO and hydrocarbons
converted to CO.

By integrating IAGOS profile measurements with IMS chemical analysis, we discern that chemical sources predominate below 2 km and above 9 km altitudes, while chemical sinks outweigh sources between 5 km and 10 km altitudes.

Aircraft in-situ measurements reveal that the North Pacific upper troposphere experienced increased pollution over a 28-year period from 1991 to 2018. Elevated CO mixing ratios were more pronounced in summer than in winter, suggesting vertical transport of CO from surface sources. Model calculations corroborate the transport of surface CO from Asian sources as the primary contributor to increased pollution in the upper troposphere. Furthermore, these calculations support the notion that the rise in CO over the North Pacific upper troposphere is attributable

to increased ground-level CO emissions.

The main findings of this study include: Illustrating vertical profiles of chemical sources and sinks near anthropogenic source regions using in-situ measurements. Presenting profiles of chemical sources compared to chemical sinks, utilizing data spanning from 2012 to 2018. Highlighting increased CO trends in the upper troposphere based on data from PEM-WEST A, PEM-WEST B, TRACE-P, HIPPO, IAGOS, and ATom. Confirming through model

calculations that chemistry plays a significant role in shaping CO profiles and trends. Demonstrating the high sensitivity of CO trends and profiles to variations in chemical sources and sinks, as evidenced by the analysis of minimum, 25th percentile, 50th percentile, 75th percentile, maximum, and mean CO data.



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
