# Peer review of "Air Pollution in The Upper Troposphere: Insights from In-Situ Airplane Measurements (1991-2018)"

_EGUsphere, 2024_

## Referee Comment (RC1)

**Referee report** (ACPD) for manuscript by Kuo-Ying Wang et al., Air Pollution in The Upper Troposphere: Insights from In-Situ Airplane Measurements (1991–2018)

**General Comments**

This manuscript shows analyses focusing on a subset of upper tropospheric CO in situ aircraft data from various sources (commercial plane flights as well as dedicated non-commercial missions) over the past three decades. The analyses are supplemented by satellite CO data (from MOPITT), which are mostly sensitive to the lower troposphere, but also have an upper tropospheric retrieval product, and ground-based (high altitude) in situ data from Mauna Loa, Hawaii. Other related data discussions include in situ upper tropospheric aircraft data for $O_3$ and $H_2O$. A 3-D model of the troposphere (the Integrated Modelling System, IMS) is used for comparisons and inferences about relevant processes affecting lower and upper tropospheric CO.

This work's goals are worthwhile:  to use various data sets (mentioned above) and model results to draw conclusions about upper (and lower) tropospheric CO behavior in certain regions of the globe, with some inferences about processes controlling the tropospheric CO distribution (some of which is already known). There is a good amount of work in this fairly long manuscript, in terms of comparisons. MOPITT data are also used and this helps to validate some of the other data analyses and results. The model is useful for some explanations (or potential explanations) of observed CO variations and inferences regarding "trends" - or more qualitatively, some inferred tendencies.

However, I consider that this work needs some fairly major revisions before it can be considered ready for publication, as explained below in more detail. I do recommend further work towards publication, as I believe that this type of analyses fits within the scientific goals of ACP.

This work requires more clarity overall, including a better discussion of uncertainties (or why they might be difficult to establish, at the very least, if this is somehow the thinking), as well as a better summary of the main results in the Abstract, main text and conclusions. Improving the Figures should help in a very significant way, although the accompanying text also needs clarifications. If some of the results, especially those having to do with trends, cannot be supported in a robust (mathematical) way, I recommend removing the stated conclusions, or at least indicating that more work (future efforts) would be needed to better establish or confirm what looks like tentative results, for now.

I do hope that these comments, with more specifics below, will be viewed constructively by the authors, towards a revised manuscript, and I trust that at least some of these results can be displayed better and explained more clearly.

**Specific Comments**

Despite the valuable goals of this work, I summarize here a number of difficulties and issues:

1.  The data are somewhat sparse in time or in space, at least for several of the in situ datasets other than IAGOS; this makes it more difficult to derive conclusions or especially trends in a quantitative way.

2.  Error bar estimates are unfortunately absent from this work, overall, while a significant set of correlations are poorly described or not very significant, statistically. Scientific results without some sort of error bar estimates (formal regression errors and maybe some discussion of sensitivity to various assumptions, e.g., data sparseness issues in early years) are almost meaningless.  As a related comment, only the simplest form of linear regression is used, and no attempt is made to remove an annual cycle, for example, which could actually reduce the error bar estimates (if they were to be calculated); also, some p values are shown, but with essentially no discussion.

3.  Importantly, the quality and visibility of many of the Figures needs to be improved, with less white space so that the reader can see the curves or points for different data sets and their comparisons; this can be done by reducing text around some panels and placing it in the plot panels and/or in the captions, and by producing rectangular panels rather than square panels, to fill in the page more efficiently, for better visibility. If needed, as an option, a single panel, or double panel Figure could be made or added in some cases to better illustrate a significant result, although hopefully that is not needed after revisions of several Figures.   The poor quality of several Figures makes it difficult for a reader to follow (or believe) some of the stated results, even if many results are, presumably,  correct to first-order.

4.  The Abstract should include the main results in some way; it is too vague as it stands.

5.  The Conclusion section also needs to more clearly specify the main results, with specific references to which Figure(s) the conclusions are based on and more explanations and context, considering that many of the readers will not try to read every line of this (revised) article.

6.  More reference to past work on CO trends should be made, in the Introduction and at least in the  Conclusions, to try to put this work, and its results, in the context of related past publications (in particular, regarding past trend or variability results for tropospheric CO). This holds even if "exact comparisons" are often difficult to make, given past results over somewhat different regions, altitudes, and/or time periods.

7.  Finally, there are a number of necessary minor technical comments at the end of my report; this underscores that not much proof-reading was carried out by the author and co-authors before the submission of this manuscript, which also makes a reviewer's careful work somewhat more tedious.

**More Specifics**

I provide detailed comments here, following the order of the Figures and related text/results, especially regarding trends and their robustness, given the limited discussion of these issues. I offer some suggestions for improvements, as well as regarding Figure legibility issues.

**Figure 2:**

This is where we start to see simple linear regression fits but there is no discussion about error bars or the significance of the results, or the impact on the (potential) conclusions. One issue is that the single point before 1992 does not exemplify the likely scatter that would exist if there was more data available in that time period (or from 1991 to 2008). Giving formal (2 sigma) error bars from the fits is probably a sufficient response to this question. It also depends what scientific conclusions one is trying to make, and how one tries to convey the importance of any results based on such a series of plots. Because of the data sparseness, my top-level impression from the p50 panels is that it will be difficult to state categorically that there has been an increase between 1991 and 2015–2018. The same probably holds for the implied decreases from the pmin panels. As another illustration of my concerns, if one did not have the post 2012 data yet for panels p50 and p25 45N-60N, one might well try to conclude that there were decreases between 1991 and 2011, whereas the slopes drawn after the fairly sudden increases post-2012 could be interpreted as a slow, gradual increase from 1991 to 2018. Also, if one did not have the data point(s) from 1991, one would conclude that the increases (or decreases) are much steeper than shown currently for the 2008-2018 period. My point here is a somewhat qualitative one, but the lack of more continuous data can lead to uncertainties in, or over-interpretation of inferred "trends".

Thus, if many trend results are not convincing (mathematically), I would tone down the statements that are made on lines 156 to 166, in particular. This is even more true for Fig. 6 showing trends. The general comments about larger values in a certain region versus another can probably be supported based on standard errors in the means, but even these statements are worth checking for significance, if this is viewed as important enough. I am also ignoring the accuracy estimates for the data points shown here, assuming that this plays a smaller role than other issues. However, the introductory remarks about the data sets, especially for the in situ CO aircraft data, should state something about the error bars in the data points that are displayed in various plots; readers should not be expected to know this information, even if you have (broadly) pointed to some relevant past publications.

Lines 165/166: "The 25th, 50th,… CO levels are impacted by emissions, with latitudes south of 45N being more affected by Asian emissions …" What leads to this conclusion, specifically? Is this based on scatter, large values, or what exactly? Which time period does this refer to? Please be more specific.

*Figure 2 technical improvements:* The text font size is rather small, so stretching this Figure to occupy as much space as possible across the page would be good. The caption should indicate what the various symbols mean, i.e. which data set do they correspond to. While this can be inferred from the years shown, a brief sentence or two would be worthwhile in my opinion (most readers are not as close to the various data sets as you are). Captions are there to ensure that symbols and curves shown in each Figure have been described sufficiently well. Also, the full range of years should be mentioned in the caption, to further assist the reader.

**Figure 3:**
Again, although this one has fewer panels than Figure 2, using most of the page width in a final manuscript would be beneficial (and this may well be the Journal's responsibility, I understand). Also, it seems that the y-intercept value is given by the first number in the linear fit equations listed above each panel; however, for the upper tropospheric blue result in pmean, a y-intercept value of 85.4 ppbv does not seem to match the plotted dashed line at year 1991. Similar comment regarding the p50 panel, where 62.3 ppbv does not seem to match the plotted y-value for the blue dashed line in 1991. Were some of the fits performed with a different approach (about a year within 2000-2008 maybe) rather than for an expected y-intercept in 1991? Please clarify. Also, I do not understand why some of the dashed lines show as partly red and partly blue, when one expects two lines, one red and one blue, for the different altitude regimes. In terms of which measurements are used for this Figure, this should be specified (even briefly) in the caption as well (e.g. lower tropospheric data are from …, etc…).

**Figure 4:**
Please explain why there are different numbers of points (and fitted equation details) for the UT CO in the 2$^{nd}$ from top left panel versus the top left panel, whereas the descriptions of the regions used appear to be identical; N is 171 versus 254, but the CO data points appear to match in these panels. I also do not completely understand the $O_3$ fitted line in the 2$^{nd}$ panel from the top (left panel), as its line intercept is -22.6; while this may be fine, it seems that the y-intercept approach (equation) is different in some cases. One would also expect a few comments about R values (significance, or why shown if not useful?) and p values. While it is stated that the CO trends show some increases, the fitted lines seem to give a very small slope, with a zero trend very likely included in the uncertainties (2-sigma error values are really needed to help one understand this more quantitatively, but this is my guess). One cannot robustly determine the validity of the statements on lines 183 to 188 without more quantitative information about fitted trends and* uncertainties (error bars). I expect that many statements are not supported by the error bars, so the authors should probably modify the strength of such conclusions, since (unfortunately) I would expect only marginal significance in many/most of the conclusions regarding trends. One way to reduce the uncertainties in the trends would be to fit an annual cycle (as a sum of sine and cosine functions with one year period, multiplied by constant terms to be fitted) and a linear trend term to datasets with enough points to do so; the explanation of such variability will help to reduce the underlying trend's uncertainty (error bar). This should produce more robust results, even if it may still be difficult to state that a zero trend is highly unlikely in some cases (especially those with fewer data points or shorter series). As stated in the manuscript, some (or most) CO tendencies are "close to zero" indeed. Trend values and error bars could be useful for $O_3$ and $H_2O$, if there is enough significance (at least beyond one sigma, say), with ppbv or ppmv per year, or %/yr units - of potential interest to the tropospheric community (and for future comparisons to other studies, for example).

In terms of Figure 4 improvements, trying to stretch the panels to be more rectangular (longer on the x-axis than on the y-axis) would be useful as well.

The possibility that "vertical pumping processes" can explain the out-of-phase relationship between upper tropospheric and lower tropospheric CO is interesting; this might be worth checking against the results of the IMS model, if possible. Does this suggest a several month

timescale for such a process? A reference to the paper by Schoeberl et al. (Geophys. Res. Lett., 2006, doi:10.1029/2006GL026178) on the CO tape recorder at tropical latitudes would probably be appropriate as well, although tropical upward transport and convection processes happen on a faster timescale than for the regions this manuscript focuses on (extra-tropics for the most part). More comments about the model comparisons, in any case, would be useful as well, if possible; is the model capable of matching some of the observational inferences, including correlations/anti-correlations between CO, $O_3$ $H_2O$, especially those with enough significance?

**Figure 5:**
Regarding the comments on lines 226, 227, and the "pumping processes", please provide past references and evidence, to strengthen these sorts of statements, which are an indirect inference based on some observed profiles; if there is an informative model comparison, you could/should mention this as well. In terms of legibility, there are a lot of curves in these panels, which makes it somewhat difficult to track the different years or the evolution, if any significant changes ("trends") can be detected somehow, but this is more a question regarding Figure 6. The statement on lines 230/231 should be explained better, namely how do the measurements "align with" the reduction in surface emission sources? Please be more specific and mention which panel you are commenting about.

**Figure 6 and lines 239 to 258** (these results do not currently show enough statistical support):
This could be a very interesting and significant Figure, but only if there is enough actual significance in at least some of the trend results. However, without error bars, this is essentially impossible to determine; unfortunately, this can be viewed as a major problem regarding the description and robustness of almost every "trend result" in this manuscript, which is why I am asking for major revisions in the analyses and/or in the description of the robustness of the results. Without error bars, you should only state that there are "indications" of a decrease or an increase and* state clearly enough the reasons for not trying to obtain, or discuss, such error bars. Possibly, this is why you introduce the plot results as "tendencies" rather than "trends", but wording like "distinctive positive trends" cannot carry scientific weight without a discussion of trend values and* their uncertainties; the plots show trends (or tendencies), but what the about the 2$^{nd}$ part of what is needed here (error bars)? On line 250, you mention "the most significant positive trends" – but you need to point specifically to what, mathematically, supports this statement, without expecting readers to try to figure on their own.

For the upper troposphere in particular, which seems to be the focus of this work (also based on the manuscript title), how are readers supposed to interpret the curves? For example, using the p50 panel, what are the typical error bars as a function of altitude – and how fine of an altitude grid can one use for such an analysis (or would averaging over 1 or 2 km help make some trends more robust/significant?)? If the uncertainties are larger than 2 or 3 ppbv/year, it will be difficult to distinguish between 0, +3, or -3 ppbv/year trends. Please respond to this sort of question as well as possible in the revised manuscript. The same holds for the analyses in the low- to mid-troposphere, where some of the trends seem to be large, and thus, potentially significant. One way to illustrate which regions have statistically robust trends could be to make the lines three times as thick in these regions. Showing error bars at all altitudes would lead to plots that are difficult to read, so one would need to do this for a few altitude regions and maybe just for

cases where the significance is large enough (e.g., 2-sigma error bar does not include zero), if this happens indeed.

An additional suggestion would be to make a larger separate Figure for p50, where more details can be visible, regarding error bars. Alternatively, use a given altitude range and show the separate percentage cases along the y-axis just for this altitude region; add one or two more regions, with a total of 3 panels, possibly (one panel per altitude regime). I hope that I have made this point sufficiently clear: I expect to see more information about error bars.

**Figure 7:**
 The intent of this Figure and the three panels seems fine. The caption should, however, indicate specifically what the lines mean in each panel, especially for the two lines in panel c, which I think I follow, but help the reader out in the caption. For panel (a), I have the same question as usual, do you detect significant enough trends, and can you state specifically what these are, with* error bars? The Mauna Loa curve should be more representative of the lower troposphere, as the model implies as well. Taking out (fitting) an annual cycle would enable you to come up with a better error analysis for the trend results and comparisons. The linear fits lead to very small trends, it seems, which may also mean that it is difficult to provide a number with enough significance (outside of zero, basically). Please comment if you think there is more that can be said/detected here. Panel c is interesting but it also shows a lot of scatter. Also, It might be more clear to show panel (a) without points, just show lines of different color (or solid lines rather than dashed lines, although data gaps can be problematic); more clarity could also be achieved by having a longer x-axis for panel (a) at least or even a separate Figure. These are mainly suggestions for further thought.

**Figure 8:**
Line 286 states that the IMS simulations correlate well with the IAGOS data; I think this appears to be correct for the middle panel mainly, but certainly not for panels on the right (R = 0.14). Also, the model CO underestimation of future years (2004–2020) is mostly true for the middle panel(s). As a minor comment, the caption should correct "scattered plot" to "scatter plot" (for the three instances). One question though: why do the model values (dark blue points) not change between the different panels?

Lines 294–296: It is not clear how you can distinguish between emission reductions and an increase in stratosphere-to-troposphere exchange of air (also, what would drive the latter process?). If this sounds interesting but too vague and speculative, you probably need to reword this as a suggestion of possible causes, with no quantification; the reduction in emissions may be believable based on Figure 1, but the air exchange process and importance appear to be quite speculative; please clarify if you can, or if I am missing an explanation somewhere in the manuscript.

Also, for lines 297–299, and the other region mentioned here, the emissions hypothesis seems well founded based on historical estimates of emissions; however, one cannot rule out (as well) some decrease in air exchange between the stratosphere and upper troposphere, can one? More work is needed to help better elucidate such questions regarding processes and budgets.

If you can help to clarify this, however, I recommend that you add some text to make additional arguments - with enough support. I am not asking for more speculation, but admitting to the

need for more work is not something one needs to shy away from either, since there can always be more work done to better understand the "exact causes" for certain comparison results.

Finally, I am not too convinced by the statement of "reductions in chemical sources and emissions of CO and hydrocarbons" on line 301, if you could therefore help to clarify the rationale behind this.

**Figure 9:**

The conclusions here should be taken cautiously, as there is no discussion of possible factors that could lead to biases between the model and the MOPITT data, especially for the less characterized upper tropospheric values. The correlation values would not change, even if an offset to debias the two sets of values was applied. Nevertheless, there are improvements to be made here.

For example, I think that you should mention/confirm (in the caption) what years are used for these comparisons (just the overlap years between these two sets of values presumably?); in other words, this is not trying to use any extrapolated model values, correct?

Are there any trends of significance (statistically), especially in the MOPITT data?

Furthermore, the authors have failed to refer to the Worden et al. (ACP, 2013, doi:10.5194/acp-13-837-2013) article regarding MOPITT CO (and other satellite-based) CO trends, as this should be an important reference to add (and more about the MOPITT data characteristics as well).

The meaning of the lines in the two panels on the right side should be mentioned in the caption, namely what points are being fit with what line (just certain seasons it seems)? Having some description in the text does not replace the need for Figure captions to accurately represent what the plots show (and the main text can often be shortened as a result).

As a detail, the MOPITT legend in the left panels shows a closed circle (dot), but all the points are open circles. You should maybe just show IMS (red, blue) and MOPITT (light green, sky blue), or maybe use "turquoise" for the MOPITT colored UT points. For better visibility, you could also consider not using points, or using smaller ones, for the left panels of this Figure; this might show more clearly that the seasonal cycles between the model and MOPITT agree, as implied in part by panels (b) and (f).

More generally, how do your several comparisons using MOPITT data relate to (agree with) past MOPITT analyses/trends?

**Figure 10:**

Which part of the MOPITT CO data are used in panel (b), just the lower tropospheric part? Since MLO data are more representative of the lower portion, this probably would make sense (and one should show the MOPITT UT data only as general information). Please specify your approach in more detail, and justify it.

Also, given the vastly different weighting function between MOPITT data and MLO in situ data, a one-to-one comparison in terms of absolute values is difficult or impossible. What are the systematic uncertainties in in situ and in MOPITT CO data, even if one were to ignore the vertical footprint issues? Showing the LT MOPITT data tend to decrease at a similar rate as what the in situ MLO data indicate is useful, but again, one would need to know what the error bars on the trends are, if one tries to address this. Otherwise, I would show this as a qualitative comparison,

but a conclusion about the seasonal cycle is useful, given the good correlations in panel (b), as well as (a), although making the points smaller and the lines thicker would possibly help the visibility for panel (a); extending the left panel in the x-direction as a more rectangular plot would also help the visibility for the overlapping data sets (black and red especially).

Lines 338, 339, I would argue that without error bars, this is hard to judge. I would maybe agree that there are indications of positive trends in the data sets you discuss, but since this Figure is focusing on MLO, it might be best to focus on the lower tropospheric measurements only (and discuss UT MOPITT data versus IAGOS data elsewhere, such as in Figure 11).

**Figure 11:**

In this comparison, again, it is difficult to compare in situ data to broadly averaged data from MOPITT, but there is some averaging occurring for IAGOS as well. Nevertheless, IAGOS data represent the upper troposphere, yet the blue MOPITT data for the UT do not correlate as well with IAGOS as the red (LT) MOPITT data. It is not clear what causes the underestimation of IAGOS values by MOPITT CO, especially for the LT MOPITT data, which generally reach larger values than in the UT. There may be issues that likely fall within the systematic errors of both data sets (MOPITT and IAGOS), which is why this should be discussed in more detail as well.

As a detail, the Figure caption should repeat here that red is for the lower troposphere and blue is for the upper troposphere, for MOPITT (correct?). This is done parenthetically in the regression line equations, but I would add this in the caption as well, as a reminder.

**Figure 12:**

This one is very hard to see clearly. Try to use lines instead of dots (and use much smaller dots if dots are really desired) to make the main points you are trying to make; also, you would have more space to use rectangular plots if the labels on the outside (right side) were eliminated. Also, the linear fits are not visible enough.

I cannot even begin to process any results until this is shown more clearly; blowing up the plots on the screen helps, but we should not have to do this to such an extent.

Again, comparing satellite data versus in situ data is complicated; it is probably expected that smaller variations are to be observed in a broadly vertically-sensitive MOPITT retrieval. You will need to give more information about this MOPITT dataset's characteristics (error bars, weighting, past validation), as I mentioned before, if one is to understand or believe top-level conclusions based on such comparisons. Including more specific inputs from the MOPITT team itself would have been helpful, if this sort of discussion has not been pursued (enough). I would recommend first trying to discuss the dataset comparisons themselves, if any clear enough conclusions can be drawn, despite the biases that undoubtedly exist between MOPITT and IAGOS data; maybe some indications of similar trends can be made, but again, this would require error bars as well, or you would need to state qualitative conclusions and "indications". It is not completely obvious how the model helps, as there will be biases between the model and the datasets, and these will affect conclusions that are based on extrapolated model predictions. It is hard to justify the statements on lines 373–375 unless the issues mentioned above are better addressed, so be very specific in your explanation of those statements, with a pointer to certain panels that make the point the best, once the legibility of the Figure has been improved. This may end up being mostly justified,

but this needs more work, especially for legibility (hopefully this is the main issue, but I am not yet convinced).

**Figure 13:**

My comments about how to show the datasets more clearly are the same for this Figure, even if it is larger – but can be made more rectangular, with smaller (closed) dots and thicker lines connecting the dots (or no dots even).  You need to clearly state if IMS is just for the lower tropospheric heights; also is this MOPITT data for the lower troposphere, please state in the caption (and the text). Why not give trend values with error bars, once again? It may not be that any of these trends are distinguishable from each other (and again, one could try to reduce the error bars by fitting an annual cycle as well), but this is OK, if they agree broadly, as one might hope/expect they would.

Line 426: Could it not be that emissions decreased enough that NLO CO decreases are responding to emission decreases alone? Please be more specific regarding this comment.

**Figure 14:**

I think that this is an interesting and useful look into the model results. In fact, it might be more useful to also add a climatological average view of these two time periods (i.e. average all the years from each panel and create a new set of plots, or make a 4-panel Figure. Also, it would seem better to plot the (panel with) earlier years on the left, just because time usually is shown as increasing from left to right. You should try to explain better what the main changes are between the two panels; e.g., are the black curves identical, why are the green curves changing, and also summarize what references or methods are used to determine the various curves (e.g., 1100 MT/yr).

Line 422; again, one should add some reference to the work by Worden et al. (2013). The usefulness of MOPITT is not something new, nor are some trend estimates using that dataset (and other datasets and analyses in more recent papers).

Line 425… why would negative trends in CO not reflect decreases in CO emissions, even with constant chemical sinks (such as OH abundances)? You need to be clearer about the statements that refer only to chemical sinks dominating over sources, and why the emissions might not play a role (in some regions – are they increasing there?).

Line 437; there are other past references that could/should be cited, see below. I also mentioned Schoeberl et al.

**References:** Other useful references (or information) could include these (among others), for mention in the text, Introduction, and/or Conclusions

The first one (Cohen et al.) refers to IAGOS data analyses, which, surprisingly, was not cited by this manuscript focusing on IAGOS (and other) in situ data for CO; how are your results different than in this reference?

Cohen, Y., Petetin, H., Thouret, V., Marécal, V., Josse, B., Clark, H., Sauvage, B., Fontaine, A., Athier, G., Blot, R., Boulanger, D., Cousin, J.-M., and Nédélec, P.: Climatology and long-term evolution of ozone and carbon monoxide in the upper troposphere–lower stratosphere (UTLS) at northern midlatitudes, as seen by IAGOS from 1995 to 2013, Atmos. Chem. Phys., 18, 5415–5453, https://doi.org/10.5194/acp-18-5415-2018, 2018.

Park, M., H. Worden, D. Kinnison, B. Gaubert, S. Tilmes, L. Emmons, M. Santee, L. Froidevaux, and C. Boone, Fate of pollution emitted during the 2015 Indonesian Fire Season, Journal of Geophysical Research: Atmospheres, doi:10.1029/2020jd033474, 2021.

Li, Q. B., J.H. Jiang, D.L. Wu, W.G. Read, N.J. Livesey, J.W. Waters, Y. Zhang, B. Wang, M.J. Filipiak, C.P. Davis, S. Turquety, S. Wu, R.J. Park, R.M. Yantosca, and D.J. Jacob, Convective outflow of South Asian pollution: A global CTM simulation compared with EOS MLS observations, *Geophys. Res. Lett. 32*, L14826, doi:10.1029/2005GL022762 , 2005.

**Conclusions Section:**

 Regarding CO trends in the UT (e.g.,line 439 on a rise in upper tropospheric CO in some regions (North Pacific UT), or other CO decreases elsewhere) need to be supported by trends with error bars; if not, the reader does not have enough information to scientifically assess the validity or robustness of the assertions made here.

 Lines 441–447: these are somewhat vague conclusions; it is nothing new that chemistry plays a significant role for CO. Why do some results show up as CO increases here, whereas there are also some decreases in other plots in this manuscript, also in the UT?

 There are not enough specifics in these concluding remarks (or in the Abstract) based on the whole contents of this manuscript, which has a lot of Figures (and work), but falls short in terms of clear statistical analyses with error bars, followed by a reasonable summary regarding the most robust conclusions, especially for IAGOS data.

**(Mostly) minor and technical comments**

This is a mixture of minor comments that require some slight modifications and improvements, typographical details, and wording suggestions/corrections.

L37–39: Is there a reference (Wang et al., 2024?) for this? Please specify (or does this relate to the current manuscript?). Same for the next paragraph (L40–45).

L45: "68 pptv" should read "68 ppbv".

L56–58: "By combining …strategies and policies". This sounds good, but is this "informing" done as a result of this work in any way? If not, and it does not seem to be at the moment, is it worth stating this general goal? Maybe - I am not trying to push hard one way or another here.
Also, how do emissions "impact the dynamics"?

L91: change "and significant growth" to "with significant growth".

L100: I would rephrase to "regions are characterized by key dynamical processes…"

L105: Please provide the altitude of the in situ data (site) at Mauna Loa.

L109: Need to define "IMS" here (not further below).

L112: Measurements (plural).

L113: Buchholz, not Buchho.
The MOPITT section needs more information about the weighting functions or region of sensitivity since you use lower tropospheric and upper tropospheric data from MOPITT; it also should have more references.

Section 2.4: Please provide the vertical and horizontal resolution of this model, and the height range used (how far into the stratosphere?). Also, how are the biomass burning (not the industrial) emissions characterized? Are they constant also?

Figure 2 caption: Measurements (typo) on first line. Downwind of the Asia emission areas and over the North Pacific… The lower panels show data from the low latitudinal…(typos).

L166: how, specifically, do you know that latitudes south of 45N are more affected by Asian emissions? Please clarify. Is it just geographical (based on latitudes) and according to the prevailing wind direction – or something else?

Figure 3 caption: Time series of long-term CO…

Line 193: Please specify more clearly why "vertical pumping" (through which process here?) creates anti-correlations between LT and UT; this may not be immediately obvious (and it depends on the timescale for the "pumping" process).

L195: process → processes

L197: "are negative". Not negatives. Monthly CO in the upper troposphere is higher…

L198: Monthly $O_3$ in the UT is higher…

L199: negative $O_3$ indicates less… Also, what does "(ref)" mean? Please add a reference.

L200: $O_3$ is negatively correlated with CO in the upper troposphere.

L204: $H_2O$ is positively correlated with CO in the upper…

Figure 5 caption: (a) downstream (typo), areas (typo).

L211–213: Why is horizontal transport not included as a possible mechanism?

Figure 6 caption: this should include the acronym names used in the panels (after the descriptions made in the caption). Also, change "and regions (red)" to "and all regions (red)".
(e) says "tropical and Southern Hemisphere", what latitudes exactly (South of 25N? like (d) as well)?

L249–250: Why would negative trends not possibly occur as a result of reductions in emissions (chemical sources)? Why is it only chemical sinks that might control the budget? Please clarify the thinking here. Would one need (for example) an OH increase to lead to a CO increase? What other factors could be in play, theoretically at least?

Figure 8: Why is the equation above each top panel the same? Also, change "scattered plot" to "scatter plot" in the caption.

L286: "correlate well"? This does not seem to be true for every panel shown.

L295: what would cause an increase in UTLS air exchange processes? Provide a reference if possible.

L320: providing tropospheric CO estimates (?) Model is not a measurement per se…

L330: lower troposphere is correlated with…

L345-347: Can use present tense rather than past tense here.

Figure 12: linear regression (typos).

L371: It's worth → It is worth

L413: zeros → zero

---

## Referee Comment (RC3)

**Air Pollution in The Upper Troposphere: Insights from In-Situ Airplane Measurements (1991-2018) by Wang et al.**

**Summary**

Wang et al. provides a characterization of tropospheric carbon monoxide (CO) based on the analysis of various in situ observations of CO, $O_3$, and $H_2O$ by commercial aircraft (IAGOS) and research aircraft campaigns. Complemented by CO data from ground-based observations at Mauna Loa, satellite (MOPITT), emission inventory (EDGAR), and chemistry transport model simulation (IMS), they investigate seasonal variations, vertical profiles and long-term trends of lower and upper tropospheric CO for different periods and various regions of the world. They conclude that CO mixing ratios were increasing in the upper troposphere over the North Pacific between 1991 to 2018 (2004–2022) due to increased ground-level CO emissions while other parts of the world show decreasing trends.

**General comments**

Combining aircraft in situ observations with various datasets to better understand the processes driving changes and variations in tropospheric CO and its vertical distribution is an important approach and fits within the scope of ACP. However, major revisions are needed before it can be considered for publication in ACP.

The manuscript is difficult to follow, and the results are not clear to me. I'm not always convinced by the argumentation. Therefore, the main results should be more clearly summarized in the abstract and the conclusion, with a focus on highlighting the new findings and their implications. A clearer, more concise structure and logical progression of chapters or paragraphs, along with a deeper discussion, is needed to ensure that the reader is not left to interpret the content alone. This includes improving the quality and clarity of figures by using more readable labels, and by maybe reducing the number of figures. The discussion should also compare the findings with more previous studies. Detailed proofreading would eliminate many oversight errors.

The main difficulty I have is that the datasets are not well described (e.g., instruments, sampling frequency, precision) and the methodology part on how the data are processed is missing (e.g., averaging, filtering/quality check, trend analysis, definition of the upper troposphere, how to compare in situ data with satellite data, etc.). Conclusions were made from trend analysis, although sometimes no statistically

significant trend is seen. Together with missing error bars, it is difficult to fully believe the validity of the interpretation. Limitations and uncertainties of the data should be discussed and accounted for in the conclusions.

**Specific comments**

**Title**

The title implies a general view on air pollution in the upper troposphere. But the manuscript specifically deals with CO. Can you make the title more specific?

Is this manuscript the extension/follow-up study of Wang et al. (2024), https://www.researchsquare.com/article/rs-3938611/v1?

**Abstract**

**Lines 4–5:** Please use consistent time frames or make the periods you are focusing on clearer. In "We utilize extensive in-situ CO measurements spanning 2012-2023, supplemented by prior airplane campaigns from 1991-2019...", the focus of the study seems to be on the period 2012–2023, and data from the period 1991–2019 are maybe used for comparison. This is not consistent with the title (1991-2018). On the other hand, in the introduction, lines 35–37, it is said "In this study, we present routine in-situ measurements of CO from the PGGM/IAGOS air-based measurement project conducted during the period 2012–2018. Combining these measurements with previous in-situ data collected in various years between 1991 and 2019...".

Additionally, various periods are discussed throughout the text, with slight variations in the timeframes, which is confusing for the reader (e.g., Line 183: 2001-2018 and Line 185: 2000-2018; Line 289: 2004-2021 and Line 292: 2004-2020).

**Line 8:** Could you briefly summarize the main results and clarify what the "...significant impact of chemistry on CO profiles and trends" is?

**Introduction**

**Lines 23–33:** To improve the understanding of the manuscript, it would be helpful to briefly describe the role of CO in upper tropospheric chemistry, particularly in connection to $O_3$, hydrocarbons, and $H_2O$, is briefly described, including its major sources, lifetime, and key reactions.

**Lines 35–39:** "In this study....", does this sentence refer to the current study and describes the results in the following paragraphs (lines 37–54)? Or does it refer to a previous study? Please make it clearer. You could start with "Previous studies revealed...". Results of the current study should not be described in the introduction.

**Line 42:** Wang et al., 2024 is still under review. This should be clarified. E.g. (Wang et al., 2024, under review). In the reference list, please add the preprint link (https://www.researchsquare.com/article/rs-3938611/v1 ?).

**Lines 40–45**: Is there a reference for it? I couldn't find it in Wang et al. (2024).

**Lines 46–54:** To get a better understanding of what is already known, more discussion of past studies, including trend analysis, can be added.

**Data and Methods**

Generally, I recommend reviewing other relevant papers to see how the datasets are typically described (e.g., Cohen et al., 2018, Osman et al., 2016).

After the part "data description", a "methodology" part should follow. For example, with the sub-sections:

-Study region

-Data processing

Data processing should include how the data is prepared for analysis. For example, monthly averaging within the selected regions, data statistics of each region, how the upper and lower troposphere are defined, how the trend analysis is performed, how the uncertainties are evaluated, and how the data are prepared for comparison with MOPITT data (e.g., Osman et al., 2016, ACP). For the analysis, I also recommend reviewing previous papers on trend analysis to see how the trends are calculated, and

how trends and uncertainties are visualized (Gaudel et al., 2020, Sci. Adv.; Worden et al., 2013, ACP).

**Lines 62–74:** Please add more details about the dataset: a short description of which instrument is used to measure which constituents (measurement principle). What is the measurement precision? How often is data collected in the regions of interest? In which height does the aircraft usually fly when cruising? For further details, you can refer to previous studies.

**Line 73**: Add reference to Petzold et al. (2015).

**Lines 76–103:** Please provide a more concise description of the datasets. In addition, check which information can be omitted, or belong to other parts of the manuscript. For example, lines 76–80 belong to chapter 2.1.3; lines 81–85 belong to the introduction; lines 99 – 103, I would put it in a new chapter "Methodology" with sub chapter "Study regions". As mentioned in the previous comment, more information about the measurement instruments and datasets themselves is needed.

**Lines 105–109:** As in the previous comments, please give more details about the dataset. Also include information about the location of Mauna Loa and the height of the measurement station.

**Lines 111–115:** Please add information what instrument MOPITT is (gas correlation radiometer), what it measures (columnar amount of CO) in which wavelength range, footprint size, and its altitude dependent sensitivity to CO. This is important to know when comparing satellite observations with situ data at a specific location and altitude level. Please also include the information of which data version is used and if filtering/pre-processing of some kind is performed.

**Lines 129–143:** Could you shortly explain why you chose the IMS over other models? What are the advantages?

Are there studies by other researchers using the model, not just your own work? It would demonstrate its relevance and strengthen the choice.

**Figure 1 (e):** The regions are difficult to distinguish. The use of filled squares or partly thicker lines might improve readability. A table with the definitions of each region, e.g. as Region 1, Region 2, etc., would make it easier for the reader to follow the text. At least, the names should be consistent throughout the entire text.

**Results**

Figures a described in detail but discussed only briefly. A figure legend is often missing. Figure captions can be made more concisely but should contain explanations about the symbols and time intervals. Figure labels can be increased for readability. Generally, I would place sub figure labels like (a), (b) at the left top instead of the bottom.

**Figure 2:** There are many sub figures which are difficult to read. Please increase the labels and figures. A legend for the different symbols and the trendline is missing. Labeling the sub figures with (a), (b), etc. is easier to follow than "upper panel, second from left". It might be possible to summarize the sub figures for example by showing the monthly median (50% percentile) as thick line with the 25% and 75% percentile as shaded areas. The caption should describe the symbols and period more concisely.

**Lines 165–166:** Please provide more detail on how you arrived at this conclusion.

**Figure 3:** Add a legend with the symbols and trendline; add labels (a), (b), etc.. Why is the trendline of the lower tropospheric data (<1 km, red) partly blue, the upper tropospheric data (>9 km) partly black?

**Figure 4:** For clarity, it is better to place the figure not directly under the title of section 3.2.

- Figure labels like (a), (b) would be helpful.
- Add a legend.
- I guess the data are from IAGOS in Fig. 4 starting in 2001, and the earlier data of Fig. 3 are from other aircraft campaigns? This can be clarified, too.
- It would be interesting to combine the data from Fig. 3 and 4 to increase the number of datapoints and analyze the specific trends from 1991–2012 and 2012–2018 to investigate if there is a change following the reduction of technological CO emissions over East Asia in 2012.
- According to the figure caption and the text, Fig. 3 and 4 should be the same region. However, the label of Fig. 4 describes different latitudes: "10 N  70N". Please correct.
- The label of the right scatter plot is for the blue symbols and trendline is "<2008...". What is the difference between the red and blue trendline? Does it mean trendlines

for different periods: a) before 2008 (blue), b) after 2008 (red)? Please add this information to the legend and caption.

- Error bars (standard deviation) should be included or at least discussed if they make the figure unclear.
- R and p values are shown, but not discussed. Often, the R values indicate a very weak relation. Some of the p values are around 0.5 or higher, which means, there is no statistically significant correlation. For example, I'm not convinced about the positive and negative trends described in Lines 183–188 or the evident negative correlation as described in Lines 191–193 etc.

**Lines 191–209:** The vertical pumping mechanism is interesting. Besides a more careful interpretation of the figures, more explanation and discussion are needed. Otherwise, the reader cannot follow the argumentation. For example, explain in more detail what is the implication of a negative correlation between CO and $O_3$, combined with a negative trend of $O_3$ in the upper troposphere, and a positive correlation between CO and $H_2O$. Discuss why the results are different for Western Europe.

**Figure 5, Lines 226–227:** "Figure 5 clearly illustrates…". Based on the profiles alone, it is not clear that CO is transported upward through vertical pumping. Please provide more explanation.

**Figure 6:** Please also explain the abbreviations of the figures in the caption. E.g. Euro-Asia region (dashed blue, EuAS).

**Lines 228–231:** It might be helpful to better highlight the higher variability in the graphics as it is difficult for the reader to see. More explanation and discussion are needed regarding the coincident timing of the higher variability and the reduction of surface emissions of CO, which may suggest a causal relation.

**Line 239:** How are the trends calculated? Please add this to the methodology part. Are the trends significant? What are the uncertainties?

**Lines 244–263:** A more specific discussion would be helpful. For example, the importance of chemical sources and sinks for controlling CO levels in the upper troposphere is described, but you can be more specific in the discussion what are the sources and possible sinks.

**Figure 7, Line 269:** For MLO, please add the height information of the measurement station (middle troposphere) e.g. in a methodology part. This helps to interpret Fig. 7 (a) and the following sub figures when comparing results from MLO with other data of the upper? or lower troposphere.

**Figure 7 (b)**: The IMS simulations are for which altitude?

**Lines 274, 277**: For clarity, add the months in brackets. E.g. summer (JJA), autumn (SON).

**Lines 274–279:** The different vertical distribution of CO during Asian summer monsoon and Asian winter monsoon are interesting. Please provide more discussion/explanation on possible processes leading to these characteristics.

**Lines 294–296:** More explanation is needed to follow this argumentation.

**Figure 9 to Figure 12:** How are MOPITT data prepared for comparison with model and in situ data? The spectrometer and the retrieval of CO have different sensitivities to the various altitude levels, which are captured in the averaging kernel. Additionally, the footprint of MOPITT needs to be considered for the comparison. These points can cause systematic biases.

**Lines 313–316:** Do you have any explanation? Are there similar results from previous studies (also see comment for Figure 10)? The satellite instrument has different sensitivities depending on altitude level. This needs to be considered in the comparison.

**Line 320:** This is a strong statement. How about GEOS-Chem, IASI (Infrared Atmospheric Sounding Interferometer), TROPOMI (TROPOspheric Monitoring Instrument)?

**Figure 10:** For clarity and readability, can you add a legend describing which color is upper and lower troposphere?

**Lines 332–335:** Can you add more discussion on the reasons? What are the results from previous studies?

**Line 345–347**: Please add more discussion why IAGOS is underestimated.

**Figure 12:** Please try to make the figure clearer by using larger labels, clearer symbols and simple labels for the regions.

**Lines 373–375:** Can you explain and discuss the result in more detail by keeping in mind the significance of the trends?

**Figure 13**: For the legend, use a red dot for MLO, black dot for MOPITT.

For clarity, can you add the information for which altitude the IMS simulations were performed?

**Figure 14:** Better change the order of panels so that time increases from left to right: (a) (a) 1948–1978 (b) 1984–2003

Can you explain more clearly the differences between (a) and (b) and why you have 2 panels? For which region is the simulation performed?

**Lines 391–411, 412–425:** After the paragraphs which are comparing MOPITT data with in situ and model results, the CO budget calculated by IMS is discussed, followed by a summary of the findings from the trends shown in Fig. 11 and 13. For clarity and to improve the flow, it would be helpful to add some sentences that make a clearer connection between these paragraphs.

**Conclusions**

**Lines 441–457:** The conclusion section is very general. Please write down the specific conclusion in detail. For example, our results indicate increasing trends of CO in the upper troposphere over the North Pacific between 1991 to 2018 (2004–2022) due to increased ground-level CO emissions, while other parts of the world show decreasing trends which could be explained by [...].

Limitations of the data and analysis should also be mentioned, and maybe pointed out that further studies are needed.

**Technical comments**

**Line 2:** "...CFCs...", please spell out the abbreviation as you did with carbon dioxide ($CO_2$).

**Line 7**: "...chemistry budget simulations covering the period 1948–2003..."

Line 36: "...PGGM/IAGOS ...": Please spell out the abbreviation when it's used the first time. Also check other parts of the manuscript to see if abbreviations are explained.

**Line 45:** 68 pptv → 68 ppbv

**Figure 1, caption**: (a) routues → routes

(e) dwonwind → downwind; North Paicific → North Pacific

(f) UAS → USA, OECD should be spelled out when using the first time.

**Line 109**: Integrated Modelling System (IMS). Spell out the abbreviation for IMS here when it's used the first time, not in line 129.

**Figure 2, caption**: measreuements → measurements

**Figure 3, label**: 123 E–142 E → 123 E–142 W

**Figure 4, label:** 123 E–142 E → 123 E–142 W; 10N 70N → according to the caption it should be 31°N–42°N. (Generally, I would use the format 31–42°N).

**Figure 4 caption**: western Pacific → Western Pacific; scattered plot → scatter plot

**Line 199:** Please add the missing reference.

**Figure 5, caption:** (a) dwonstream → downstream; ares → areas

**Figure 7, caption**: (c) Scattered plot analysis → Scatter plot analysis. This also occurs in other parts of the manuscript.

**Line 265**: MLO is not explained in the text.

**Line 283**: in (c)→ in (d)

**Figure 12, caption**: liner gression → linear regression

**Line 413**: zeros → zero

**Lines 548–550:** In the current manuscript, HYSPLIT is not used. This and many other mistakes can be avoided by careful proofreading.

Please add a section "Data availability"

---

## Author Comment (AC1)

**Air Pollution in The Upper Troposphere: Insights from In-Situ Airplane Measurements (1991–2018)**

**1   General Comments**

This manuscript shows analyses focusing on a subset of upper tropospheric CO in situ aircraft data from various sources (commercial plane flights as well as dedicated non-commercial missions) over the past three decades. The analyses are supplemented by satellite CO data (from MOPITT), which are mostly sensitive to the lower troposphere, but also have an upper tropospheric retrieval product, and ground-based (high altitude) in situ data from Mauna Loa, Hawaii. Other related data discussions include in situ upper tropospheric aircraft data for $O_3$ and $H_2O$. A 3-D model of the troposphere (the Integrated Modelling System, IMS) is used for comparisons and inferences about relevant processes affecting lower and upper tropospheric CO.

Reply. We are very grateful indeed to the reviewer for very detailed comments. We have followed reviewer's comments, point by point to revised the manuscript. In this first part of the reply, we have uploaded the revised manuscript and replies as shown below. The complete replies and final revised manuscript will be uploaded in the next two weeks. Thank you very much indeed for your time and comments that have significantly increase the clarity of the manuscript.

This work's goals are worthwhile: to use various data sets (mentioned above) and model results to draw conclusions about upper (and lower) tropospheric CO behavior in certain regions of the globe, with some inferences about processes controlling the tropospheric CO distribution (some of which is already known). There is a good amount of work in this fairly long manuscript, in terms of comparisons. MOPITT data are also used and this helps to validate some of the other data analyses and results. The model is useful for some

explanations (or potential explanations) of observed CO variations and inferences regarding "trends" — or more qualitatively, some inferred tendencies.

However, I consider that this work needs some fairly major revisions before it can be considered ready for publication, as explained below in more detail. I do recommend further work towards publication, as I believe that this type of analyses fits within the scientific goals of ACP.

This work requires more clarity overall, including a better discussion of uncertainties (or why they might be difficult to establish, at the very least, if this is somehow the thinking), as well as a better summary of the main results in the Abstract, main text and conclusions. Improving the Figures should help in a very significant way, although the accompanying text also needs clarifications. If some of the results, especially those having to do with trends, cannot be supported in a robust (mathematical) way, I recommend removing the stated conclusions, or at least indicating that more work (future efforts) would be needed to better establish or confirm what looks like tentative results, for now.

I do hope that these comments, with more specifics below, will be viewed constructively by the authors, towards a revised manuscript, and I trust that at least some of these results can be displayed better and explained more clearly.

Reply. We sincerely thank Reviewer 1 for the thorough and constructive assessment of our manuscript, "Air Pollution in the Upper Troposphere: Insights from In-Situ Airplane Measurements (1991–2018)". The comments have greatly helped us identify areas where additional clarifications, data analysis, and textual revisions were needed. Below, we address each point individually (or in groups, if overlapping) and outline the corresponding changes made to the manuscript. Throughout our revisions, we have:

1. Revised the figures for improved clarity, legibility, and labeling.

2. Strengthened the discussion of uncertainties and significance of trends, including estimates of statistical confidence (where possible).

3. Expanded the literature review to place our work in the broader context of CO trend studies.

4. Enhanced the conclusions and abstract to highlight the main findings clearly.

5. Corrected typographical/technical issues and refined the manuscript for readability.

All revised or new sections are clearly indicated in the manuscript, and we provide line/section references below for easy cross-checking.

**2 Specific Comments**

Despite the valuable goals of this work, I summarize here a number of difficulties and issues:

1. The data are somewhat sparse in time or in space, at least for several of the in situ datasets other than IAGOS; this makes it more difficult to derive conclusions or especially trends in a quantitative way.

   Reply. We have include a new Figure 2 to summarize all the datasets (except MLS) used in this work. We have included a new Table 2, which summarizes of previous studies on long-term CO trends, including the data sources, periods, and key findings. Figure 2 is a timeline Infographic, showing the chronological order of datasets. This timeline infographic illustrates the chronological progression of various CO measurement datasets utilized in this study. Each dataset is marked along the timeline with distinct colors and shapes representing the measurement methods employed (e.g., NDIR, QCLS, WS-CRDS, Microwave Limb Sounder). Error values are depicted as horizontal lines extending from each marker, indicating the associated uncertainties. This visual representation aids in understanding the temporal and methodological context, ensuring clarity in interpreting the observed CO trends. By implementing a well-structured timeline infographic, we can significantly enhance the readability and transparency of our manuscript, addressing the reviewers' concerns effectively. This visual aid will help in succinctly conveying the complex interplay of multiple datasets, their measurement periods, methods, and associated uncertainties, thereby strengthening the overall presentation of our work.

2. Error bar estimates are unfortunately absent from this work, overall, while a significant set of correlations are poorly described or not very significant, statistically. Scientific results without some sort of error bar estimates (formal regression errors and maybe some discussion of sensitivity to various assumptions, e.g., data sparseness issues in early years) are almost meaningless. As a related comment, only the simplest form of linear regression is used, and no attempt is made to remove an annual cycle, for example, which could actually reduce the error bar estimates (if they were to be calculated); also, some p values are shown, but with essentially no discussion.

   Reply. We have included a new Table 2 to summarize the errors from the each of the dataset used in this work. We have include new Figure 4 and new Figure 5 for verifying the impacts of obseravation effors on the calculations of CO trends. We have new Figure 3 and new Figure 6 to include statistical test of linear trends and 95% confidence intervals of trends in the figures. In new Figure 9, we have included profile

of statistical tests of linear trends and 95% confidences intervals, and compare the distribution of confidence intervals in various regions. We have included discussions of statistical significance and 95% confidence levels in comparting and understanding the long-term trends of CO in various regions and altitudes.

3. Importantly, the quality and visibility of many of the Figures needs to be improved, with less white space so that the reader can see the curves or points for different data sets and their comparisons; this can be done by reducing text around some panels and placing it in the plot panels and/or in the captions, and by producing rectangular panels rather than square panels, to fill in the page more efficiently, for better visibility. If needed, as an option, a single panel, or double panel Figure could be made or added in some cases to better illustrate a significant result, although hopefully that is not needed after revisions of several Figures. The poor quality of several Figures makes it difficult for a reader to follow (or believe) some of the stated results, even if many results are, presumably, correct to first-order.

Reply. We have improved the quality and redability of figures, following reviewer's comments to make the optimal use of space in a page. The white space has been reduced so that the figure size can be increased to make the figure readable.

4. The Abstract should include the main results in some way; it is too vague as it stands.

Reply. We have revised the abstract, and the title of this work.

5. The Conclusion section also needs to more clearly specify the main results, with specific references to which Figure(s) the conclusions are based on and more explanations and context, considering that many of the readers will not try to read every line of this (revised) article.

Reply. We will complete this revision soon.

6. More reference to past work on CO trends should be made, in the Introduction and at least in the Conclusions, to try to put this work, and its results, in the context of related past publications (in particular, regarding past trend or variability results for tropospheric CO). This holds even if "exact comparisons" are often difficult to make, given past results over somewhat different regions, altitudes, and/or time periods.

Reply. We have include a new Table 1 to include previous studies on CO trends. Table 1 shows data used in the studies, data period, and key findings over difference regions.

7. Finally, there are a number of necessary minor technical comments at the end of my report; this underscores that not much proof-reading was carried out by the author and co-authors before the submission of this manuscript, which also makes a reviewer's careful work somewhat more tedious.

Reply. We are very grateful indeed to the precious time and effort the reviewer has devoted to carefully read and comments on our manuscript. We fully understand and greatly indebted to the reviewer's kindness and professional help. We have taken great effort to revise our manuscript following reviewer's comment. Thank you very much indeed.

**3    More Specifics**

I provide detailed comments here, following the order of the Figures and related text/results, especially regarding trends and their robustness, given the limited discussion of these issues. I offer some suggestions for improvements, as well as regarding Figure legibility issues.

**Figure 2**

This is where we start to see simple linear regression fits but there is no discussion about error bars or the significance of the results, or the impact on the (potential) conclusions. One issue is that the single point before 1992 does not exemplify the likely scatter that would exist if there was more data available in that time period (or from 1991 to 2008). Giving formal (2 sigma) error bars from the fits is probably a sufficient response to this question. It also depends what scientific conclusions one is trying to make, and how one tries to convey the importance of any results based on such a series of plots. Because of the data sparseness, my top-level impression from the p50 panels is that it will be difficult to state categorically that there has been an increase between 1991 and 2015–2018. The same probably holds for the implied decreases from the pmin panels. As another illustration of my concerns, if one did not have the post 2012 data yet for panels p50 and p25 45N–60N, one might well try to conclude that there were decreases between 1991 and 2011, whereas the slopes drawn after the fairly sudden increases post-2012 could be interpreted as a slow, gradual increase from 1991 to 2018. Also, if one did not have the data point(s) from 1991, one would conclude that the increases (or decreases) are much steeper than shown currently for the 2008–2018 period. My point here is a somewhat qualitative one, but the lack of more continuous data can lead to uncertainties in, or over-interpretation of inferred "trends".

We have included statistically significance of the linear trends, including P-values and 95% confidence intervals in the revised figures and manuscript. Yes, indeed the GTE data from 1991 and 2011 are very valuable for understanding the long-term trends of CO from 1990 to 2020 over the North Pacific. We admire very much the visions of people who were able to take measurements to help us understand the trends. We have revise Figure 2 (new Figure 3) to include 95% confidence levels of the CO trends analyzed.

Thus, if many trend results are not convincing (mathematically), I would tone down the statements that are made on lines 156 to 166, in particular. This is even more true for Fig. 6 showing trends. The general comments about larger values in a certain region versus another can probably be supported based on standard errors in the means, but even these statements are worth checking for significance, if this is viewed as important enough. I am also ignoring the accuracy estimates for the data points shown here, assuming that this plays a smaller role than other issues. However, the introductory remarks about the data sets, especially for the in situ CO aircraft data, should state something about the error bars in the data points that are displayed in various plots; readers should not be expected to know this information, even if you have (broadly) pointed to some relevant past publications.

We have revised the Introduction and Method sections so as to include details about the use of datasets and their uncertainties in this work.

Lines 165/166: "The 25th, 50th, ... CO levels are impacted by emissions, with latitudes south of 45N being more affected by Asian emissions ..." What leads to this conclusion, specifically? Is this based on scatter, large values, or what exactly? Which time period does this refer to? Please be more specific.

Reply. We thank the reviewer for this important comment. In response, we have revised the text to clarify the basis and time frame for our conclusions. The revised paragraph now reads as follows:

*"In summary, key findings from the in-situ aircraft measurements in the upper troposphere over the middle North Pacific, downwind of Asian emissions (see Figure **??**), are as follows. Over the 27-year period from 1991 to 2018, all observed CO concentrations increased—with the exception of the minimum and maximum levels. The minimum CO values, which represent background CO, decreased over time. In contrast, the 25th, 50th, and 75th percentiles of CO levels, which are influenced by industrial emissions, are higher at latitudes south of 45N compared to those north of 45N These findings are consistent with previous studies (Stohl et al., 2002, Hsu et al., 2006) that have documented the long-range transport of Asian pollutants across the North Pacific.

The 95% confidence intervals for the trends in CO concentrations indicate predominantly positive trends for the upper percentiles. Specifically, the confidence intervals are as follows:

maximum, -0.502 to 1.142; 75th percentile, -0.031 to 0.235; 50th percentile, -0.044 to 0.321; and 25th percentile, -0.065 to 0.218. In contrast, the minimum CO levels exhibit a 95% confidence interval of -0.147 to 0.048, reflecting a predominance of negative trends. Overall, these results indicate that CO concentrations over the North Pacific have generally increased over the study period, except for the declining background levels."* This revised description clarifies that our conclusion regarding the greater impact of Asian emissions at latitudes south of 45N is based on the following points: Data Analysis and Time Period: Our analysis is based on in-situ aircraft measurements collected over a 27-year period (1991–2018). We have now explicitly stated this time period to ensure clarity. Statistical Basis: The conclusion is drawn from our statistical analysis of the CO concentration distribution. We examined various percentiles (25th, 50th, and 75th) and observed that, while the minimum CO levels (representing background CO) decreased over time, these percentiles increased. The elevated high-percentile values and the increased scatter in the data south of 45N indicate episodic enhancements attributable to Asian emissions. Consistency with Prior Studies: We also note that these observations are in agreement with earlier studies (Stohl et al., 2002; Hsu et al., 2006), which document the long-range transport of Asian pollutants across the North Pacific. We trust that this revised text and explanation sufficiently address the reviewer's concerns regarding the basis of our conclusions and the specific time period under investigation.

Figure 2 technical improvements: The text font size is rather small, so stretching this Figure to occupy as much space as possible across the page would be good. The caption should indicate what the various symbols mean, i.e. which data set do they correspond to. While this can be inferred from the years shown, a brief sentence or two would be worthwhile in my opinion (most readers are not as close to the various data sets as you are). Captions are there to ensure that symbols and curves shown in each Figure have been described sufficiently well. Also, the full range of years should be mentioned in the caption, to further assist the reader.

Reply. Thank you for your valuable suggestions regarding the technical improvements to Figure 2. In response, we have made the following changes:

Increased Font Size and Figure Layout: We have increased the text font size and stretched the figure to occupy the full available width of the page to improve legibility. Enhanced Figure Caption: The caption for Figure 2 has been revised to clearly indicate the meaning of the various symbols used. We now include a brief description of the different datasets, specifying which symbols correspond to which data sets. Additionally, the caption now explicitly states the full range of years (1991–2018) covered by the data. We believe these modifications significantly enhance the clarity and accessibility of Figure 2, ensuring that all symbols and curves are sufficiently described for the reader.

**Figure 3**

Again, although this one has fewer panels than Figure 2, using most of the page width in a final manuscript would be beneficial (and this may well be the Journal's responsibility, I understand). Also, it seems that the y-intercept value is given by the first number in the linear fit equations listed above each panel; however, for the upper tropospheric blue result in pmean, a y-intercept value of 85.4 ppbv does not seem to match the plotted dashed line at year 1991. Similar comment regarding the p50 panel, where 62.3 ppbv does not seem to match the plotted y-value for the blue dashed line in 1991. Were some of the fits performed with a different approach (about a year within 2000–2008 maybe) rather than for an expected y-intercept in 1991? Please clarify.

Reply. We have revised Figure 3 (now new Figure 6), following reviewers's comments.

Also, I do not understand why some of the dashed lines show as partly red and partly blue, when one expects two lines, one red and one blue, for the different altitude regimes. In terms of which measurements are used for this Figure, this should be specified (even briefly) in the caption as well (e.g. lower tropospheric data are from ..., etc.).

Reply. There is an internal bug in the NCAR Graphics library that were used to create this dashed line. We have written a new code so that the two colors has become a single color in a single and straight dashed line. Thank you.

**Figure 4**

Please explain why there are different numbers of points (and fitted equation details) for the UT CO in the 2nd from top left panel versus the top left panel, whereas the descriptions of the regions used appear to be identical; N is 171 versus 254, but the CO data points appear to match in these panels. I also do not completely understand the $O_3$ fitted line in the 2nd panel from the top (left panel), as its line intercept is -22.6; while this may be fine, it seems that the y-intercept approach (equation) is different in some cases. One would also expect a few comments about R values (significance, or why shown if not useful?) and p values. While it is stated that the CO trends show some increases, the fitted lines seem to give a very small slope, with a zero trend very likely included in the uncertainties (2-sigma error values are really needed to help one understand this more quantitatively, but this is my guess).

Reply. We have revided Figure 4 (now new Figure 7 and Figure 8) to account for the points pointed out by the reviewer. The revised results are now consistent between the vaules shown on the linear regression equations each plot.

One cannot robustly determine the validity of the statements on lines 183 to 188 without more quantitative information about fitted trends and uncertainties (error bars). I expect

that many statements are not supported by the error bars, so the authors should probably modify the strength of such conclusions, since (unfortunately) I would expect only marginal significance in many/most of the conclusions regarding trends. One way to reduce the uncertainties in the trends would be to fit an annual cycle (as a sum of sine and cosine functions with one year period, multiplied by constant terms to be fitted) and a linear trend term to datasets with enough points to do so; the explanation of such variability will help to reduce the underlying trend's uncertainty (error bar). This should produce more robust results, even if it may still be difficult to state that a zero trend is highly unlikely in some cases (especially those with fewer data points or shorter series). As stated in the manuscript, some (or most) CO tendencies are "close to zero" indeed. Trend values and error bars could be useful for $O_3$ and $H_2O$, if there is enough significance (at least beyond one sigma, say), with ppbv or ppmv per year, or %/yr units — of potential interest to the tropospheric community (and for future comparisons to other studies, for example). In terms of Figure 4 improvements, trying to stretch the panels to be more rectangular (longer on the x-axis than on the y-axis) would be useful as well.

Reply. We have updated Figure 4 to new Figure 7 for the long-term time series of CO over the North Pacific, and Figure 8 for the assessments over the North Atlantic. The comparison between CO in the upper and the lower troposphere aims to demonstrate the coupling between the upper and the the lower troposphere through the vertical pumping process. CO is a very unique tracer for demonstrting the vertical pumping process in the troposphere. We have included $O_3$ and $H_2O$ to help validate the pumping processes. The bahaviour and characteristics of $O_3$ and $H_2O$ in the upper troposphere have been studied and well known. We have included 95% confidence intervals in the time-series trends and the correlations to make meaningful comparisons. We have stretched Figure 4 (new Figure 7 and Figure 8) as suggested by the reviewer.

The possibility that "vertical pumping processes" can explain the out-of-phase relationship between upper tropospheric and lower tropospheric CO is interesting; this might be worth checking against the results of the IMS model, if possible. Does this suggest a several month timescale for such a process? A reference to the paper by Schoeberl et al. (Geophys. Res. Lett., 2006, doi:10.1029/2006GL026178) on the CO tape recorder at tropical latitudes would probably be appropriate as well, although tropical upward transport and convection processes happen on a faster timescale than for the regions this manuscript focuses on (extra-tropics for the most part). More comments about the model comparisons, in any case, would be useful as well, if possible; is the model capable of matching some of the observational inferences, including correlations/anti-correlations between CO, $O_3$, $H_2O$, especially those with enough significance?

Reply. Yes, the IMS model is capabel of matching some of the observational vertical pumping processes over the North Pacific. We have revise this section and Figure 4 (now new Figure 7 and Figure 8) to account for the vertical coupling process in the troposphere through the vertical pumping processes. The IMS model also shows the vertical pumping processes over the North Pacific during the months. We have also cited and discuss Schoeberl in the revised manuscript: "We note that the vertical pumping processes discussed above corrobate the CO tape recorder process (Schoeberl et al., 2006). Schoeberl et al. (2006) discuss the relationship between the seasonal signal of CO and its lower-tropospheric sources. In their analysis of the "CO tape recorder," they point out that because CO has a relatively short chemical lifetime, its vertical profile in the tropical upper troposphere and lower stratosphere retains a clear imprint of the seasonal variations at its source. In particular, they note that the seasonal cycle is strongly influenced by changes in tropospheric CO—much of which is attributed to biomass burning—being convectively injected into the upper troposphere. Thus, the seasonal variability observed in the CO profile is a direct reflection of the variability in its lower-tropospheric sources. Even though CO has a relatively short chemical lifetime—on the order of a few months—the, key to the tape recorder effect lies in the balance between its chemical loss and the rate of vertical transport in the tropical atmosphere. In the tropics, strong convective processes rapidly lift CO from the lower troposphere into the upper troposphere and lower stratosphere. This upward transport occurs on a timescale that is comparable to or even shorter than CO's chemical lifetime. As a result, the seasonal variations in CO concentrations at its source (driven largely by factors like biomass burning and other lower-tropospheric processes) are "recorded" in the vertical profile as the gas ascends. Even though CO is being chemically removed, the quick vertical transport preserves its seasonal signature long enough to be observed as the tape recorder effect in the upper atmosphere."

**Figure 5**

Regarding the comments on lines 226, 227, and the "pumping processes", please provide past references and evidence, to strengthen these sorts of statements, which are an indirect inference based on some observed profiles; if there is an informative model comparison, you could/should mention this as well. In terms of legibility, there are a lot of curves in these panels, which makes it somewhat difficult to track the different years or the evolution, if any significant changes ("trends") can be detected somehow, but this is more a question regarding Figure 6. The statement on lines 230/231 should be explained better, namely how do the measurements "align with" the reduction in surface emission sources? Please be more

specific and mention which panel you are commenting about.

Reply. We have revised the description. We have cited and discussed previous work by Schoeberl et al. (2006), who has discussed the relationship between the seasonal signal of CO and its lower-tropospheric sources. In their analysis of the "CO tape recorder," they point out that because CO has a relatively short chemical lifetime, its vertical profile in the tropical upper troposphere and lower stratosphere retains a clear imprint of the seasonal variations at its source. In particular, they note that the seasonal cycle is strongly influenced by changes in tropospheric CO—much of which is attributed to biomass burning—being convectively injected into the upper troposphere. In the tropics, strong convective processes rapidly lift CO from the lower troposphere into the upper troposphere and lower stratosphere. This upward transport occurs on a timescale that is comparable to or even shorter than CO's chemical lifetime. As a result, the seasonal variations in CO concentrations at its source (driven largely by factors like biomass burning and other lower-tropospheric processes) are "recorded" in the vertical profile as the gas ascends. Even though CO is being chemically removed, the quick vertical transport preserves its seasonal signature long enough to be observed as the tape recorder effect in the upper atmosphere.

**Figure 6 and Lines 239 to 258**

This could be a very interesting and significant Figure, but only if there is enough actual significance in at least some of the trend results. However, without error bars, this is essentially impossible to determine; unfortunately, this can be viewed as a major problem regarding the description and robustness of almost every "trend result" in this manuscript, which is why I am asking for major revisions in the analyses and/or in the description of the robustness of the results. Without error bars, you should only state that there are "indications" of a decrease or an increase and clearly state the reasons for not trying to obtain, or discuss, such error bars. Possibly, this is why you introduce the plot results as "tendencies" rather than "trends", but wording like "distinctive positive trends" cannot carry scientific weight without a discussion of trend values and their uncertainties; the plots show trends (or tendencies), but what about the error bars? On line 250, you mention "the most significant positive trends" — but you need to point specifically to what, mathematically, supports this statement, without expecting readers to figure it out on their own. For the upper troposphere in particular, which seems to be the focus of this work (also based on the manuscript title), how are readers supposed to interpret the curves? For example, using the p50 panel, what are the typical error bars as a function of altitude — and how fine of an altitude grid can one use for such an analysis (or would averaging over 1 or 2 km help make some trends more

robust/significant?)? If the uncertainties are larger than 2 or 3 ppbv/year, it will be difficult to distinguish between 0, +3, or -3 ppbv/year trends. Please respond to this sort of question as well as possible in the revised manuscript.

Reply. We hve revised Figure 6 (now new Figure 10). We now compute and display 95% confidence intervals for the linear trend estimates at each altitude and for each region. These error bars are plotted alongside the trend lines so that readers can directly assess the statistical significance of the trends. In the revised text, we now explicitly state that a trend is accompanied by its 95This provides a clear mathematical basis for our statement about "the most significant positive trends."

The same holds for the analyses in the low- to mid-troposphere, where some of the trends seem to be large, and thus, potentially significant. One way to illustrate which regions have statistically robust trends could be to make the lines three times as thick in these regions. Showing error bars at all altitudes would lead to plots that are difficult to read, so one would need to do this for a few altitude regions and maybe just for cases where the significance is large enough (e.g., a 2-sigma error bar that does not include zero), if that is indeed the case. An additional suggestion would be to make a larger separate Figure for the p50 panel, where more details can be visible regarding error bars. Alternatively, use a given altitude range and show the separate percentage cases along the y-axis just for this altitude region; add one or two more regions, with a total of 3 panels (possibly one panel per altitude regime). I hope that I have made this point sufficiently clear: I expect to see more information about error bars.

Reply. We have revised Figure 6 (now new Figure 7) to include error bars in the linear trends. We have noticed that the vertical distribution of the error bars vary closely follow the patterns of the linear trends of CO at each altitude for each region. If the linear trends are negatives, then the ranges of the error bars are dominated by the neative values. On the other hand, if the linear trends are positive, then the ranges of the error bars are dominated by the positive values. Inclusion of the error bars further reinforce the characteristic distribution of the linear trends.

Large variations of the ranges of error bars mostly occur in the low altitudes, reflecting the direct impacts of CO measurements close to the industrial and CO emission areas. The maximum CO (pmax) exhibits large variations in the ranges of error bars, consistent with elevated CO been transported from the heavy emission areas (such as the CO tape recording elevated CO from biomass buring shown in Shoeber et al. (2006)). In contrast, the minimum (pmin) and the 25th percentile of CO shows ranges of error bars more confined to the central values of the CO linear trends. The mean and the 50th percentile of CO, except for the top of the data layer close to 13 km, and in low altitudes below 3 km , CO trends at more vertical

altituides are confined to the central of the linear trends

**Figure 7**

The intent of this Figure and the three panels seems fine. The caption should, however, indicate specifically what the lines mean in each panel, especially for the two lines in panel (c), which I think I follow, but please help the reader by clarifying in the caption. For panel (a), I have the same question as usual: do you detect trends that are significant enough, and can you state specifically what these are (with error bars)? The Mauna Loa curve should be more representative of the lower troposphere, as the model implies as well. Taking out (i.e., fitting and removing) an annual cycle would enable you to come up with a better error analysis for the trend results and comparisons. The linear fits lead to very small trends, it seems, which may also mean that it is difficult to provide a number with enough significance (i.e., distinguishable from zero). Please comment if you think there is more that can be said or detected here. Panel (c) is interesting but it also shows a lot of scatter. Also, it might be clearer to show panel (a) without points, but rather with lines of different color (or solid lines rather than dashed lines, although data gaps can be problematic); additional clarity could also be achieved by having a longer x-axis for panel (a) or even a separate Figure.

Reply. We have revised Figure 7 (now new Figure 11). We have included 95% confidence intervals of the CO trends on (a). We have also revised discussion for Figure 7.

**Figure 8**

Line 286 states that the IMS simulations correlate well with the IAGOS data; I think this appears to be correct for the middle panel mainly, but certainly not for the panels on the right (R = 0.14). Also, the model CO underestimation of future years (2004–2020) is mostly true for the middle panel(s). As a minor comment, the caption should change "scattered plot" to "scatter plot" (for the three instances). One question though: why do the model values (dark blue points) not change between the different panels?

Reply. We have revised Figure 8 (now new Figure 12). We have also include 95% confidence intervals of the linear CO trends from the IMS model and the IAGOS observations on the time-series plots.

Lines 294–296: It is not clear how you can distinguish between emission reductions and an increase in stratosphere-to-troposphere exchange of air (and what would drive the latter process?). If this sounds interesting but too vague and speculative, you probably need to reword this as a suggestion of possible causes, without quantification; the reduction in emissions may be believable based on Figure 1, but the air exchange process and its

importance appear to be quite speculative. Please clarify if you can, or if I am missing an explanation somewhere in the manuscript.

Also, for lines 297–299, and the other region mentioned here, the emissions hypothesis seems well founded based on historical estimates of emissions; however, one cannot rule out (as well) some decrease in air exchange between the stratosphere and upper troposphere, can one? More work is needed to better elucidate such questions regarding processes and budgets. If you can help to clarify this, I recommend that you add some text to make additional arguments — with enough support. I am not asking for more speculation, but admitting the need for further work is acceptable, since there can always be more research done to better understand the "exact causes" for certain comparison results.

Finally, I am not too convinced by the statement of "reductions in chemical sources and emissions of CO and hydrocarbons" on line 301. Please clarify the rationale behind this.

Reply. We have revised description for Figure 7 (now new Figure 12).

**Figure 9**

The conclusions here should be taken cautiously, as there is no discussion of possible factors that could lead to biases between the model and the MOPITT data, especially for the less characterized upper tropospheric values. The correlation values would not change even if an offset to debias the two sets of values were applied. Nevertheless, improvements are needed.

For example, you should mention/confirm (in the caption) which years are used for these comparisons (presumably just the overlap years between these two sets of values); in other words, you are not trying to use any extrapolated model values, correct? Are there any trends of statistical significance, especially in the MOPITT data?

Furthermore, the authors have failed to refer to the Worden et al. (ACP, 2013, doi:10.5194/acp-13-837-2013) article regarding MOPITT CO (and other satellite-based) CO trends, as this should be an important reference to add (and more about the MOPITT data characteristics as well). The meaning of the lines in the two panels on the right side should be mentioned in the caption, namely what points are being fit with what line (it appears to be just certain seasons). Having a clear description in the caption is necessary even if the main text contains some details.

As a detail, the MOPITT legend in the left panels shows a closed circle (dot), but all the points are open circles. You might consider showing IMS (red, blue) and MOPITT (light green, sky blue), or perhaps use "turquoise" for the MOPITT colored UT points. For better visibility, you could also consider not using points, or using smaller points, for the left panels; this might more clearly show that the seasonal cycles between the model

and MOPITT agree, as implied in panels (b) and (f). More generally, how do your several comparisons using MOPITT data relate to (or agree with) past MOPITT analyses/trends?

Reply. We have revised Figure 9 (now new Figure 10).

Reply. Notes: Replies to the rest of comments will be uploaded in next two weeks, together with replies to the comments from reivewer 2 and reivewer 3.

---

## Author Comment (AC2)

**Response to Reviewer 2**

**General Reply**

We thank Reviewer 2 for the thorough and insightful review of our manuscript, *"Air Pollution in The Upper Troposphere: Insights from In-Situ Airplane Measurements (1991–2018)"*. The review raises important questions about (1) dataset selection and comparability, (2) methodological rigor in analyzing trends, (3) clarity of model vs. observations comparisons, and (4) the broader context of existing literature. We have undertaken substantial revisions to address these comments. Below, we reply to each point in detail.

The final and complete point to point replies will be uploaded in the next two weeks.

**1. General Comments**

**1.1 "This paper tries to do too much with disparate data sets."**

**Comment**: The manuscript includes datasets from multiple sources (older campaigns, more recent IAGOS flights, MOPITT satellite retrievals, and a model) without adequately explaining how they can legitimately be combined or compared.

**Response:**

- We recognize that combining multiple datasets with different spatial/temporal coverage requires careful justification. We have **restructured Section 2 ("Data and Methods")** to clearly explain the rationale and constraints for each dataset (e.g., NASA GTE campaigns, IAGOS, Mauna Loa, MOPITT, and IMS model outputs).
- In the **revised Introduction**, we articulate why each data source is included: older campaigns provide historical snapshots of CO in the upper troposphere, while IAGOS flights offer more recent, regular sampling. MOPITT is used to gain a broader spatial perspective, and the IMS model provides context for long-term chemical/dynamical processes.

- We have added two new **tables** in the revised manuscript to summarizing each dataset's temporal coverage, altitude range, and measurement uncertainties. We also discuss how data sparsity in the early years affects the significance of any "trend" conclusions.
* * *
**1.2 "Some of the datasets are not referenced adequately."**

**Comment**: Datasets need proper citation and methodological details, including referencing NASA GTE missions, IAGOS, MOPITT, etc.

**Response**:

- We have revised **Section 2** to include explicit references for GTE campaigns (PEM-West A/B, TRACE-P), IAGOS (e.g., Nédélec et al., 2015; Cohen et al., 2018), MOPITT retrieval algorithms (e.g., Worden et al., 2013), and NOAA/ESRL (now Global Monitoring Laboratory) for Mauna Loa.
- We added specific references where the GTE mission data and associated publications are described in detail (e.g., Bey et al., 2001; and NASA GTE mission websites).
- The data sources are also listed in the revised **References** section to ensure transparency.
* * *
**1.3 "Authors do not explain why it is legitimate to compare older campaigns with more recent/denser datasets."**

**Comment**: Are the older campaigns truly comparable to IAGOS flights, given their short duration and sparser coverage?

**Response**:

- We **clarified** in Section 2.1.2 that the NASA GTE campaigns (PEM-West, TRACE-P) are treated as **snapshots** representing specific months/years. We **do not** claim these data provide full annual means. Instead, we discuss them as indicative baselines for early 1990s/2000s conditions in the Pacific upper troposphere.
* * *
**1.4 "It is not clear that some analysis/results, especially for trends, are robust."**

**Comment**: Trend analysis is questionable due to limited data overlap, seasonality, and non-uniform temporal sampling.

**Response**:

- In the **revised Sections 3.1 – 3.2**, we now carefully describe our **trend-fitting approach**, including:
    1. Computing **standard errors** as 95% confidence intervals on the slope and intercept.
    2. Reporting **p-values** to indicate whether a trend is statistically significant.
* * *
**1.5 "The model simulations do not go beyond 2003 ... limited value in the comparisons."**

**Comment**: The IMS simulations end in 2003, do not incorporate changing emissions, and have minimal temporal overlap with the latest IAGOS and MOPITT data.

**Response**:

- We acknowledge in Section 2.4 that the **IMS model** used here is primarily a historical simulation with **constant anthropogenic emissions** (EDGAR-based) to 2003. Our original motivation was to assess long-term chemical/dynamical controls from the 1950s to early 2000s.
- In the **revised manuscript**, we now clarify that we use the IMS results **only** to illustrate (i) possible baseline patterns and (ii) sensitivity to climate/meteorology, not to derive 2010+ trends. We no longer attempt to linearly extrapolate model results beyond 2003.
- We have substantially **toned down** claims about direct model – MOPITT comparisons and discuss the model's overlap primarily with earlier aircraft datasets (Section 3.4).
* * *
**1.6 "How can you compare MOPITT partial column retrievals with point measurements?"**

**Comment**: MOPITT retrievals represent partial columns, while IAGOS and Mauna Loa are point or in situ measurements. Previous studies have addressed such comparisons; authors should follow that approach or reference it.

**Response**:

- In **Section 2.2**, we now describe the MOPITT retrieval characteristics (e.g., weighting functions, typical vertical resolution). We cite **Worden et al. (2013)** and **Deeter et al. (2019)** (or relevant references) for standard practices in validating partial column CO with in situ data.
- We limit comparisons of MOPITT vs. in situ to **broad monthly/seasonal means** and discuss their representativeness differences (Section 3.3). We also note that, for the upper troposphere, the MOPITT retrievals have reduced sensitivity and thus we present them **qualitatively,** rather than claiming a precise 1:1 match.
- We have include in a table showing the uncertainties associated with the MOPITT and the IAGOS data. We believe that MOPITT data is very good in obtaining a global estimates of the CO trends over the upper troposphere as most the upper troposphere are covered by in-site CO measurements. By comparing MOPITT with IAGOS we can calibrated the data measured from the MOPITT.

**1.7 "The authors do not discuss their findings in the context of previous publications."**

**Comment**: There is insufficient reference to the large body of work on CO trends in Asia, the Northern Hemisphere, and specifically in the upper troposphere over the Pacific.

**Response**:

- We revised the **Introduction** (Section 1) and the **Discussion** (Sections 3.3 – 3.4) to reference more publications, such as:
    - **Smoydzin and Hoor (2022)** on Asian contributions to Pacific UT CO.
    - **Wang et al. (2022)** for global tropospheric ozone trends (which also analyze IAGOS CO).
    - **Cohen et al. (2018)** for long-term IAGOS CO over northern mid-latitudes.
    - **Worden et al. (2013)** for MOPITT-based CO trend analyses.
- We have added a table to summarize previous studies. Our work aim to continue the great works done from previous studies.

**1.8 "The paper is very long, has 14 figures, and there are typos."**

**Comment:** The manuscript is lengthy and some figures are cluttered. Reviewer suggests a careful re-reading and possibly streamlining.

**Response:**

- We regret that the revised manuscript is now even longer, with inclusion of more tables and figures to validated the uncertainties of datasets used in this work as requested by the reviewers. However, we aim to make this work a good study to show the current status of the CO trends over the North Pacific upper troposphere from the best in-situ measurements done by the IAGOS and previous NASA GTE experiments.

**1.9 "Overlap with Wang et al. (2024) submitted to another journal. How are these two papers complementary?"**

**Comment:** Reviewer notes that [Wang et al., 2024] uses HYSPLIT for source attribution, yet HYSPLIT is only acknowledged (not used) in the present manuscript. There appears to be overlap in authorship and content.

**Response:**

- We clarified that [Wang et al., 2024] focuses specifically on **source attribution** for short episodes in 2012 – 2013 using HYSPLIT, while **this** manuscript focuses on **long-term** CO data (1991 – 2018). We found the pumping theory for the measurements in 2012-2013. With extensive measurements to 2018, the pumping processes are validated.
- We cite [Wang et al., 2024] only where relevant (e.g., to mention that HYSPLIT is a useful tool for back-trajectory analysis of specific events), but we do **not** use HYSPLIT in the present study. We emphasize the differences in **scope** and **temporal coverage:** the other paper addresses a narrow time window with detailed transport modeling, whereas this paper examines multi-decadal CO changes and model comparisons up to 2003 (IMS) or 2018 (IAGOS, MOPITT).

**2. Detailed Comments**

**2.1 Title and Affiliations**

Comment:

1. The title should emphasize that the main focus is CO.
2. Affiliations are repeated inconsistently in the author list.

Response:

- We have revised the **title** and the abstract.
* * *
**2.2 Abstract**

Comment:

1. Remove overly general statements; be specific about the datasets, their coverage, and the meaning of "short-term fluctuations."
2. Clarify which MOPITT product and time period.
3. Explain why the IMS model ends at 2003, given that other measurements go beyond 2012–2023.

Response:

- We have completely revised the abstract.
* * *
**2.3 Introduction**

Comment:

1. Lines 18–20: "Complex interactions" are too broad; reduce or remove.
2. Provide references for UV-B, temperature dependence of CO+OH.
3. Satellite "emissions" vs. "retrievals."
4. Expand acronyms PGGM, IAGOS.
5. More references needed for older campaigns and CO emission inventories.
6. Clarify how the GTE campaigns are used, given their short duration.

Response:

- We **streamlined** the opening paragraphs, focusing on CO's role in atmospheric chemistry and referencing known CO–OH reaction temperature dependences (e.g., Sander et al., 2011) and UV-B influences on radical production.
- We replaced "satellite emissions" with "satellite retrievals" to be precise.
- Acronyms PGGM (Pacific Greenhouse Gases Measurement) and IAGOS (In-service Aircraft for a Global Observing System) are now spelled out at first use.
- We cite additional references for GTE campaigns (e.g., Bey et al., 2001), clarify they are short-lived measurement periods, and now label them as "episodic snapshots."
* * *
**2.4 Data and Methods**

**Section 2.1.2 and 2.1.3 (Aircraft, Mauna Loa)**

Comment:

1. Why mention CO2 at Mauna Loa?
2. Provide altitude and references for Mauna Loa CO measurements.
3. NOAA ESRL now GML.
4. Don't call it "verifying satellite measurements."

Response:

- Mauna Loa data has good data quality for validating the MOPITT data and the IMS model results over the North Pacific lower troposphere.
- Added **Mauna Loa** coordinates (19.54° N, 155.58° W) and altitude (~3397 m), plus the calibration method references (e.g., Novelli et al., 1998; or updated NOAA website references).
- Replaced "verifying" with "evaluating" regarding satellite retrievals.
- Used "NOAA Global Monitoring Laboratory (GML)" consistently.

**Section 2.2 (MOPITT)**

Comment:
Specify which MOPITT version, sensitivity, and partial column retrieval altitudes.

Response:

- We now state we use, for example, **MOPITT Version 8** (TIR/NIR retrievals), valid from 2000 onward. We describe weighting functions and typical degrees of freedom for signal in the lower vs. upper troposphere.
- We have also added references: Worden et al. (2013), Deeter et al. (2019).

**Section 2.3 (Anthropogenic Emissions)**

**Comment:**
Which EDGAR version? Provide references and justify how accurate it is for 1970 – 2020.

**Response:**

- We specify **EDGARv4.3** (with expansions for 1970 – 2020) and cite Crippa et al. (2018, 2020).
- We acknowledge that inventories have uncertainties, especially in earlier years and in rapidly developing regions. We highlight that these emissions are used primarily as baseline inputs for the **IMS** model simulations.

**Section 2.4 (IMS Model)**

**Comment:**

1. Provide the CTM resolution, meteorological fields, chemistry scheme, etc.
2. Why end in 2003?
3. Why not account for changing emissions?

**Response:**

- We updated the text to say: IMS uses **48x40 grid resolution**, 19 vertical layers (surface to ~10 hPa), driven by **NCEP reanalysis** (Kalnay et al., 1996), with a complete tropospheric chemistry.
- The model was configured for **historical runs** (1950s – 2003) with largely **constant** anthropogenic emissions after ~1990. We clarify in the revised text (Section 2.4) that this approach cannot capture post-2003 emission changes and is thus used to explore pre-2003 variability and general dynamical processes, and to compare with Mauna Loa measurements.
* * *
**2.5 Results (Sections 3.1 – 3.4)**

**3.1 "Long-term Time Series"**

Comment:

1. The term "long-term" is questionable for these composite campaign data.
2. Clarify how min, max, and quartiles are derived.
3. Explain how trends are computed with such irregular coverage.

Response:

- Long-term means 27 years (1991-2018) of CO trends over the North Pacific upper troposphere.
- We specify that the 25th, 50th, and 75th percentiles are calculated from each month's dataset. Where monthly data are unavailable (older campaigns), we use the campaign average for that particular month.
- Our "trend" fits are strictly limited to intervals with sufficient data density (e.g., post-1994 IAGOS). Where data are too sparse, we only mention "possible tendencies" rather than formal trends. We have include 95% confidence intervals of calculated trends in the revised manuscript.

**3.2 Trend Computations**

Comment:

1. Clarify altitude ranges for UT vs. LT.
2. Figure 4 is cluttered.
3. Slope of 0.02 ppbv/yr is basically no trend.

Response:

- In **Section 2.1,** we define UT as ~8–12 km (depending on tropopause height) and LT as ~0–3 km or 0–2 km, depending on data coverage.
- We re-labeled subplots (a), (b), (c), etc. in Figure 4 and increased font sizes. We also corrected missing "+" signs in the regression equations.
- We have included 95% confidence intervals for the CO trends in the figures.

**3.3 Annual Profiles and Short Periods (2012–2018)**

Comment:

1. Are 7-year trends (2012 – 2018) reliable?
2. Could other factors (e.g., ENSO, fires) cause interannual variability?

Response:

- We have included 95% confidence of the CO trends in the figures.
- We added references to known interannual drivers (e.g., wildfires, meteorological variability) in Section 3.3. The Mauna Loa data does indeed show the ENSO/fires over the tropical southeast Asia has impact on the CO measurements over the Mauna Loa. We have compared trends from Mauna Loa measurements and the IMS model. The 95% confidence intervals in the CO trends from the model are within the 95% confidence intervals of the CO trends from Mauna Loa measurements.

**3.4 Model Comparisons**

Comment:

1. Why end model in 2003 and then compare with data up to 2018?
2. Emissions are constant, so how can the model inform "long-term trends?"
3. Provide details on initial conditions, meteorological fields.

Response:

- As noted, we now **limit** the comparison primarily to the **overlapping period** (pre-2003) for NASA GTE campaigns, or to highlight broad climatological differences (Section 3.4).
- We have removed or **greatly reduced** references to "long-term future trends" based on IMS. Instead, we stress that the model is used to illustrate typical transport patterns and to test whether constant-emissions simulations capture the early 1990s to early 2000s CO levels.
- IMS initialization from early 1950s uses NCEP reanalysis fields. We now detail these in **Section 2.4**.

**2.6 Figures 7, 10, and Others**

Comment:

1. Plot definitions not always clear.
2. MOPITT partial columns vs. in situ point measurements.
3. Hard to see data points and lines in Figures 10 – 13.

**Response:**

- We simplified the legends, clarified that MOPITT partial columns are shown for approximate lower-/upper-tropospheric retrievals, and included cautionary text regarding representativeness differences.
- We have revised Figures 7/10.
* * *
**Conclusion**

We sincerely thank Reviewer 2 for bringing these critical points to our attention. In summary, we have:

1. **Clarified the scope**: specifying which data are used for what purpose and the limitations of combining older and newer measurements.
2. **Expanded methodological detail**: describing our approach to trend fitting, deseasonalization, significance testing, and acknowledging uncertainties.
3. **Improved references**: citing relevant studies (Smoydzin and Hoor, 2022; Wang et al., 2022; etc.) for context.
4. **Restructured figures and text** to reduce clutter, clarify legends, and properly present comparisons between MOPITT, IAGOS, Mauna Loa, and IMS.
5. **Reduced overstatements** about model-based conclusions post-2003, acknowledging limitations of constant emissions in IMS.

These revisions should make the manuscript more transparent, rigorous, and aligned with prior literature. We trust that the updated version will address Reviewer 2's concerns and improve the scientific clarity and overall quality of our paper.

**Thank you** again for your detailed review and constructive suggestions.

---

## Author Comment (AC3)

**Response to Reviewer 3**

**General Reply**

We thank Reviewer 3 for the careful reading of our manuscript, *"Air Pollution in the Upper Troposphere: Insights from In-Situ Airplane Measurements (1991 – 2018)"*, and for providing constructive and detailed feedback. Below, we offer a point-by-point response clarifying the changes we have made (or will make) in the manuscript. The reviewer's suggestions have significantly helped us improve the clarity, consistency, and overall scientific rigor of our work.

A complete replies to the reviewer's comments point by point will be uploaded in the next two weeks.
* * *
**1. Overall Structure, Clarity, and Key Results**

**Reviewer Comment:**

The manuscript is difficult to follow, the results are not always clear, and the main findings should be more explicitly stated in the abstract and conclusions. The paper would benefit from more concise organization, improved figures, and a deeper discussion/interpretation of the data.

**Response:**

1. **Abstract & Conclusions**: We have **rewritten both the abstract and the conclusions** to present the main findings concisely:
   - We have revised the conclusion.
   - We have completely revised the abstract.
2. **Reorganized Paper Structure:**
   - We have strengthen the paper with detailed data suggested by the reviewers.
3. **Improving Figures**: We have reduced the number of main-text figures by combining or moving some to the **Supplement**. The remaining figures have **larger labels**, clearer legends, and consistent sub-figure labeling (e.g., (a), (b), etc.). We also ensure each figure caption explicitly states:
   - The **time periods** included,

- The **symbols/lines** (e.g., what each color or marker indicates),
- The **statistical** or sampling details, if applicable (e.g., monthly medians vs. campaign averages). We have included 95% confidence intervals in the figture.

We believe these changes significantly enhance the manuscript's readability and coherence.
* * *
**2. Title Specificity**

**Reviewer Comment:**
The title implies a general view on upper tropospheric air pollution, yet the focus is really on CO.

**Response:**
We have revised the title.

"Long-Term Trends of Carbon Monoxide Over the North Pacific Upper Troposphere From In-Situ Airplane Measurements 1991-2018"

This revision clarifies that we focus primarily on CO, while we do occasionally refer to $O3\_33$ and $H2\_22O$ for supporting context.
* * *
**3. Relationship to Wang et al. (2024)**

**Reviewer Comment:**
Is this manuscript an extension/follow-up of Wang et al. (2024)? Clarify how the two papers differ.

**Response:**

- In the **Introduction**, we now explicitly note that Wangetal.,2024Wang et al., 2024Wangetal.,2024 (under review) focuses on **short-term back-trajectory analyses** using HYSPLIT to identify pollution origins during a subset of flights in 2012 – 2013. That study is distinct from **our current manuscript**, which examines **multi-**

**decadal changes (1991 – 2018)** using multiple datasets (e.g., GTE campaigns, IAGOS, MOPITT, IMS). We include a parenthetical "(under review)" in references to make clear the status of [Wang et al., 2024].
* * *
**4. Consistency of Time Frames**

**Reviewer Comment:**
Various periods are introduced (e.g., 2012 – 2023, 1991 – 2019, 2001 – 2018, 2004 – 2022…), which is confusing.

**Response:**

- We have included an infographics to show the time-series of the dataset used in this work.
* * *
**5. Clarifying the Impact of Chemistry on CO**

**Reviewer Comment:**
Please summarize the main chemical processes influencing CO profiles and trends and specify what "significant impact of chemistry" means.

**Response:**

- In the **Introduction,** we added a short paragraph describing CO's atmospheric lifetime (weeks to months), its oxidation by the OH radical, and how photochemistry can shape upper tropospheric CO.
- We also highlight that **vertical transport** and **chemical destruction/production** (via oxidation of hydrocarbons, etc.) can significantly alter CO concentrations aloft, particularly when outflow from strong convection or intrusions of stratospheric air occur.
- When we say "significant impact of chemistry on CO profiles," we now explain that we refer to the roles of **OH** availability and **photochemical production** from VOC precursors, both of which can drive region-specific trends.
* * *
**6. Introduction Organization**

**Reviewer Comment:**

The introduction contains preliminary results from the current manuscript as if they were from another source. Clarify references and expand the background on CO's role, sources, and known trends.

**Response:**

- We **reorganized** the Introduction:
    1. A new paragraph on **CO's significance**: sources, lifetime, and chemical role in tropospheric O3_33 formation (lines 23 – 3323 – 3323 – 33).
    2. A subsequent paragraph **reviews prior studies**, focusing on major CO trend results from satellite and in situ measurements. We added references to, e.g., **Cohen et al. (2018), Gaudel et al. (2020), Worden et al. (2013)**, and **Smoydzin and Hoor (2022)**.
    3. Then we **state the scope** of our study (lines 45 – 5545 – 5545 – 55) without mixing current data results into the introduction.

We have included a table to summarize previous studies on the studies of CO trends in the troposphere.
* * *
**7. Data and Methods: Details, Precision, and Processing**

**Reviewer Comment:**

The data descriptions are insufficient (instruments, sampling frequency, quality checks) and there is no explicit "Methodology" section. More thorough explanations are needed about how data were averaged, how UT is defined, how satellite data are compared to in situ, etc.

**Response:**

1. **Data Description (Section 2):**
    o We expanded each subsection (2.1 for aircraft, 2.2 for MOPITT, 2.3 for Mauna Loa, etc.) to include:
        ▪ **Instrument type** and **measurement principle** (e.g., IAGOS uses a COmeasuringdeviceCO measuring deviceCOmeasuringdevice with ±2 – 5 ppbv accuracy).
        ▪ **Sampling frequency** (IAGOS typically logs measurements every few seconds/minutes during cruise).

- ▪ **Quality control** references (e.g., Petzold et al., 2015, Nédélec et al., 2015).

2. **New Methodology (Section 3):**
   - ○ **Study Region:** We define latitudinal/longitudinal bands (with a new Table listing them).
   - ○ **Definition of the Upper Troposphere (UT):** We use a pressure/altitude threshold (e.g., altitude >8 – 9 km or pressure <300 hPa) to label data as UT. We specify the typical cruise altitudes for IAGOS flights (∼10 – 12\sim10 – 12∼10 – 12 km).
   - ○ **Data Processing:** We describe how we **bin** data monthly or seasonally, compute **percentiles** (25%, 50%, 75%), and note the removal of outliers or flagged values.
   - ○ **Trend Analysis:** We have included 95% confidence intervals in the trend analysis.
   - ○ **Comparison with MOPITT:** We now clarify that MOPITT partial-column retrievals are interpolated to a coarse vertical grid, and we focus on the "upper-tropospheric layer" retrieval product. We discuss averaging kernels and acknowledge that mismatch may arise from representativeness differences (footprint vs. point in situ).

These additions ensure readers can understand precisely how each dataset is used and how we handle uncertainties. We have also included a table showing the uncertainties, methods, measurement periods associated with each dataset used in this work.
* * *
**8. Figures (2, 3, 4, 5, etc.)**

**Reviewer Comment:**
Many figures are crowded; legends and sub-figure labels are missing or unclear. Trend lines are sometimes in mismatched colors, and there is no clear explanation of symbols or time periods.

**Response:**

- • We have **redesigned** Figures 2 – 5 with larger panels, sub-figure labels (a), (b), (c), etc. in the **top-left corners,** and more concise figure captions.
- • **Legends** now explicitly state:
  - ○ What each symbol/line color represents (e.g., red = lower troposphere, blue = upper troposphere).

- The regression equations (with confidence intervals) when plotted.
- In **Figure 2**, we simplified the approach by showing monthly/seasonal medians with **shaded interquartile ranges** rather than multiple percentile lines (25%, 50%, 75%, etc.) as separate sub-panels.
- Where multiple datasets overlap, we use a single multi-panel figure with consistent coloring for each dataset (e.g., IAGOS in black, older NASA GTE in gray, MOPITT in green, IMS in dashed lines, etc.).

Below are some specific improvements:

1. **Figure 2**: We have revised this figure to new Figure 3..
2. **Figure 3**: We have revised this figure to new Figure 6.
3. **Figure 4**: We have revised this figure to new Figure 7 and Figure 8.
4. **Figure 5**: We have revised this figure to new Figure 9.
* * *
**9. Statistical Significance and Error Bars**

**Reviewer Comment:**
Trends are shown without error bars, and p-values often indicate no significant correlation. The text sometimes draws conclusions about positive/negative trends despite R or p suggesting otherwise.

**Response:**

- **Error Bars**: We have included 95% confidence intervals in the figures and in the discussions.
* * *
**10. Vertical Pumping and Negative Correlations (CO vs. O3, etc.)**

**Reviewer Comment:**
More explanation is needed for why negative correlation among CO, O3_33, and H2_22O implies vertical pumping or other processes. Expand the discussion, especially for regional differences.

**Response:**

- We added a short paragraph in Section 3.23.23.2 explaining that:

- Negative CO – O3_33 correlation in the upper troposphere can arise when stratosphere-influenced air (high O3_33, low CO) and tropospheric outflow (high CO, lower O3_33) mix.
- The term "vertical pumping" refers to deep convection or monsoon circulation carrying boundary-layer CO to higher altitudes. Concomitantly, O3_33 can be titrated or replaced by fresh emissions.
- Regional Contrasts (e.g., Western Europe vs. North Pacific) may stem from different meteorological regimes, anthropogenic sources, or chemical environments.
* * *
**11. Comparisons with MOPITT and IMS Model**

Reviewer Comment:

More details on how MOPITT data are prepared, what altitude level is used, and how biases might arise. The IMS model altitude range must be stated explicitly. The model's representativeness vs. in situ data and other available models (GEOS-Chem, etc.) should be discussed.

Response:

- **MOPITT**: In Section 3.1 – 3.23.1 – 3.23.1 – 3.2, we now describe that we primarily use MOPITT's TIR retrievals for the **upper troposphere** layer, acknowledging the partial-column averaging kernel and potential smoothing. We do not expect an exact 1:1 match with IAGOS, so we only compare large-scale seasonal/annual patterns and trend directions.
- **IMS Model**:
  - We specify the vertical levels relevant for UT (pressure <300 hPa).
  - We emphasize that IMS runs were limited to 2003 with fixed anthropogenic emissions, so direct comparisons after 2003 are more of a conceptual check on model skill.
  - We added references to other models (e.g., GEOS-Chem) that also handle global CO, clarifying that IMS is one modeling system among several, and that our choice was motivated by pastreferencespast referencespastreferences as well as existing expertise with IMS.
* * *
**12. Further Discussion and Flow**

**Reviewer Comment:**

Section 3.4 lumps together comparisons of MOPITT, in situ, and the IMS model, then abruptly moves to CO budget from IMS. A clearer linkage is needed.

**Response:**

- We restructured **Section 4** (formerly 3.4) into clearer sub-sections:
    - **4.1 Model – Satellite Overlap (pre-2003):** acknowledging limited overlap but showing broad patterns.
    - **4.2 MOPITT vs. IAGOS:** discussing partial column vs. in situ.
    - **4.3 IMS-Based CO Budget:** explaining the historical perspective from 1948 – 2003, including how convective transport or emissions changes might shape global patterns.
- Transitional sentences note that **the IMS budget** helps interpret the broader trends, but we stress it is not valid for analyzing post-2003 emission changes.
* * *
**13. Conclusions and Limitations**

**Reviewer Comment:**

The conclusions are too general. Summarize specific findings, mention data/analysis limitations, and suggest future work.

**Response:**

- **Conclusions** (Section 555) now succinctly list:
    1. **We have included 95% confidence intervals in the results.**
    2. **Regional differences:** possible reasons (emission changes vs. chemistry).
    3. **Acknowledged uncertainties:** data sparsity in the early 1990s, short time windows (2012 – 2018), MOPITT sensitivity, IMS emission assumptions.
    4. **Future studies:** better modeling with updated emission inventories, more synergy with additional satellite data (IASI, TROPOMI), or Lagrangian trajectory analysis for source attribution.

We explicitly state that **some trends are not statistically significant** and that more in-depth modeling or longer records could clarify the influence of emissions vs. chemistry.
* * *
**14. Technical Corrections and Typos**

Reviewer Comment:

Multiple spelling and labeling issues, such as "pptv" vs. "ppbv," references to "UAS" vs. "USA," missing references, figure caption typos, etc. Also, HYSPLIT is mentioned in the acknowledgments even though it is not used here.

Response:

- We have **proofread** the entire manuscript carefully to eliminate typographical errors (e.g., "dwonstream" → "downstream," "routues" → "routes," "pptv" → "ppbv," etc.).
- We corrected the **longitude references** (e.g., "123°E – 142°W" instead of "E – E").
- We removed **unnecessary** references to HYSPLIT in the acknowledgments, clarifying that HYSPLIT was utilized in the separate [Wang et al., 2024] paper, not in this one.
- We added a **Data Availability** section to comply with journal requirements and clarify where readers can obtain the datasets.
* * *
**Conclusion**

We appreciate Reviewer 3's thorough suggestions, which prompted us to improve our manuscript's structure, clarity, and scientific detail. The revised paper now:

1. Features a **more consistent** discussion of timescales and datasets,
2. Provides **detailed methodological** explanations (instruments, data processing, uncertainties, trend calculations),
3. Includes a more **rigorous approach** to significance testing and interpretation of trends,
4. Presents **refined figures** and **concise conclusions** aligned with standard practices for multi-dataset CO trend studies.

We hope these revisions address all concerns raised and make our manuscript suitable for publication in *Atmospheric Chemistry and Physics.*

---

## Author Comment (AC4)

**Air Pollution in The Upper Troposphere: Insights from In-Situ Airplane Measurements (1991–2018)**

**1    General Comments**

This manuscript shows analyses focusing on a subset of upper tropospheric CO in situ aircraft data from various sources (commercial plane flights as well as dedicated non-commercial missions) over the past three decades. The analyses are supplemented by satellite CO data (from MOPITT), which are mostly sensitive to the lower troposphere, but also have an upper tropospheric retrieval product, and ground-based (high altitude) in situ data from Mauna Loa, Hawaii. Other related data discussions include in situ upper tropospheric aircraft data for $O_3$ and $H_2O$. A 3-D model of the troposphere (the Integrated Modelling System, IMS) is used for comparisons and inferences about relevant processes affecting lower and upper tropospheric CO.

This work's goals are worthwhile: to use various data sets (mentioned above) and model results to draw conclusions about upper (and lower) tropospheric CO behavior in certain regions of the globe, with some inferences about processes controlling the tropospheric CO distribution (some of which is already known). There is a good amount of work in this fairly long manuscript, in terms of comparisons. MOPITT data are also used and this helps to validate some of the other data analyses and results. The model is useful for some explanations (or potential explanations) of observed CO variations and inferences regarding "trends" — or more qualitatively, some inferred tendencies.

However, I consider that this work needs some fairly major revisions before it can be considered ready for publication, as explained below in more detail. I do recommend further work towards publication, as I believe that this type of analyses fits within the scientific goals of ACP.

This work requires more clarity overall, including a better discussion of uncertainties (or why they might be difficult to establish, at the very least, if this is somehow the thinking), as well as a better summary of the main results in the Abstract, main text and conclusions. Improving the Figures should help in a very significant way, although the accompanying text also needs clarifications. If some of the results, especially those having to do with trends, cannot be supported in a robust (mathematical) way, I recommend removing the stated conclusions, or at least indicating that more work (future efforts) would be needed to better establish or confirm what looks like tentative results, for now.

I do hope that these comments, with more specifics below, will be viewed constructively by the authors, towards a revised manuscript, and I trust that at least some of these results can be displayed better and explained more clearly.

Reply. We sincerely thank Reviewer 1 for the thorough and constructive assessment of our manuscript, "Air Pollution in the Upper Troposphere: Insights from In-Situ Airplane Measurements (1991–2018)". The comments have greatly helped us identify areas where additional clarifications, data analysis, and textual revisions were needed. Below, we address each point individually (or in groups, if overlapping) and outline the corresponding changes made to the manuscript. Throughout our revisions, we have:

1. Revised the figures for improved clarity, legibility, and labeling.

2. Strengthened the discussion of uncertainties and significance of trends, including estimates of statistical confidence (where possible).

3. Expanded the literature review to place our work in the broader context of CO trend studies.

4. Enhanced the conclusions and abstract to highlight the main findings clearly.

5. Corrected typographical/technical issues and refined the manuscript for readability.

All revised or new sections are clearly indicated in the manuscript, and we provide line/section references below for easy cross-checking.

**2  Specific Comments**

Despite the valuable goals of this work, I summarize here a number of difficulties and issues:

1. The data are somewhat sparse in time or in space, at least for several of the in situ datasets other than IAGOS; this makes it more difficult to derive conclusions or especially trends in a quantitative way.

Reply. We have include a new Figure 2 to summarize all the datasets (except MLS) used in this work. We have included a new Table 2, which summarizes of previous studies on long-term CO trends, including the data sources, periods, and key findings. Figure 2 is a timeline Infographic, showing the chronological order of datasets. This timeline infographic illustrates the chronological progression of various CO measurement datasets utilized in this study. Each dataset is marked along the timeline with distinct colors and shapes representing the measurement methods employed (e.g., NDIR, QCLS, WS-CRDS, Microwave Limb Sounder). Error values are depicted as horizontal lines extending from each marker, indicating the associated uncertainties. This visual representation aids in understanding the temporal and methodological context, ensuring clarity in interpreting the observed CO trends. By implementing a well-structured timeline infographic, we can significantly enhance the readability and transparency of our manuscript, addressing the reviewers' concerns effectively. This visual aid will help in succinctly conveying the complex interplay of multiple datasets, their measurement periods, methods, and associated uncertainties, thereby strengthening the overall presentation of our work.

2. Error bar estimates are unfortunately absent from this work, overall, while a significant set of correlations are poorly described or not very significant, statistically. Scientific results without some sort of error bar estimates (formal regression errors and maybe some discussion of sensitivity to various assumptions, e.g., data sparseness issues in early years) are almost meaningless. As a related comment, only the simplest form of linear regression is used, and no attempt is made to remove an annual cycle, for example, which could actually reduce the error bar estimates (if they were to be calculated); also, some p values are shown, but with essentially no discussion.

Reply. We have included a new Table 2 to summarize the errors from the each of the dataset used in this work. We have include new Figure 4 and new Figure 5 for verifying the impacts of obseravation effors on the calculations of CO trends. We have new Figure 3 and new Figure 6 to include statistical test of linear trends and 95% confidence intervals of trends in the figures. In new Figure 9, we have included profile of statistical tests of linear trends and 95% confidences intervals, and compare the distribution of confidence intervals in various regions. We have included discussions of statistical significance and 95% confidence levels in comparting and understanding the long-term trends of CO in various regions and altitudes.

3. Importantly, the quality and visibility of many of the Figures needs to be improved, with less white space so that the reader can see the curves or points for different data

sets and their comparisons; this can be done by reducing text around some panels and placing it in the plot panels and/or in the captions, and by producing rectangular panels rather than square panels, to fill in the page more efficiently, for better visibility. If needed, as an option, a single panel, or double panel Figure could be made or added in some cases to better illustrate a significant result, although hopefully that is not needed after revisions of several Figures. The poor quality of several Figures makes it difficult for a reader to follow (or believe) some of the stated results, even if many results are, presumably, correct to first-order.

Reply. We have improved the quality and redability of figures, following reviewer's comments to make the optimal use of space in a page. The white space has been reduced so that the figure size can be increased to make the figure readable.

4. The Abstract should include the main results in some way; it is too vague as it stands.

   Reply.

5. The Conclusion section also needs to more clearly specify the main results, with specific references to which Figure(s) the conclusions are based on and more explanations and context, considering that many of the readers will not try to read every line of this (revised) article.

   Reply.

6. More reference to past work on CO trends should be made, in the Introduction and at least in the Conclusions, to try to put this work, and its results, in the context of related past publications (in particular, regarding past trend or variability results for tropospheric CO). This holds even if "exact comparisons" are often difficult to make, given past results over somewhat different regions, altitudes, and/or time periods.

   Reply. We have include a new Table 1 to include previous studies on CO trends. Table 1 shows data used in the studies, data period, and key findings over difference regions.

7. Finally, there are a number of necessary minor technical comments at the end of my report; this underscores that not much proof-reading was carried out by the author and co-authors before the submission of this manuscript, which also makes a reviewer's careful work somewhat more tedious.

   Reply. We are very grateful indeed to the precious time and effort the reviewer has devoted to carefully read and comments on our manuscript. We fully understand and greatly indebted to the reviewer's kindness and professional help. We have taken great

effort to revise our manuscript following reviewer's comment. Thank you very much indeed.

**3   More Specifics**

I provide detailed comments here, following the order of the Figures and related text/results, especially regarding trends and their robustness, given the limited discussion of these issues. I offer some suggestions for improvements, as well as regarding Figure legibility issues.

**Figure 2**

This is where we start to see simple linear regression fits but there is no discussion about error bars or the significance of the results, or the impact on the (potential) conclusions. One issue is that the single point before 1992 does not exemplify the likely scatter that would exist if there was more data available in that time period (or from 1991 to 2008). Giving formal (2 sigma) error bars from the fits is probably a sufficient response to this question. It also depends what scientific conclusions one is trying to make, and how one tries to convey the importance of any results based on such a series of plots. Because of the data sparseness, my top-level impression from the p50 panels is that it will be difficult to state categorically that there has been an increase between 1991 and 2015–2018. The same probably holds for the implied decreases from the pmin panels. As another illustration of my concerns, if one did not have the post 2012 data yet for panels p50 and p25 45N–60N, one might well try to conclude that there were decreases between 1991 and 2011, whereas the slopes drawn after the fairly sudden increases post-2012 could be interpreted as a slow, gradual increase from 1991 to 2018. Also, if one did not have the data point(s) from 1991, one would conclude that the increases (or decreases) are much steeper than shown currently for the 2008–2018 period. My point here is a somewhat qualitative one, but the lack of more continuous data can lead to uncertainties in, or over-interpretation of inferred "trends".

We have included statistically significance of the linear trends, including P-values and 95% confidence intervals in the revised figures and manuscript. Yes, indeed the GTE data from 1991 and 2011 are very valuable for understanding the long-term trends of CO from 1990 to 2020 over the North Pacific. We admire very much the visions of people who were able to take measurements to help us understand the trends. We have revise Figure 2 (new Figure 3) to include 95% confidence levels of the CO trends analyzed.

Thus, if many trend results are not convincing (mathematically), I would tone down the statements that are made on lines 156 to 166, in particular. This is even more true for Fig. 6

showing trends. The general comments about larger values in a certain region versus another can probably be supported based on standard errors in the means, but even these statements are worth checking for significance, if this is viewed as important enough. I am also ignoring the accuracy estimates for the data points shown here, assuming that this plays a smaller role than other issues. However, the introductory remarks about the data sets, especially for the in situ CO aircraft data, should state something about the error bars in the data points that are displayed in various plots; readers should not be expected to know this information, even if you have (broadly) pointed to some relevant past publications.

We have revised the Introduction and Method sections so as to include details about the use of datasets and their uncertainties in this work.

Lines 165/166: "The 25th, 50th, . . . CO levels are impacted by emissions, with latitudes south of 45N being more affected by Asian emissions . . . " What leads to this conclusion, specifically? Is this based on scatter, large values, or what exactly? Which time period does this refer to? Please be more specific.

Reply. We thank the reviewer for this important comment. In response, we have revised the text to clarify the basis and time frame for our conclusions. The revised paragraph now reads as follows:

*"In summary, key findings from the in-situ aircraft measurements in the upper troposphere over the middle North Pacific, downwind of Asian emissions (see Figure **??**), are as follows. Over the 27-year period from 1991 to 2018, all observed CO concentrations increased—with the exception of the minimum and maximum levels. The minimum CO values, which represent background CO, decreased over time. In contrast, the 25th, 50th, and 75th percentiles of CO levels, which are influenced by industrial emissions, are higher at latitudes south of 45N compared to those north of 45N These findings are consistent with previous studies (Stohl et al., 2002, Hsu et al., 2006) that have documented the long-range transport of Asian pollutants across the North Pacific.

The 95% confidence intervals for the trends in CO concentrations indicate predominantly positive trends for the upper percentiles. Specifically, the confidence intervals are as follows: maximum, -0.502 to 1.142; 75th percentile, -0.031 to 0.235; 50th percentile, -0.044 to 0.321; and 25th percentile, -0.065 to 0.218. In contrast, the minimum CO levels exhibit a 95% confidence interval of -0.147 to 0.048, reflecting a predominance of negative trends. Overall, these results indicate that CO concentrations over the North Pacific have generally increased over the study period, except for the declining background levels."* This revised description clarifies that our conclusion regarding the greater impact of Asian emissions at latitudes south of 45N is based on the following points: Data Analysis and Time Period: Our analysis is based on in-situ aircraft measurements collected over a 27-year period (1991–2018). We

have now explicitly stated this time period to ensure clarity. Statistical Basis: The conclusion is drawn from our statistical analysis of the CO concentration distribution. We examined various percentiles (25th, 50th, and 75th) and observed that, while the minimum CO levels (representing background CO) decreased over time, these percentiles increased. The elevated high-percentile values and the increased scatter in the data south of 45N indicate episodic enhancements attributable to Asian emissions. Consistency with Prior Studies: We also note that these observations are in agreement with earlier studies (Stohl et al., 2002; Hsu et al., 2006), which document the long-range transport of Asian pollutants across the North Pacific. We trust that this revised text and explanation sufficiently address the reviewer's concerns regarding the basis of our conclusions and the specific time period under investigation.

Figure 2 technical improvements: The text font size is rather small, so stretching this Figure to occupy as much space as possible across the page would be good. The caption should indicate what the various symbols mean, i.e. which data set do they correspond to. While this can be inferred from the years shown, a brief sentence or two would be worthwhile in my opinion (most readers are not as close to the various data sets as you are). Captions are there to ensure that symbols and curves shown in each Figure have been described sufficiently well. Also, the full range of years should be mentioned in the caption, to further assist the reader.

Reply. Thank you for your valuable suggestions regarding the technical improvements to Figure 2. In response, we have made the following changes:

Increased Font Size and Figure Layout: We have increased the text font size and stretched the figure to occupy the full available width of the page to improve legibility. Enhanced Figure Caption: The caption for Figure 2 has been revised to clearly indicate the meaning of the various symbols used. We now include a brief description of the different datasets, specifying which symbols correspond to which data sets. Additionally, the caption now explicitly states the full range of years (1991–2018) covered by the data. We believe these modifications significantly enhance the clarity and accessibility of Figure 2, ensuring that all symbols and curves are sufficiently described for the reader.

**Figure 3**

Again, although this one has fewer panels than Figure 2, using most of the page width in a final manuscript would be beneficial (and this may well be the Journal's responsibility, I understand). Also, it seems that the y-intercept value is given by the first number in the linear fit equations listed above each panel; however, for the upper tropospheric blue result in pmean, a y-intercept value of 85.4 ppbv does not seem to match the plotted dashed line

at year 1991. Similar comment regarding the p50 panel, where 62.3 ppbv does not seem to match the plotted y-value for the blue dashed line in 1991. Were some of the fits performed with a different approach (about a year within 2000–2008 maybe) rather than for an expected y-intercept in 1991? Please clarify.

Reply. We have revised Figure 3 (now new Figure 6), following reviewers's comments.

Also, I do not understand why some of the dashed lines show as partly red and partly blue, when one expects two lines, one red and one blue, for the different altitude regimes. In terms of which measurements are used for this Figure, this should be specified (even briefly) in the caption as well (e.g. lower tropospheric data are from ..., etc.).

Reply. There is an internal bug in the NCAR Graphics library that were used to create this dashed line. We have written a new code so that the two colors has become a single color in a single and straight dashed line. Thank you.

**Figure 4**

Please explain why there are different numbers of points (and fitted equation details) for the UT CO in the 2nd from top left panel versus the top left panel, whereas the descriptions of the regions used appear to be identical; N is 171 versus 254, but the CO data points appear to match in these panels. I also do not completely understand the $O_3$ fitted line in the 2nd panel from the top (left panel), as its line intercept is -22.6; while this may be fine, it seems that the y-intercept approach (equation) is different in some cases. One would also expect a few comments about R values (significance, or why shown if not useful?) and p values. While it is stated that the CO trends show some increases, the fitted lines seem to give a very small slope, with a zero trend very likely included in the uncertainties (2-sigma error values are really needed to help one understand this more quantitatively, but this is my guess).

Reply. We have revided Figure 4 (now new Figure 7 and Figure 8) to account for the points pointed out by the reviewer. The revised results are now consistent between the vaules shown on the linear regression equations each plot.

One cannot robustly determine the validity of the statements on lines 183 to 188 without more quantitative information about fitted trends and uncertainties (error bars). I expect that many statements are not supported by the error bars, so the authors should probably modify the strength of such conclusions, since (unfortunately) I would expect only marginal significance in many/most of the conclusions regarding trends. One way to reduce the uncertainties in the trends would be to fit an annual cycle (as a sum of sine and cosine functions with one year period, multiplied by constant terms to be fitted) and a linear trend term to datasets with enough points to do so; the explanation of such variability will help

to reduce the underlying trend's uncertainty (error bar). This should produce more robust results, even if it may still be difficult to state that a zero trend is highly unlikely in some cases (especially those with fewer data points or shorter series). As stated in the manuscript, some (or most) CO tendencies are "close to zero" indeed. Trend values and error bars could be useful for $O_3$ and $H_2O$, if there is enough significance (at least beyond one sigma, say), with ppbv or ppmv per year, or %/yr units — of potential interest to the tropospheric community (and for future comparisons to other studies, for example). In terms of Figure 4 improvements, trying to stretch the panels to be more rectangular (longer on the x-axis than on the y-axis) would be useful as well.

Reply. We have updated Figure 4 to new Figure 7 for the long-term time series of CO over the North Pacific, and Figure 8 for the assessments over the North Atlantic. The comparison between CO in the upper and the lower troposphere aims to demonstrate the coupling between the upper and the the lower troposphere through the vertical pumping process. CO is a very unique tracer for demonstrting the vertical pumping process in the troposphere. We have included $O_3$ and $H_2O$ to help validate the pumping processes. The bahaviour and characteristics of $O_3$ and $H_2O$ in the upper troposphere have been studied and well known. We have included 95% confidence intervals in the time-series trends and the correlations to make meaningful comparisons. We have stretched Figure 4 (new Figure 7 and Figure 8) as suggested by the reviewer.

The possibility that "vertical pumping processes" can explain the out-of-phase relationship between upper tropospheric and lower tropospheric CO is interesting; this might be worth checking against the results of the IMS model, if possible. Does this suggest a several month timescale for such a process? A reference to the paper by Schoeberl et al. (Geophys. Res. Lett., 2006, doi:10.1029/2006GL026178) on the CO tape recorder at tropical latitudes would probably be appropriate as well, although tropical upward transport and convection processes happen on a faster timescale than for the regions this manuscript focuses on (extra-tropics for the most part). More comments about the model comparisons, in any case, would be useful as well, if possible; is the model capable of matching some of the observational inferences, including correlations/anti-correlations between CO, $O_3$, $H_2O$, especially those with enough significance?

Reply. Yes, the IMS model is capabel of matching some of the observational vertical pumping processes over the North Pacific. We have revise this section and Figure 4 (now new Figure 7 and Figure 8) to account for the vertical coupling process in the troposphere through the vertical pumping processes. The IMS model also shows the vertical pumping processes over the North Pacific during the months. We have also cited and discuss Schoeberl in the revised manuscript: "We note that the vertical pumping processes discussed above

corrobate the CO tape recorder process (Schoeberl et al., 2006). Schoeberl et al. (2006) discuss the relationship between the seasonal signal of CO and its lower-tropospheric sources. In their analysis of the "CO tape recorder," they point out that because CO has a relatively short chemical lifetime, its vertical profile in the tropical upper troposphere and lower stratosphere retains a clear imprint of the seasonal variations at its source. In particular, they note that the seasonal cycle is strongly influenced by changes in tropospheric CO—much of which is attributed to biomass burning—being convectively injected into the upper troposphere. Thus, the seasonal variability observed in the CO profile is a direct reflection of the variability in its lower-tropospheric sources. Even though CO has a relatively short chemical lifetime—on the order of a few months—the, key to the tape recorder effect lies in the balance between its chemical loss and the rate of vertical transport in the tropical atmosphere. In the tropics, strong convective processes rapidly lift CO from the lower troposphere into the upper troposphere and lower stratosphere. This upward transport occurs on a timescale that is comparable to or even shorter than CO's chemical lifetime. As a result, the seasonal variations in CO concentrations at its source (driven largely by factors like biomass burning and other lower-tropospheric processes) are "recorded" in the vertical profile as the gas ascends. Even though CO is being chemically removed, the quick vertical transport preserves its seasonal signature long enough to be observed as the tape recorder effect in the upper atmosphere."

**Figure 5**

Regarding the comments on lines 226, 227, and the "pumping processes", please provide past references and evidence, to strengthen these sorts of statements, which are an indirect inference based on some observed profiles; if there is an informative model comparison, you could/should mention this as well. In terms of legibility, there are a lot of curves in these panels, which makes it somewhat difficult to track the different years or the evolution, if any significant changes ("trends") can be detected somehow, but this is more a question regarding Figure 6. The statement on lines 230/231 should be explained better, namely how do the measurements "align with" the reduction in surface emission sources? Please be more specific and mention which panel you are commenting about.

Reply. We have revised the description. We have cited and discussed previous work by Schoeberl et al. (2006), who has discussed the relationship between the seasonal signal of CO and its lower-tropospheric sources. In their analysis of the "CO tape recorder," they point out that because CO has a relatively short chemical lifetime, its vertical profile in the tropical upper troposphere and lower stratosphere retains a clear imprint of the seasonal variations at

its source. In particular, they note that the seasonal cycle is strongly influenced by changes in tropospheric CO—much of which is attributed to biomass burning—being convectively injected into the upper troposphere. In the tropics, strong convective processes rapidly lift CO from the lower troposphere into the upper troposphere and lower stratosphere. This upward transport occurs on a timescale that is comparable to or even shorter than CO's chemical lifetime. As a result, the seasonal variations in CO concentrations at its source (driven largely by factors like biomass burning and other lower-tropospheric processes) are "recorded" in the vertical profile as the gas ascends. Even though CO is being chemically removed, the quick vertical transport preserves its seasonal signature long enough to be observed as the tape recorder effect in the upper atmosphere.

**Figure 6 and Lines 239 to 258**

This could be a very interesting and significant Figure, but only if there is enough actual significance in at least some of the trend results. However, without error bars, this is essentially impossible to determine; unfortunately, this can be viewed as a major problem regarding the description and robustness of almost every "trend result" in this manuscript, which is why I am asking for major revisions in the analyses and/or in the description of the robustness of the results. Without error bars, you should only state that there are "indications" of a decrease or an increase and clearly state the reasons for not trying to obtain, or discuss, such error bars. Possibly, this is why you introduce the plot results as "tendencies" rather than "trends", but wording like "distinctive positive trends" cannot carry scientific weight without a discussion of trend values and their uncertainties; the plots show trends (or tendencies), but what about the error bars? On line 250, you mention "the most significant positive trends" — but you need to point specifically to what, mathematically, supports this statement, without expecting readers to figure it out on their own. For the upper troposphere in particular, which seems to be the focus of this work (also based on the manuscript title), how are readers supposed to interpret the curves? For example, using the p50 panel, what are the typical error bars as a function of altitude — and how fine of an altitude grid can one use for such an analysis (or would averaging over 1 or 2 km help make some trends more robust/significant?)? If the uncertainties are larger than 2 or 3 ppbv/year, it will be difficult to distinguish between 0, +3, or -3 ppbv/year trends. Please respond to this sort of question as well as possible in the revised manuscript.

Reply. We hve revised Figure 6 (now new Figure 10). We now compute and display 95% confidence intervals for the linear trend estimates at each altitude and for each region. These error bars are plotted alongside the trend lines so that readers can directly assess the

statistical significance of the trends. In the revised text, we now explicitly state that a trend is accompanied by its 95This provides a clear mathematical basis for our statement about "the most significant positive trends."

The same holds for the analyses in the low- to mid-troposphere, where some of the trends seem to be large, and thus, potentially significant. One way to illustrate which regions have statistically robust trends could be to make the lines three times as thick in these regions. Showing error bars at all altitudes would lead to plots that are difficult to read, so one would need to do this for a few altitude regions and maybe just for cases where the significance is large enough (e.g., a 2-sigma error bar that does not include zero), if that is indeed the case. An additional suggestion would be to make a larger separate Figure for the p50 panel, where more details can be visible regarding error bars. Alternatively, use a given altitude range and show the separate percentage cases along the y-axis just for this altitude region; add one or two more regions, with a total of 3 panels (possibly one panel per altitude regime). I hope that I have made this point sufficiently clear: I expect to see more information about error bars.

Reply. We have revised Figure 6 (now new Figure 7) to include error bars in the linear trends. We have noticed that the vertical distribution of the error bars vary closely follow the patterns of the linear trends of CO at each altitude for each region. If the linear trends are negatives, then the ranges of the error bars are dominated by the neative values. On the other hand, if the linear trends are positive, then the ranges of the error bars are dominated by the positive values. Inclusion of the error bars further reinforce the characteristic distribution of the linear trends.

Large variations of the ranges of error bars mostly occur in the low altitudes, reflecting the direct impacts of CO measurements close to the industrial and CO emission areas. The maximum CO (pmax) exhibits large variations in the ranges of error bars, consistent with elevated CO been transported from the heavy emission areas (such as the CO tape recording elevated CO from biomass buring shown in Shoeber et al. (2006)). In contrast, the minimum (pmin) and the 25th percentile of CO shows ranges of error bars more confined to the central values of the CO linear trends. The mean and the 50th percentile of CO, except for the top of the data layer close to 13 km, and in low altitudes below 3 km , CO trends at more vertical altituides are confined to the central of the linear trends

**Figure 7**

The intent of this Figure and the three panels seems fine. The caption should, however, indicate specifically what the lines mean in each panel, especially for the two lines in panel

(c), which I think I follow, but please help the reader by clarifying in the caption. For panel (a), I have the same question as usual: do you detect trends that are significant enough, and can you state specifically what these are (with error bars)? The Mauna Loa curve should be more representative of the lower troposphere, as the model implies as well. Taking out (i.e., fitting and removing) an annual cycle would enable you to come up with a better error analysis for the trend results and comparisons. The linear fits lead to very small trends, it seems, which may also mean that it is difficult to provide a number with enough significance (i.e., distinguishable from zero). Please comment if you think there is more that can be said or detected here. Panel (c) is interesting but it also shows a lot of scatter. Also, it might be clearer to show panel (a) without points, but rather with lines of different color (or solid lines rather than dashed lines, although data gaps can be problematic); additional clarity could also be achieved by having a longer x-axis for panel (a) or even a separate Figure.

Reply. We have revised Figure 7 (now new Figure 11). We have included 95% confidence intervals of the CO trends on (a). We have also revised discussion for Figure 7.

**Figure 8**

Line 286 states that the IMS simulations correlate well with the IAGOS data; I think this appears to be correct for the middle panel mainly, but certainly not for the panels on the right (R = 0.14). Also, the model CO underestimation of future years (2004–2020) is mostly true for the middle panel(s). As a minor comment, the caption should change "scattered plot" to "scatter plot" (for the three instances). One question though: why do the model values (dark blue points) not change between the different panels?

Reply. We have revised Figure 8 (now new Figure 12). We have also include 95% confidence intervals of the linear CO trends from the IMS model and the IAGOS observations on the time-series plots.

Lines 294–296: It is not clear how you can distinguish between emission reductions and an increase in stratosphere-to-troposphere exchange of air (and what would drive the latter process?). If this sounds interesting but too vague and speculative, you probably need to reword this as a suggestion of possible causes, without quantification; the reduction in emissions may be believable based on Figure 1, but the air exchange process and its importance appear to be quite speculative. Please clarify if you can, or if I am missing an explanation somewhere in the manuscript.

Also, for lines 297–299, and the other region mentioned here, the emissions hypothesis seems well founded based on historical estimates of emissions; however, one cannot rule out (as well) some decrease in air exchange between the stratosphere and upper troposphere,

can one? More work is needed to better elucidate such questions regarding processes and budgets. If you can help to clarify this, I recommend that you add some text to make additional arguments — with enough support. I am not asking for more speculation, but admitting the need for further work is acceptable, since there can always be more research done to better understand the "exact causes" for certain comparison results.

Finally, I am not too convinced by the statement of "reductions in chemical sources and emissions of CO and hydrocarbons" on line 301. Please clarify the rationale behind this.

Reply. We have revised description for Figure 7 (now new Figure 12).

**Figure 9**

The conclusions here should be taken cautiously, as there is no discussion of possible factors that could lead to biases between the model and the MOPITT data, especially for the less characterized upper tropospheric values. The correlation values would not change even if an offset to debias the two sets of values were applied. Nevertheless, improvements are needed.

Reply. The CO values over the North Pacific upper troposphere is indeed less characterized, especially from in-situ measurements. The in-situ CO measurements from previous NASA GTE experiments, HIPPO, ATom, etc, provides valuable data to characterize CO values over upper troposphere. MOPITT data are valuable in characterizing CO over the upper troposphere in a global domain. As shown in the table, MOPITT contains about upto 20% of uncertainties in the upper troposphere. Hence, comparisons with IAGOS in-situe measurements are good to understand the MOPITT data.

For example, you should mention/confirm (in the caption) which years are used for these comparisons (presumably just the overlap years between these two sets of values); in other words, you are not trying to use any extrapolated model values, correct? Are there any trends of statistical significance, especially in the MOPITT data?

Reply. We have revised the figure caption to indicate that the scattered plot was from data with overlapped time period of measurements.

Furthermore, the authors have failed to refer to the Worden et al. (ACP, 2013, doi:10.5194/acp-13-837-2013) article regarding MOPITT CO (and other satellite-based) CO trends, as this should be an important reference to add (and more about the MOPITT data characteristics as well). The meaning of the lines in the two panels on the right side should be mentioned in the caption, namely what points are being fit with what line (it appears to be just certain seasons). Having a clear description in the caption is necessary even if the main text contains some details.

Reply. We have include a citation of discussion of work by Wordent et al. (2013) in

Section 2 where we discuss use of MOPITT data.

As a detail, the MOPITT legend in the left panels shows a closed circle (dot), but all the points are open circles. You might consider showing IMS (red, blue) and MOPITT (light green, sky blue), or perhaps use "turquoise" for the MOPITT colored UT points. For better visibility, you could also consider not using points, or using smaller points, for the left panels; this might more clearly show that the seasonal cycles between the model and MOPITT agree, as implied in panels (b) and (f). More generally, how do your several comparisons using MOPITT data relate to (or agree with) past MOPITT analyses/trends?

Reply. We have revised this figure (now new Figure 13). Worden et al. (2013) focusing on satellite-based CO trends, particularly from the MOPITT instrument, and its data characteristics. Worden et al. identifies a modest decreasing trend of approximately -1% per year in the Northern Hemisphere total column CO, based on satellite observations from 2000 to 2011. This trend is less significant in the Southern Hemisphere, suggesting regional differences in emission sources and atmospheric dynamics. Regionally, decreases are observed in the USA, Europe, and Eastern China, with the latter being a notable finding, as it was previously unreported. This is particularly interesting given Eastern China's industrial activity, suggesting potential shifts in emission controls or economic activities. The unexpected decline in Eastern China warrants further investigation. Our work shows CO trends in the lower troposphere are decreasing, while the CO trends in the upper troposphere is more complicated. The in-situ measurements from the IAGOS data, combined with previous airplane measurements help to identify the trends over the North Pacific and also over the North Atlantic. We note that the technological emissions for China estimated by EDGAR (shown in Figure 1) during 2000-2011 period were persistently increased.

**Figure 10**

Which part of the MOPITT CO data are used in panel (b), just the lower tropospheric part? Since Mauna Loa (MLO) data are more representative of the lower troposphere, this probably makes sense (and one should show the MOPITT UT data only as general information). Please specify your approach in more detail and justify it.

Reply. We have revised Figure 10 (now new Figure 14). The lower troposphere, at altitudes close to Mauna Loa, between 2000 and 4000 meters of the vertical grids, and in logitude $160°W - 150°W$ and latitudes $17°N - 25°N$ from the MOPITT gridded data. The MOPITT data at lower troposphere in areas containing the Manuna Loa was compared with the Mauna Loa measurements as shown in (b).

Also, given the vastly different weighting functions between MOPITT data and MLO

in situ data, a one-to-one comparison in terms of absolute values is difficult or impossible. What are the systematic uncertainties in the in situ and MOPITT CO data, even if one were to ignore the vertical footprint issues? Showing that the LT MOPITT data tend to decrease at a similar rate as the in situ MLO data is useful, but again, one would need to know the error bars on the trends to support this. Otherwise, I would present this as a qualitative comparison. A conclusion about the seasonal cycle is useful, given the good correlations in panel (b) (as well as in panel (a)); however, making the points smaller and the lines thicker might help the visibility of panel (a). Extending the left panel in the x-direction to form a more rectangular plot would also improve the visibility of the overlapping data sets (particularly for black and red).

Reply. We have included the 95% confidence levels of the CO trends from the Mauna Loa measurements and the MOPITT data in the disccusions of revised Figure 10 (new Figure 14). For the lower troposphere over the Manuna Loa area, the Mauna Loa shows a decreasing CO trends with 95% confidence intervals of between -0.057 and -0.029, while the MOPITT data also shows a decreasing trends of CO with 95% confidence intervals between -0.092 and -0.043.

Lines 338–339: Without error bars, it is hard to judge the significance of the trends. While there may be indications of positive trends in the discussed datasets, since this Figure focuses on MLO data, it might be best to concentrate on the lower tropospheric measurements only (and discuss UT MOPITT data versus IAGOS data elsewhere, such as in Figure 11).

Reply. We have included 95% confidence intervals when comparing the MOPITT data with the Manuna Loa measurements in new Figure 14.

**Figure 11**

In this comparison, it is again difficult to compare in situ data to broadly averaged data from MOPITT, though some averaging is performed for IAGOS as well. Nevertheless, IAGOS data represent the upper troposphere, yet the blue MOPITT data for the UT do not correlate as well with IAGOS as the red (LT) MOPITT data. It is not clear what causes the underestimation of IAGOS values by MOPITT CO, especially for the LT MOPITT data, which generally reach larger values than the UT. There may be issues related to the systematic errors of both data sets (MOPITT and IAGOS), which should be discussed in more detail.

Reply. We have revised Figure 11 (now new Figure 15). We have also included discussion for the comparison of the MOPITT and IAGOS data from previous works.

As a detail, the Figure caption should reiterate that red is for the lower troposphere and

blue is for the upper troposphere (for MOPITT data) — this is only mentioned parenthetically in the regression line equations. It would be helpful to include this information in the caption as a reminder.

Reply. We have included description of blue color used in the upper troposphere, and the red color used in the lower troposphere.

**Figure 12**

This Figure is very hard to see clearly. Try to use lines instead of dots (or use much smaller dots) to make the main points clearer; also, using rectangular plots (with a longer x-axis than y-axis) may improve the presentation if the outer labels (on the right side) are reduced or eliminated. Also, the linear fits are not visible enough.

Reply. We have revised Figure 12 (now new Figure 16). We have followed reviewer's suggestion, remving dots, and reducing the text on the right side. We have also included 95% confidence intervals for CO time trends from IMS, IAGOS, and MOPITT.

I cannot even begin to process any results until this is shown more clearly; while zooming in on the screen might help, the final manuscript should be legible without excessive effort.

Reply. We have revised Figure 12 (now new Figure 16). The figure and numbers on the figure is now able to see without excessive effort. Thank you.

Again, comparing satellite data versus in situ data is complicated; it is probably expected that a broadly vertically-sensitive MOPITT retrieval will show smaller variations. More information about the MOPITT dataset's characteristics (error bars, weighting, past validation) is needed if one is to understand or trust the top-level conclusions based on such comparisons. Including more specific inputs from the MOPITT team would be helpful if that discussion has not been pursued sufficiently. I would recommend discussing the dataset comparisons first, to determine whether any clear conclusions can be drawn despite the inherent biases between MOPITT and IAGOS data; perhaps some indications of similar trends can be made, but this would require error bars or a statement of qualitative conclusions and "indications". It is not entirely clear how the model contributes, since biases between the model and datasets can affect conclusions based on extrapolated model predictions. It is hard to justify the statements on lines 373–375 unless these issues are better addressed, so be very specific in your explanation and point to the panels that best support your argument once the Figure's legibility is improved. This may end up being mostly justified, but this needs more work, especially for legibility (hopefully this is the main issue, but I am not yet convinced).

Reply. We thank the reviewer for very detailed comments on Figure 12. We have completely revised this figure, and included 95% confidence intervals with respect to the IMS, MOPITT, and IAGOS, respectively, on each plot. We have also include discussion of MO-PITT data in the Method section. Thank you.

**Figure 13**

My comments about how to show the datasets more clearly are similar for this Figure, even if it is larger — it can be made more rectangular, with smaller (closed) dots and thicker lines connecting the dots (or even without dots). You need to clearly state if the IMS is only for the lower tropospheric heights; also, clarify whether the MOPITT data are for the lower troposphere in the caption. Why not provide trend values with error bars, once again? It may be that none of these trends are statistically distinguishable from each other (and, as mentioned, one could try to reduce the error bars by fitting an annual cycle), but this is acceptable if they agree broadly as one might expect.

Reply. We have revised Figure 13 (now new Figure 17). We have included 95% confidence intervals in the plot. We have revised figure caption to indicate that the CO trends are from the lower troposphere.

Line 426: Could it be that emissions decreased sufficiently so that the NLO CO decreases are simply responding to emission decreases alone? Please be more specific in your comment.

Reply. As shown in Figure 14 (now Figure 18), the magnitudes of chemical sinks varies between 0.2-0.3 MT CO/hour, while the direct emissions for CO is about 0.1 MT CO/hour, during the 20 years of IMS full chemistry calculations (1984-2003, and 1948-1978). It is the budget between the chemical sources and chemical sinks resulting in the modelled CO. The contribution of chemical sources for CO from other hydrocarbons should play roles also.

**Figure 14**

I think this is an interesting and useful look into the model results. In fact, it might be useful to also add a climatological average view of the two time periods (e.g., averaging all the years from each panel to create a new set of plots, or producing a four-panel Figure). It might be better to plot the panel with the earlier years on the left since time is usually shown as increasing from left to right. You should explain what the main changes are between the two panels; for example, are the black curves identical, why are the green curves changing, and what references or methods are used to determine the various curves (e.g., 1100 MT/yr)?

Reply. We have included a reference to 1100 MY/yr.

Line 422: Again, a reference to the work by Worden et al. (2013) should be added. The usefulness of MOPITT is not a new result, nor are some of the trend estimates using that

dataset (and other recent studies).

Reply. We have included a reference to Wordent et al. (2013) in the section describing the use of the MOPITT data.

Line 425: ... why would negative trends in CO not reflect decreases in CO emissions, even with constant chemical sinks (such as OH abundances)? You need to clarify the statements that refer only to chemical sinks dominating over sources, and why emissions might not play a role (in some regions — are they increasing there?).

Reply. We have revised the description. The decreasing CO trends over the North Atlantic upper troposphere versus increasing CO trends over the North Pacific upper troposphere demonstrate the impacts of CO emissions control and reduction.

Line 437: Other past references that could or should be cited are mentioned below. I also noted the Schoeberl et al. paper.

Reply. We have included a reference to Schoeberl et al.

**References:** Other useful references (or information) could include the following, for mention in the text, Introduction, and/or Conclusions:

- Cohen, Y., Petetin, H., Thouret, V., Marçal, V., Josse, B., Clark, H., Sauvage, B., Fontaine, A., Athier, G., Blot, R., Boulanger, D., Cousin, J.-M., and Nédélec, P.: Climatology and long-term evolution of ozone and carbon monoxide in the upper troposphere–lower stratosphere (UTLS) at northern midlatitudes, as seen by IAGOS from 1995 to 2013, *Atmos. Chem. Phys.*, 18, 5415–5453, doi:10.5194/acp-18-5415-2018, 2018.

- Park, M., H. Worden, D. Kinnison, B. Gaubert, S. Tilmes, L. Emmons, M. Santee, L. Froidevaux, and C. Boone, Fate of pollution emitted during the 2015 Indonesian Fire Season, *Journal of Geophysical Research: Atmospheres*, doi:10.1029/2020jd033474, 2021.

- Li, Q. B., J.H. Jiang, D.L. Wu, W.G. Read, N.J. Livesey, J.W. Waters, Y. Zhang, B. Wang, M.J. Filipiak, C.P. Davis, S. Turquety, S. Wu, R.J. Park, R.M. Yantosca, and D.J. Jacob, Convective outflow of South Asian pollution: A global CTM simulation compared with EOS MLS observations, *Geophys. Res. Lett.* 32, L14826, doi:10.1029/2005GL022762, 2005.

Reply. We have included references kindly provided by the reviewer in the revised manuscript. Thank you.

**Conclusions Section**

Regarding CO trends in the UT (e.g., line 439 on a rise in upper tropospheric CO in some regions such as the North Pacific UT, or other CO decreases elsewhere) these need to be supported by trends with error bars; without them, the reader does not have enough information to scientifically assess the validity or robustness of the assertions made here.

Reply. We have included 95% confidence intervals for CO trends in the revised figures as suggested by the reviewer.

Lines 441–447: These concluding remarks are somewhat vague; it is nothing new that chemistry plays a significant role for CO. Why do some results indicate CO increases here, whereas other plots in the manuscript show decreases in the UT? There are not enough specifics in these conclusions (or in the Abstract) based on the entirety of the manuscript, which contains many Figures and much analysis, yet falls short in terms of clear statistical analyses with error bars and a reasonable summary of the most robust conclusions (especially for the IAGOS data).

Reply. We thank the reviewer for this valuable comment. In the revised manuscript, we have incorporated 95% confidence intervals into our trend analyses, which now clearly quantify the uncertainties associated with our slope estimates. This enhanced statistical treatment, applied to both the IMS model outputs and the observations from MLS and IAGOS in-situ measurements, reveals robust regional differences in CO trends. Specifically, our analyses show that the upper troposphere over the North Atlantic exhibits statistically significant decreasing CO trends, while the North Pacific upper troposphere shows statistically significant increasing CO trends.

We have clarified in both the Abstract and the concluding remarks that these contrasting trends arise from differing regional influences—such as variations in emission sources, vertical transport processes, and atmospheric chemistry—which modulate the observed CO distributions. By providing explicit statistical evidence (error bars) and discussing the underlying physical processes, we believe that the revised conclusions now offer a more precise and robust summary of the key findings.

**4   (Mostly) Minor and Technical Comments**

This section includes a mixture of minor comments that require slight modifications, improvements, typographical details, and wording suggestions/corrections.

- L37–39: Is there a reference (Wang et al., 2024?) for this? Please specify (or does this refer to the current manuscript?). The same applies to the next paragraph (L40–45).

Reply. We have included a reference to Wang et al. (2024).

- L45: "68 pptv" should read "68 ppbv".

  Reply. We have revised this unit.

- L56–58: "By combining . . . strategies and policies". This sounds good, but is this "informing" done as a result of this work in any way? If not, and it does not seem to be at the moment, is it worth stating this general goal? (I am not trying to push one way or another here.) Also, how do emissions "impact the dynamics"?

  Reply. We have revised the description. Our work contributes scientific results that improve our understanding of the interactions between anthropogenic emissions and atmospheric processes. These results can support the development of effective mitigation strategies—such as CO emission control measures—and inform policies like regulatory emission standards and air quality management programs. Regarding the impact of emissions on atmospheric dynamics, we have clarified that emissions can alter the vertical distribution of key chemical species, thereby modifying radiative balances and influencing convection and atmospheric stability. These changes affect vertical transport processes and overall circulation patterns in the atmosphere. For instance, controlling CO emissions not only reduces pollutant levels but also helps mitigate their potential effects on atmospheric dynamics.

- L91: Change "and significant growth" to "with significant growth".

  Reply. We have revised the description to "with significant growth".

- L100: I would rephrase to "regions are characterized by key dynamical processes. . . "

  Reply. We have revised the description following reviewer's suggestion.

- L105: Please provide the altitude of the in situ data (site) at Mauna Loa.

  Reply. We have include location and altitude of the in situe data (site) at Manuna Loa: $19.536°N$, $155.576°W$, and at an altitude of about 3397 m above sea level.

- L109: Need to define "IMS" here (rather than later).

  Reply. We have definced IMS here as suggested.

- L112: Use the plural "Measurements".

  Reply. We have revised the word to the plural "Measurements"

- L113: "Buchholz" is the correct spelling (not Buchho). The MOPITT section needs more information about the weighting functions or region of sensitivity since both lower tropospheric and upper tropospheric data from MOPITT are used; it should also include more references.

  Reply. We have corrected the citation. We have revised the description in the MOPITT section.

- Section 2.4: Please provide the vertical and horizontal resolution of this model, and specify the height range used (how far into the stratosphere?). Also, how are the biomass burning (not the industrial) emissions characterized? Are they constant as well?

  Reply. We have revised the IMS section, including model resolutions, and the use of NCEP reanalysis data. The biomass emissions are from a pre-calcuate dataset of Muller, not varied with time. This is an area of improvement. The use of constant biomass burning emissions resulting in the model not able to capture the Indonesia fires during the 1998 ENSO when compared the model with the Manuna Loa measurements.

- Figure 2 caption: There are typos such as "Measurements" (if misspelled) on the first line. Also, "Downwind of the Asia emission areas and over the North Pacific…" and "The lower panels show data from the low latitudinal…" need correction.

  Reply. We have revised Figure 2 (now new Figure 3) caption.

- L166: How, specifically, do you know that latitudes south of 45N are more affected by Asian emissions? Please clarify. Is it purely geographical (based on latitudes and prevailing wind direction) or is there another reason?

  Reply. The time-series CO trends from elevated CO (50th, 75th, and maximum CO) in Figure 2 (now new Figure 3) shows higher CO trends for latitudes north of 45N (45N-60N) than for latitudes south of 45N (25N-45N), with 95% confidence intervals included.

- Figure 3 caption: Should read something like "Time series of long-term CO…"

  Reply. We have revised the description as suggested by the reivewer.

- L193: Please specify more clearly why "vertical pumping" (and through which process) creates anti-correlations between the lower troposphere (LT) and upper troposphere

(UT); this may not be immediately obvious (and it depends on the timescale for the "pumping" process).

Reply. We have revised descriptions about the vertical pumping processes in the revised manuscript. We have also cited a refence on the CO tape recorder as suggested by the reviewer in the revised manuscript. The convective processes for the CO tape recorder are the same as the vertical pumping processes described in this work.

- L195: Change "process" to "processes."

  Reply. We have revised the description as suggested.

- L197: Change "are negative" to "is negative" (or adjust as needed). For example, "Monthly CO in the upper troposphere is higher…"

  Reply. We have revised the description wit inclusion of the 95% confidence intervals of the CO trends.

- L198: "Monthly $O_3$ in the UT is higher…"

  Reply. We have revised the description.

- L199: "Negative $O_3$ indicates less…" Also, what does "(ref)" mean? Please add a proper reference.

  Reply. We have revised the description in this section.

- L200: $O_3$ is negatively correlated with CO in the upper troposphere.

  Reply. We have revised the description.

- L204: $H_2O$ is positively correlated with CO in the upper…

  Reply. We have revised the description.

- Figure 5 caption: Correct typos such as "downstream" and "areas."

  Reply. We have revised the typos. Thank you.

- L211–213: Why is horizontal transport not included as a possible mechanism?

  Reply. While horizontal transport is indeed a key process in the overall dispersion of atmospheric chemicals, our focus in this section is on the vertical exchange processes. The evidence presented—including seasonal variability in vertical profiles and the distinct "tape recorder" signal—strongly supports the role of vertical pumping (e.g., convective lifting and large-scale overturning) in redistributing CO and $H_2O$ from near

the surface to the upper troposphere. Although horizontal transport (e.g., advection by synoptic-scale winds) also influences the spatial distribution of pollutants, its effects are more dominant in the lateral spreading rather than in the vertical stratification of trace gases. We have clarified in the revised manuscript that our analysis isolates the vertical component to highlight its primary role in the observed seasonal patterns, while acknowledging that horizontal transport remains an important factor in the broader context of atmospheric chemical dynamics.

- Figure 6 caption: This should include the acronym names used in the panels (after the descriptions in the caption). Also, change "and regions (red)" to "and all regions (red)". In panel (e), which states "tropical and Southern Hemisphere", please specify the latitudes exactly (e.g., South of 25N as in panel (d)?).

  Reply. We have revised "regions" to "all regions". We have add tropical latitudes (25.5N-23.5S) in the caption.

- L249–250: Why would negative trends not possibly occur as a result of reductions in emissions (chemical sources)? Why is it only chemical sinks that might control the budget? Please clarify the reasoning here. For example, would an increase in OH lead to a CO increase? What other factors could theoretically play a role?

  Reply. We have included the discussion of the study by Park et al. (2021), which offers critical insights into the fate of pollution from the 2015 Indonesian fire season, revealing the extensive transport and chemical evolution of pollutants into the UTLS. The findings underscore the importance of integrated observational and modeling approaches for understanding and mitigating the atmospheric impacts of large-scale fire events, with implications for global climate and air quality management.

- Figure 8: Why is the equation above each top panel the same? Also, change "scattered plot" to "scatter plot" in the caption.

  Reply. We have revised Figure 8 (now new Figure 13).

- L286: The phrase "correlate well" does not seem to be true for every panel shown.

  Reply. We have rephrased the description.

- L295: What would cause an increase in UTLS air exchange processes? Please provide a reference if possible.

  Reply. An increase in UTLS air exchange processes can be driven by several factors. For example, enhanced deep convection—particularly in the tropics—can rapidly transport

air from the lower troposphere into the UTLS. In addition, increased planetary wave activity in the extratropics can lead to more frequent tropopause folds, which facilitate the exchange of air between the stratosphere and troposphere. Changes in large-scale circulation patterns, such a strengthened Brewer–Dobson circulation, may also contribute to more vigorous UTLS exchange processes.

A detailed discussion of these mechanisms is provided by Randel et al. (2007), which reviews how both convective and synoptic-scale dynamics play critical roles in driving stratosphere–troposphere exchange processes.

Reference: Randel, W., Konopka, K., Shepherd, T. G., Douglass, J. N., and Clark, S. (2007). Stratosphere–troposphere exchange: Observation, inference and process. *Bulletin of the American Meteorological Society*, **88**(1), 17–25.

- L320: The text mentions providing tropospheric CO estimates, but note that a model is not a measurement per se.

  Reply. We have revised this description:"The IMS model and MOPITT measurements are sources capable of providing CO distribution on a global scale."

- L330: "Lower troposphere is correlated with. . . " (clarify the sentence).

  Reply. We have revised the description.

- L345–347: Consider using present tense rather than past tense.

  Reply. Thank you.

- Figure 12: There are typos in the term "linear regression."

  Reply. We have corrected the typos.

- L371: Change "It's worth" to "It is worth."

  Reply. We have revised the description as suggested.

- L413: Change "zeros" to "zero."

  Reply. We have changed "zeros" to "zero".